# Kinetic features dictate sensorimotor alignment in the superior colliculus

Ana González-Rueda[1,2✉], Kristopher Jensen[3], Mohammadreza Noormandipour[4,5], Daniel de Malmazet[1], Jonathan Wilson[1], Ernesto Ciabatti[1], Jisoo Kim[6], Elena Williams[1], Jasper Poort[6], Guillaume Hennequin[1,3] & Marco Tripodi[1✉]

The execution of goal-oriented behaviours requires a spatially coherent alignment between sensory and motor maps. The current model for sensorimotor transformation in the superior colliculus relies on the topographic mapping of static spatial receptive fields onto movement endpoints[1–6]. Here, to experimentally assess the validity of this canonical static model of alignment, we dissected the visuo-motor network in the superior colliculus and performed in vivo intracellular and extracellular recordings across layers, in restrained and unrestrained conditions, to assess both the motor and the visual tuning of individual motor and premotor neurons. We found that collicular motor units have poorly defined visual static spatial receptive fields and respond instead to kinetic visual features, revealing the existence of a direct alignment in vectorial space between sensory and movement vectors, rather than between spatial receptive fields and movement endpoints as canonically hypothesized. We show that a neural network built according to these kinetic alignment principles is ideally placed to sustain ethological behaviours such as the rapid interception of moving and static targets. These findings reveal a novel dimension of the sensorimotor alignment process. By extending the alignment from the static to the kinetic domain this work provides a novel conceptual framework for understanding the nature of sensorimotor convergence and its relevance in guiding goal-directed behaviours.

Goal-oriented behaviours rely on the integration of relevant sensory signals and their transformation into appropriate motor actions. To that end, premotor centres such as the superior colliculus (SC) need to embed connectivity matrices that ensure a meaningful alignment of spatial and motor maps[7–13]. The layered and modular structure of the SC, with its distinct sensory and motor maps[9,14–20], makes it a tractable system for the study of the connectivity rules underpinning sensorimotor convergence. It provides a conceptual blueprint for understanding the circuit and computational logic that underlies spatiomotor alignment.

To unambiguously assess the functional alignment between sensory and motor maps, one should systematically measure both motor and sensory responses, focusing particularly on the sensory tuning of well-defined motor units. However, traditionally, sensory and motor tuning within the SC have been studied either independently[16,21–23], thus precluding the identification of convergent signals, or in animals trained to orient to sensory stimuli[3,4,24–26], which impedes the unambiguous dissociation of visual responses from movement preparation signals. Moreover, these earlier studies have primarily focused on characterizing the static and spatially restricted receptive fields of visuo-motor units, while neglecting responses to kinetic visual features, which are more prominent in the intermediate and deep layers of the SC, also known as the motor layers[21,22]. This has led to the currently accepted view that sensorimotor integration in the SC results

from the systematic mapping of static sensory features, such as spatial visual receptive fields, and movement vector endpoints. Here we experimentally assessed the validity of this canonical model, first by dissecting the collicular microcircuit responsible for the transfer of visual features from the retina to genetically defined collicular motor units in vitro. Then, we characterized the responses to visual stimuli in collicular premotor units in vivo using whole-cell recordings, tetrode recordings and 2-photon Ca[2+] imaging. Finally, we characterized the conjunctive visual and motor tuning properties of identified individual collicular motor units. Contrary to the currently held view, we found that motor units in the SC have poor or no static spatial receptive fields (ssRFs) and are instead primarily tuned to kinetic visual features. Specifically, collicular motor units are preferentially tuned to visual flow of the opposite direction to the head movement that they decode. We show that a network model built on these kinetic alignment principles is ideally placed to support key ethological functions of the SC, such as prey capture, by facilitating the rapid interception of both moving and static targets in an energetically favourable manner.

In summary, these findings challenge the traditional view of the mapping of static sensory fields onto movement endpoints as the guiding principle of sensorimotor convergence and favour a kinetic model based on the alignment in vectorial space between sensory flow and movement vectors. Although in this study we focused exclusively

[1]MRC Laboratory of Molecular Biology, Cambridge, UK. [2]St Edmund's College, University of Cambridge, Cambridge, UK. [3]Department of Engineering, University of Cambridge, Cambridge, UK. [4]TCM Group, Cavendish Laboratory, University of Cambridge, Cambridge, UK. [5]Nokia Bell Labs, Cambridge, UK. [6]Department of Physiology, Development and Neuroscience, University of Cambridge, Cambridge, UK. ✉e-mail: arueda@mrc-lmb.cam.ac.uk; mtripodi@mrc-lmb.cam.ac.uk

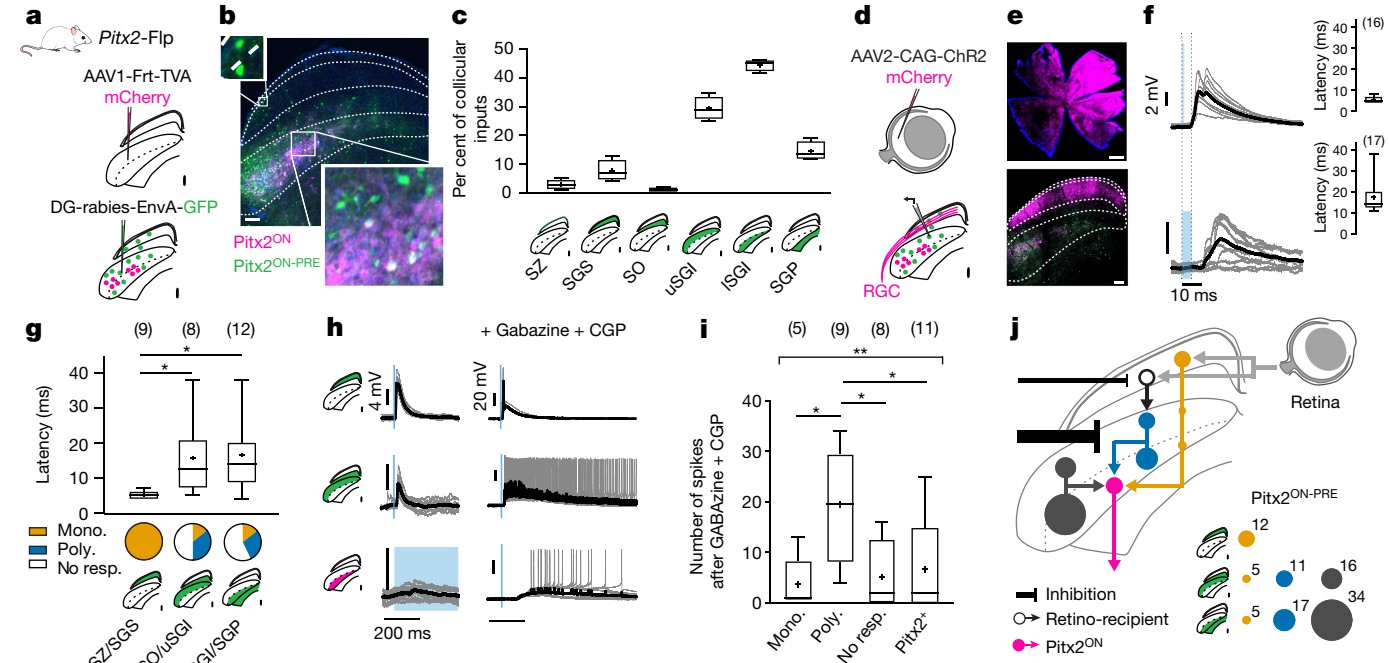

**Fig. 1 | Two functionally distinct premotor populations channel direct and indirect retinal input onto collicular motor neurons. a**, Schematic showing tracing of Pitx2[ON-PRE] neurons using glycoprotein-depleted rabies virus (DG-rabies). **b**, Immunostaining of Pitx2[ON] (mCherry) and Pitx2[ON-PRE] (GFP) neurons in mouse brain. Scale bar, 100 μm. Expanded fields show example Pitx2[ON-PRE] neurons in superficial (top) and intermediate layers (bottom). **c**, Distribution of Pitx2[ON-PRE] neurons across layers. Green shade in SC diagrams indicates the layer location of analysed neurons. $n = 4$ mice. **d**, Schematic showing injection of an adeno-associated virus (AAV) expressing ChR2 into the retina prior to rabies infection to record retinal responses. **e**, Immunostaining of retinal ganglion cells expressing ChR2 (top) and their axons in the superficial SC (bottom), showing Pitx2[ON] (mCherry in deep SC) and Pitx2[ON-PRE] neurons (GFP). Scale bars, 500 μm (top image), 100 μm (bottom image). **f**, Example traces from neurons receiving direct (top left, 1-ms light activation) and polysynaptic (bottom left, 5-ms stimulation) retinal input. Right, latency of responses. Number of neurons is shown in parentheses. **g**, Latency of response of Pitx2[ON-PRE] neurons in superficial (SZ/SGS, *sup*), upper intermediate (SO/ uSGI, *Upint*) and lower intermediate and deep layers (lSGI/SGP, *Lowint*) and the

distribution of monosynaptic (mono.) and polysynaptic (poly.) responses. One-way ANOVA, $P = 0.013$; Tukey's multiple comparison of latencies between superficial and upper intermediate premotor neurons $P_{sup-Upint} = 0.0431$, and between superficial and lower intermediate premotor neurons $P_{sup-Lowint} = 0.0161$. Number of neurons is shown in parentheses. No resp., no response. **h**, Example of the change in light-mediated responses following blockade of GABA signalling using gabazine and CGP-35348 (CGP) for Pitx2[ON-PRE] neurons in superficial (top) and intermediate layers (middle) and for a Pitx2[ON] neuron (bottom). **i**, Spiking response to 5-ms light stimulation following GABA blockade. One-way ANOVA, $P = 0.0049$; Tukey's multiple comparison: monosynaptic versus polysynaptic, $P = 0.018$; polysynaptic versus no response, $P = 0.0165$; polysynaptic versus Pitx2[+], $P = 0.018$. Number of neurons is shown in parentheses. **j**, Diagram of the visuo-motor network. With premotor neurons represented by their retinal drivers: orange, monosynaptic; blue, disynaptic; grey, none. Percentages of each neuron type estimated from **c** and **g** are shown on the right. In all box plots, the centre line shows the median, box edges delineate first and third quartiles, whiskers extend to minimum and maximum values and the cross indicates the mean.

on visuo-motor transformations in the mouse SC, the kinetic model presented here is potentially compatible with the mapping of any sensory modality onto movement vectors and might represent a general principle guiding sensorimotor convergence in other brain areas and in other species.

## Dual retino-premotor pathway in the SC

The electrophysiological and morphological properties of neurons across different layers of the SC have been widely studied[9,27–31]. However, the specific connectivity between retino-recipient neurons in the superficial layers and motor neurons in the intermediate layers remains unknown, mainly owing to the difficulty of unambiguously identifying collicular motor units. Our recent identification of genetically defined collicular motor neurons that express the transcription factor PITX2 and drive specific 3D head rotations[9] provides a useful entry point to characterize the logic of convergence of retinal input onto the collicular premotor network. Thus, we first assessed the location of neurons presynaptic to Pitx2[ON] cells (premotor collicular (Pitx2[ON-PRE]) neurons) within the SC using rabies virus-based synaptic tracing (Fig. 1a,b and Extended Data Fig. 1a). We found that Pitx2[ON-PRE] neurons were mostly located within the intermediate layers of the colliculus near to Pitx2[ON]

neurons, with a smaller fraction of about 12% of Pitx2[ON-PRE] neurons being located in the superficial visual layers (Fig. 1b,c). To assess how visual information is transmitted onto motor neurons, we then used a combination of viral tracing tools to light-activate retinal inputs while recording collicular Pitx2[ON] and Pitx2[ON-PRE] neurons (Fig. 1d and Extended Data Fig. 1c). Retinal inputs to the SC were specifically targeted by intraocular injections of AAV2-CAG-ChR2. This resulted in localized expression of ChR2 in retinal axons restricted to the superficial layers of the SC (Fig. 1e). We targeted Pitx2[ON-PRE] neurons in different collicular layers for whole-cell recording in acute brain slices and assessed the prevalence and strength of their retinal input (Fig. 1f–g). The reversal potential of monosynaptic currents following light activation indicates an exclusively excitatory retinal drive onto Pitx2[ON-PRE] (Extended Data Fig. 1d). Pitx2[ON-PRE] neurons were pooled according to their distribution across layers within three main domains: (1) stratum zonale (SZ) and stratum griseum superficiale (SGS), hereafter referred to as SZ/SGS or superficial layers; (2) the stratum opticum (SO) and upper stratum griseum intermediale (uSGI), hereafter referred to as SO/uSGI or upper intermediate layers; and (3) lower SGI (lSGI) and stratum griseum profundum (SGP), hereafter referred to as lSGI/SGP or lower intermediate and deep layers. All Pitx2[ON-PRE] neurons recorded in the superficial layers received direct monosynaptic retinal input

(approximately 5 ms latency). By contrast, 12.5% of neurons recorded in the upper intermediate layers received monosynaptic retinal input and 37.5% received polysynaptic input. Among the neurons recorded in the lower intermediate and deeper layers, 10% showed a direct monosynaptic activation and 30% showed a polysynaptic activation (Fig. 1g).

We also identified the presence of prominent tonic inhibition on Pitx2$^{ON-PRE}$ neurons by applying antagonists of type A and type B GABA (γ-aminobutyric acid) receptors (GABA$_A$ and GABA$_B$ receptors, respectively) during recordings. Specifically, intermediate Pitx2$^{ON-PRE}$ neurons receiving disynaptic retinal input were strongly disinhibited in the presence of the GABAergic receptor antagonists (Fig. 1h). This was observed as an increase in the likelihood of eliciting spikes, the number of spikes elicited (Fig. 1i) and the duration of the depolarization, which lasted up to 20 s following a 5-ms retinal stimulation (Extended Data Fig. 1j). Notably, Pitx2$^{ON-PRE}$ disinhibition led to a delayed but strong activation of Pitx2$^{ON}$ neurons following retinal activation (Fig. 1h,i). Monosynaptic SZ Pitx2$^{ON-PRE}$ neurons showed a significantly reduced response to disinhibition compared with deeper premotor and motor neurons (Fig. 1i). Although we found that half of the Pitx2$^{ON-PRE}$ neurons displayed retinal activation, this is likely to be an underestimation owing to the inability to target all retinal ganglion cells and the existence of slower response kinetics (Extended Data Fig. 1e–j).

Overall, we found that prominent retinal input reaches collicular motor neurons via two major classes of premotor neurons that mediate visuo-motor transformations. One class of premotor neurons (22% of total premotor neurons) receives direct retinal input and is located predominantly in superficial layers. The other (28% of total premotor neurons) receives disynaptic or polysynaptic retinal input, is located in the intermediate and deep collicular layers and is modulated by a tonic inhibitory gate (Fig. 1j).

## Motor layers respond to kinetic visual features

Neurons in the superficial layers of the SC display strong responses to drifting gratings and have receptive fields consistent with a retinotopic map[13,18]. To test whether collicular premotor neurons inherit any of those visual features, we first performed in vivo whole-cell recordings across visual and motor layers of the SC to assess both the subthreshold and suprathreshold tuning of collicular neurons in response to moving gratings or flashing squares (Fig. 2a and Methods).

As expected, a high proportion of neurons in superficial layers displayed prominent responses to moving gratings and well-defined spatially confined static receptive fields (ssRFs) as measured with localized flashing black or white squares (65% and 54% of neurons, respectively; Fig. 2b,d,f and Extended Data Fig. 2a,e,f). Some neurons also had morphological and physiological properties consistent with narrow-field neurons, which have been proposed to be direction-selective (DS) or orientation-selective (OS), involved in orienting behaviours[32], and to project vertically to intermediate and deep layers of the SC[27] (Extended Data Fig. 2a and Supplementary Tables 1 and 2). Indeed, we found that Pitx2$^{ON-PRE}$ neurons partially overlap with a genetically defined population of narrow-field neurons (Extended Data Fig. 1n–p). Similarly to previous studies in primary visual cortex (V1)[33], intracellular recordings in superficial layers of the SC revealed broad subthreshold tuning to all directions with a preferred tuning to the selective orientation or direction, and more sharply tuned spiking activity (Fig. 2b and Extended Data Fig. 2b).

The proportion of neurons with ssRFs was lower in intermediate and deep motor layers of the SC (11% of neurons; Fig. 2e,h). However, 56% of neurons in motor layers were still tuned to moving gratings, 20% of which displayed subthreshold tuning only (Fig. 2c,h). The greater latency and sharp subthreshold tuning in intermediate and deep layers could reflect synaptic input from the putative presynaptic narrow-field neurons recorded in the superficial layers (Fig. 2b,f and Extended Data Fig. 1p). The lack of suprathreshold tuning in those neurons is compatible with the presence of tonic inhibitory currents on the collicular premotor network (Fig. 1h), which might be further enhanced by anaesthesia. Notably, the selectivity index and the maximum response to moving gratings were significantly higher for moving than static gratings (Fig. 2g and Extended Data Fig. 2c,d).

To confirm whether kinetic features were also preferentially transferred across the SC during wakefulness, we next assessed the visual tuning of SC neurons using chronic tetrode recordings of awake head-fixed mice, while concurrently monitoring eye movements and running bouts on a wheel equipped with an accelerometer (Fig. 2i–n and Extended Data Figs. 3a–o and 4a–j). The overall proportion of OS and DS neurons decreased only slightly across layers (from 47% in superficial layers to 38% in lSGI and deep layers; Fig. 2n and Methods) but visually tuned neurons in the superficial layers were more sharply tuned (Fig. 2k) and exhibited a 1.9-fold increased firing rate at their preferred direction compared with visually tuned neurons of the lower intermediate and deep layers (Fig. 2l and Extended Data Fig. 3l). We found a bias towards DS neurons (Extended Data Fig. 3h,i), and although direction selectivity was modulated by running, this did not affect the preferred tuning of those neurons[21] (Extended Data Fig. 4d–g). Of note, a much smaller fraction of collicular neurons presented ssRFs in response to flashing squares; this fraction decreased sharply across layers, from 19% in superficial layers to 9% in lower intermediate and deep layers (Fig. 2n and Extended Data Fig. 3o).

We also assessed the responses of neurons to full-field changes in luminescence, directionally moving spots mimicking a moving target, and localized Gabor patches drifting in eight different directions (Extended Data Fig. 5 and Methods) as examples of ethologically relevant full-field static and spatially localized kinetic visual stimuli, respectively. We found that changes in luminescence elicited strong responses across the colliculus with 30% of neurons in the superficial layers and 18% in the lower intermediate and deep layer responding to either positive or negative changes in luminescence (Extended Data Fig. 5l–n). SC neurons were also tuned to target-mimicking spots moving towards and away from the centre of gaze. Although the spots only crossed 18% of the screen throughout the full stimulation protocol, 21% of all collicular neurons recorded showed a directional response to moving spots. Whereas 75% of these responses in superficial layers reflected the activation of the neuron's spatial receptive field, this decreased to 33% in lower intermediate and deep layers (Extended Data Fig. 5j). Similarly, 71% of the responses to drifting Gabor patches in superficial layers corresponded to the activation of spatial receptive fields, whereas this proportion decreased to 24% in lower intermediate and deep layers, where DS and OS neurons were more prevalent (Extended Data Fig. 5g–i), further strengthening the notion that kinetic features of the visual scene are preferentially transmitted to the motor layers of the SC. Notably, we found a conservation of the preferred tuning direction among stimulation paradigms, meaning that neurons tuned to a particular direction of gratings would be tuned to moving spots and drifting Gabor patches of the same direction (average preferred gratings direction − spots direction = 30°, average preferred gratings direction − Gabor direction = 27°; Extended Data Fig. 5k).

To confirm whether premotor neurons encode visual kinetic features, we leveraged two newly developed tools—intersectional gene expression[34] and self-inactivating rabies (SiR)[35–37]—that enabled us to selectively record from genetically defined Pitx2$^{ON-PRE}$ premotor neurons. We performed an initial injection with a mix of an AAV with a FLEX-dependent conditionally expressed TVA receptor together with either an AAV containing Flp-dependent GCamp6f, or an AAV with Cre$^{OFF}$/Flp$^{ON}$-ChR2 in Pitx2-cre::tdTomato mice, followed by a second injection with a SiR expressing the Flp recombinase (Fig. 2o). We recorded Pitx2$^{ON-PRE}$ neurons using either optetrodes, enabling us to also record the same neurons in freely moving mice, or under a 2-photon microscope, yielding higher numbers of recordings to assess their visual tuning properties in superficial and upper intermediate layers.

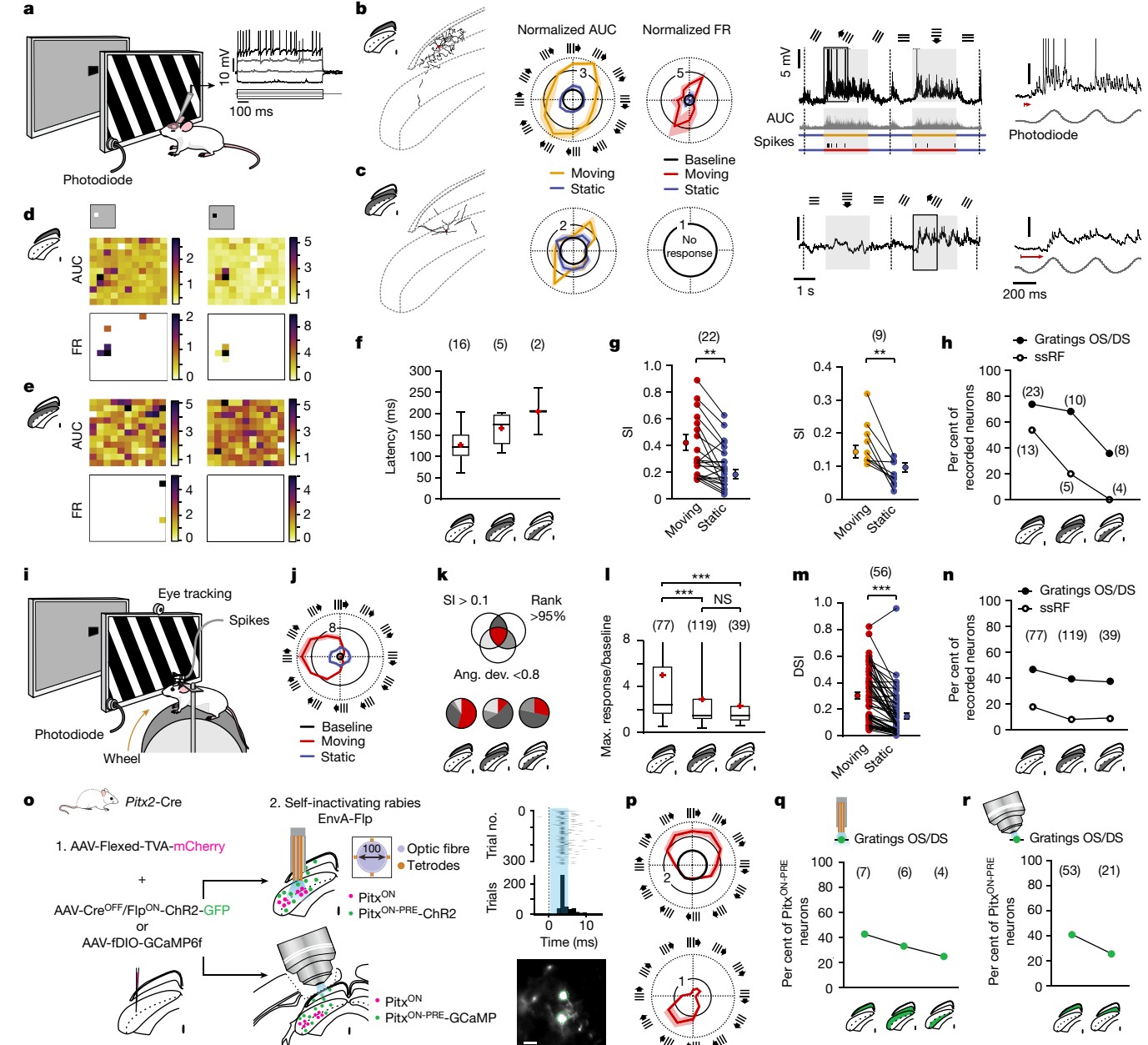

**Fig. 2 | Premotor neurons and neurons in motor layers respond preferentially to visual flow. a**, Experimental schematic (left) and intracellular responses to visual stimuli (right; $n = 58$ neurons). **b**, Example narrow-field neuron (left), and input (area under the curve (AUC)) and output (firing rate (FR)) (middle; example extracted AUC and spikes shown below the graph) tuning to moving gratings. Parallel lines and arrows indicate orientation and movement of the grating. Shaded areas represent s.e.m. The region outlined in the graph is expanded on the right; the red arrow indicates the latency. **c**, Same as **b**, for a neuron in the intermediate layer. **d**,**e**, Responses to white (left) or black (right) localized flashing squares for a neuron in superficial (**d**) or intermediate (**e**) layers (colour indicates change in input over baseline (AUC, top) or firing rate (Hz, bottom)). **f**, Latency to grating response across layers. **g**, Selectivity index (SI) of tuned neurons (SI > 0.1) for spiking (left; $P = 0.0019$) and subthreshold (right; $P = 0.0010$) activity. Two-tailed paired $t$-test. **h**, Distribution of neurons tuned to gratings and/or squares (ssRF) across layers. **i**, Experimental setup. Results were reproduced in 5 awake mice using tetrodes ($n = 238$ neurons).

**j**, Response in an example DS neuron. **k**, Tuning to gratings was determined by the SI (SI > 0.1), its rank (higher than 95% of the SIs of a shuffled distribution) and the trial-to-trial variability (angular deviation (ang. dev.) < 0.8 rad). **l**, Response to the preferred direction. Kruskal–Wallis test: $P = 6.11 \times 10^{-5}$; Dunn's multiple comparison: $P_{sup\text{-}Upint} = 0.007$, $P_{sup\text{-}Lowint} = 0.004$. Max., maximum; NS, not significant. **m**, Selectivity index for DS neurons (DSI). Two-tailed paired $t$-test: $P = 3.50 \times 10^{-11}$. **n**, Same as **h**, with tetrodes. **o**, Left, viral strategy for selective recording of Pitx2$^{\text{ON-PRE}}$ neurons with optetrodes or Ca$^{2+}$ imaging. Top right, opto-tagged neuron detection (raster of the 300 first 5-ms light stimulations and peri-stimulus time histogram for all 900 light stimulations). Bottom right, example 2-photon image of Pitx2$^{\text{ON-PRE}}$ neurons. Scale bar, 100 μm. **p**–**r**, Tuning of Pitx2$^{\text{ON-PRE}}$ neurons recorded using either optetrodes (top) or Ca$^{2+}$ imaging (bottom) (**p**) and their distribution across layers, determined using tetrodes (**q**; $n = 10$ mice) or 2-photon microscopy (**r**; n = 7 mice). **f**–**h**,**l**–**n**,**q**,**r**, Number of neurons is shown in parentheses.

We found the same preservation of tuning to kinetic features across layers of the SC as previously found in wild-type mice (Fig. 2p–r and Extended Data Fig. 6). Together, these data show that premotor neurons

in lower intermediate and deep layers of the SC receive preferential input reporting kinetic features of the visual scene, such as externally generated motion flow. This is surprising given that visual-motor

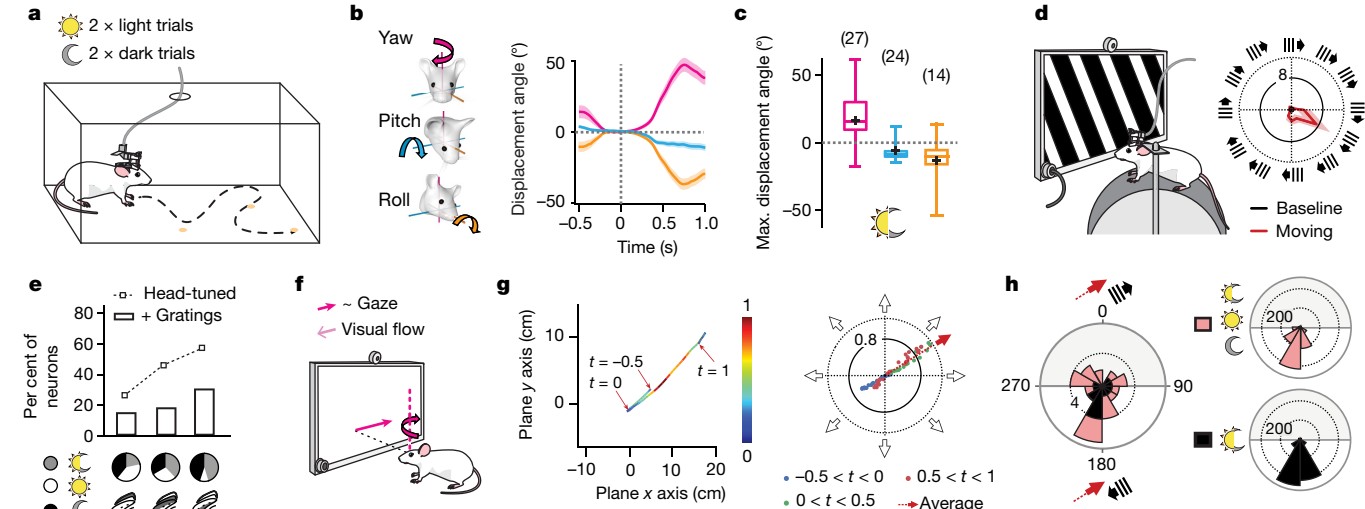

**Fig. 3 | Collicular motor units respond preferentially to externally generated visual flow of the opposite direction to their motion vector tuning. a**, Experimental scheme. Mice were recorded in an open-field arena with a white background while foraging for condensed milk droplets; twice (×5 min) with lights on and twice (×5 min) in the dark (*n* = 331 neurons recorded from 8 mice). **b**, Example spike-triggered averages (STAs) of the head-displacement angles along yaw, pitch and roll for a motor unit conserving its tuning in all four trials. **c**, Decoded head-displacement angle for all motor units (tuned in both light and dark). Number of neurons is shown in parentheses. **d**, Mice were also head restrained to assess neuronal response to visual stimuli. The tuning to moving gratings of the neuron in **b** is shown. **e**, Distribution of neurons tuned to head displacements (squares) or those tuned to both head displacements and to moving gratings (bars) across layers. Pie charts show the proportion of neurons tuned in light, dark or both light and dark. **f**, 3D head rotations measured as STAs were transformed into the same 2D coordinate system in which gratings are displayed (gaze direction). An example yaw rotation is depicted. **g**, Left, 2D projection of the STA in **b**. The colour scale indicates the normalized velocity of the movement vector used to compute the weighted average of the vector's direction (right; *t* = 0 corresponds to time 0 in **b**). **h**, Left, circular distribution of the difference between gaze direction as computed in **g** and the preferred angle to visual gratings for all visuo-motor neurons (0 equals 0° difference; *n* = 33, light red) and motor neurons only (*n* = 13, black). Right, the distribution of the circular means of 1,000 bootstraps with replacement (for all neurons: 196 ± 2°, for motor units: 187 ± 2°, error bars show 95% confidence interval).

integration in the SC has generally been hypothesized to rely on the alignment of static sensory and motor features, such as the stimulus spatial receptive field and movement endpoints[1–6]. These findings suggest the need to reconsider the computational and circuit strategies underpinning the visuo-motor alignment and the visuo-motor transformation by taking into account the kinetic domain.

## Kinetic anti-alignment of visuo-motor neurons

To characterize the degree of visuo-motor alignment, we next identified and recorded motor units tuned to 3D head rotations in awake freely moving mice. Head-movement-related neurons were identified on the basis of their firing activity while the mouse was freely foraging for droplets of condensed milk in a white open-field arena with minimal external visual information in both dark and light conditions[9,38]. Motor units, consistently tuned to specific head rotations in all light and dark trials, were—as a population—tuned to the expected head-movement vector encoded in the area of the SC used for recordings[9], which was also consistent across trials (Fig. 3a–c and Extended Data Fig. 7a–d). As expected, motor units were primarily located within the deep layers of the SC (representing 7% of neurons in putative SZ/SGS, 16% of neurons in SO/uSGI and 25% of neurons in lSGI; Fig. 3e). We also identified neurons that were tuned only in light or only in darkness that were preferentially located in more superficial layers (Fig. 3e and Extended Data Fig. 7a–k); however, we focused on well-defined motor units to avoid the confounding effect of purely visually driven responses. To confirm the absence of visually driven responses in motor units during foraging, we enforced self-generated visual flow in the open arena by covering the walls with a pattern of either horizontal or vertical bars. As expected, motor tuning was conserved in all tested conditions (Extended Data Fig. 8a–c).

Next, we assessed the visual response properties of the same units under head restraint. We measured their responses to static and moving sinusoidal gratings as well as their receptive field responses (Fig. 3d,

Extended Data Fig. 3 and Methods). Surprisingly, around half of all motor-tuned neurons across all layers were also tuned to moving gratings (60% in putative SZ/SGS, 46% in SO/uSGI and 55% in lSGI; Fig. 3e), preferentially displaying direction selectivity, and none had detectable ssRF. Notably, we found that none of the visuo-motor neurons recorded were tuned to eye movements in head-restrained conditions (Extended Data Fig. 4h–j).

To determine whether a coherent alignment exists between the head motion vector that these neurons decode and their preferred direction tuning to drifting gratings, we built a geometrical model of the head and eye of the mouse that enabled us to infer the gaze path resulting from any given movement of the head. Assuming a resting position of the mouse's head in front of the screen at the time of spike (Extended Data Fig. 9e), equivalent to that under head restraint, we translated the time course of yaw, pitch and roll angles of head rotation decoded from each single unit into an equivalent time course of gaze location on the screen (Fig. 3f). We then estimated the average direction of this gaze path which could be directly compared to the direction of a drifting grating (Fig. 3g). We subtracted the decoded gaze motion direction (in degrees) from the preferred grating direction of each visuo-motor unit. The resulting angle would be 0° if head movement and preferred gratings directions were perfectly aligned, and 180° if they were anti-aligned (Fig. 3h). The recorded motor units exhibited visual tuning largely anti-aligned to their motor tuning (190°; Rayleigh *P* = 0.03; Fig. 3h). This empirical distribution of sensorimotor alignment was indistinguishable from a random distribution with equal s.d. centred at 180°. However, it differed significantly from control distributions centred at 0°, 90° or 270° (Watson–Williams *F*-statistic 0.24 at 180° versus 43.72 at 0° for a critical value of 3.85; Extended Data Fig. 9a–d). The same anti-alignment was found in genetically defined Pitx2[ON] motor neurons (Extended Data Fig. 6a–c) and for neurons tuned to both head rotations and moving spots (Extended Data Fig. 9j). We also found an alignment between gaze direction and preferred Gabor patch location

for neurons tuned to head rotations and Gabor patches (Extended Data Fig. 9k). However, more than 90% of these units maintained their motor tuning only in light conditions but lost their head-rotation tuning in darkness, implying that, although these are not canonical motor units, they are reminiscent of visuo-motor units found in primates[39].

In freely moving conditions mice can display both gaze-shifting saccades and compensatory eye movements, aligned and anti-aligned to head rotations, respectively[40,41], with the former being predominant during goal-oriented behaviours. Although all visuo-motor units were recorded during goal-oriented foraging, we also considered the effect of purely compensatory eye movements during head rotations by implementing our geometrical model of the mouse gaze with head–eye rotations measured during a foraging task (Extended Data Fig. 9f–h). Incorporating these compensatory eye movements in the model estimating the average gaze direction did not change the results (Extended Data Fig. 9i).

Overall, these data point to a systematic congruence in kinetic space between visual and motor features such that the movement vector tuning of motor collicular units is anti-aligned to their tuning to externally generated visual flow. These findings suggest that the connectivity of the visuo-motor system in the SC is constrained within coherent visuo-motor direction columns[9,18], a hypothesis that is also supported by our assessment of the visual tuning preferences of the premotor network, whereby neurons presynaptic to Pitx2 neurons tend to cluster in direction coherent columns (Extended Data Fig. 6l,m). Given the lateralization of both visual and motor responses in the SC, the uncovered alignment would favour the rapid interception of targets moving towards the centre of the visual field, suggesting a potential ethological advantage for such an arrangement in prey capture-like scenarios.

## The kinetic model supports target interception

To assess quantitatively how these connectivity premises can be exploited to direct spatially tuned actions towards moving and static targets, we built a three-layer network model of the left SC composed of a ssRF layer, a DS layer and a motor layer. The connectivity between these layers was strictly top-down and constrained by the experimentally observed connectivity, visual tuning and sensorimotor alignment (Fig. 4a). We also incorporated in the model the experimental observation that OS and DS neurons in the superficial layer of the SC are arranged in a centripetal pattern[18,19,42], a finding also confirmed in our recordings (Fig. 4a, Extended Data Fig. 10a–c). We used this three-layer network to systematically compare how a direct retino-premotor pathway ('kinetic pathway', orange; Fig. 1j and Fig. 4a), that relies exclusively on kinetic information of the visual target, or an indirect retino-premotor pathway ('static pathway', blue; Fig. 1j and Fig. 4a), that relies on ssRF activation, would enable interception of moving and static targets.

In the model that engages the kinetic pathway, a moving target activates neurons in the DS layer that are tuned to the radial component of target motion. In turn, these DS neurons excite those motor units with opposite (anti-aligned) preferred movement vector. This operating regime supports fast interception of moving targets if the target motion includes a radial component towards the agent. When quantifying the accuracy of this model by the Euclidean distance to the target at the end of each simulated trial, we found it to be comparable to that of a canonical static model that first engages ssRF neurons (Fig. 4d,g,i).

Of note, a visuo-motor anti-alignment was the only alignment that supported target interception and was required when either of the two pathways is engaged. When we consistently shifted the alignment between the DS layer and the motor layer, the agent failed to reach the target (Fig. 4h).

The model is also compatible with reaching static targets. In this scenario, the location of the visual stimulus would drive the appropriate neurons in the ssRF layer. The ssRF neurons then feed into the DS layer according to the topographic constraints dictated by concentricity, and the DS neurons activate the motor neurons on the basis of the kinetic alignment constraints described above (Fig. 4e,f).

Because our network is constrained by the concentricity of the encoded visual features, we found that the kinetic pathway prioritizes the interception of targets with the highest chances of success by ignoring targets that are moving away from the agent. This feature of the model improved energy efficiency as measured by the total amount of movement produced over all attempted interceptions (Fig. 4j,k). Although our results indicate a preferential bias towards fast, energy-efficient target interception, the model is also consistent with tracking and reaching of eccentric targets if the retinotopic layer can produce enough drive, in agreement with our finding of a strong inhibitory gate primarily impinging on this pathway (Fig. 1h–j) and affecting direction and orientation selectivity (Extended Data Fig. 3p–s).

Overall, these simulations show that a collicular network whose sensorimotor connectivity is constrained by the type of kinetic alignment that we have uncovered experimentally here is able to sustain key ethological functions that are normally associated with collicular activity, such as prey capture.

## Discussion

Understanding the circuitry and computational logic underpinning the transformation of sensory signals into motor commands is essential to understand how animals interact with the environment. The lower intermediate and deep layers of the SC are involved in the translation of sensory signals into motor commands for the control of eye and head movements and provide a well-defined system to characterize the general logic of sensorimotor transformations. However, until now, it has remained unclear what type of sensory stimuli are conveyed to collicular motor units, through what circuitry, and what type of spatial alignment exists, if any, between the encoded sensory responses and the decoded movement vectors. Understanding these aspects—and particularly the issue of the geometrical alignment between sensory and motor signals in well-defined motor units—is key to understanding how spatio-temporal information, as conveyed by the sensory system, is translated into appropriate motor commands.

Here, we reveal a disynaptic collicular pathway that relays visual information from the eye on collicular motor units. We show that motor-tuned collicular units respond primarily to kinetic visual features. Specifically, we have uncovered the existence of a counterintuitive spatial visuo-motor alignment whereby motor-tuned collicular units preferentially receive visual flow input of the opposite direction to the movement vector that they decode. Although our superficial layer recordings support the existence of a saliency map in retinotopic space[43,44], we also uncovered a previously overlooked feature-biased saliency-to-action map conversion in deeper layers of the SC. Essentially, the logic governing sensorimotor transformation in the SC is not driven primarily by the spatial mapping of visual receptive fields and moment vector endpoint, as commonly hypothesized[1–6]; instead, it is driven by the alignment of externally generated visual flow and movement direction.

The broadly accepted model of a mapping of spatial static visual receptive field onto movement vector endpoint that we challenge here seems to originate primarily from two lines of evidence: first, from the observation that the map of visual receptive fields in the superficial SC is broadly aligned with the movement vector map of the intermediate layers; and second, from the apparently narrow receptive field of visuo-motor cells as observed in classical primate studies[3,45,46]. However, a topographic alignment of superficial receptive field and deeper movement vector maps in the SC is not in itself an indication of direct connectivity and, although prominent in superficial layers, localized visual receptive field responses tend to disappear deeper in the SC[22], weakening the support for direct flow of information from ssRF responsive neurons to collicular motor units. Importantly, the topographic

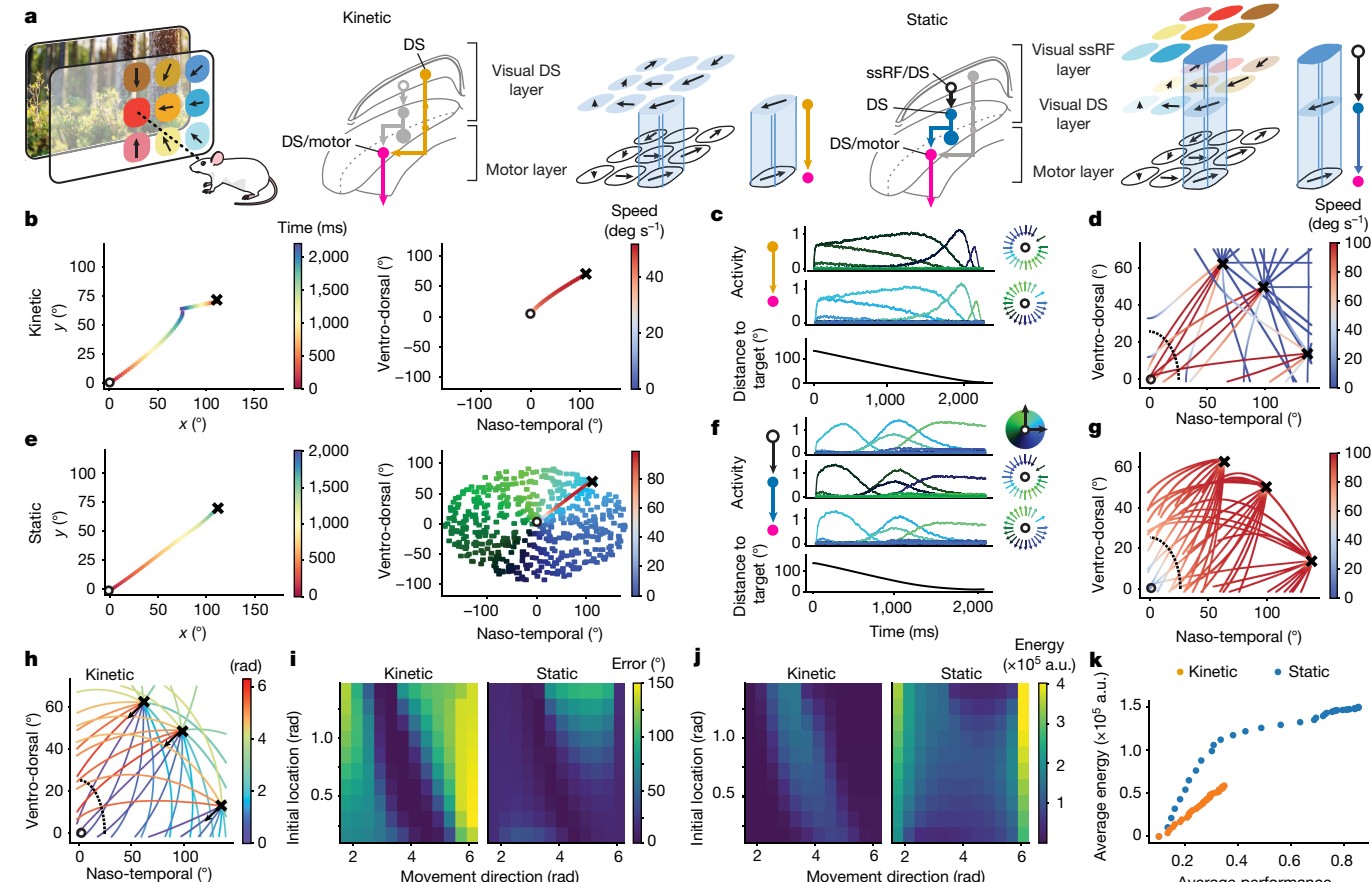

**Fig. 4 | The direct kinetic visuo-motor pathway model supports energy-efficient and fast target interception. a**, We built a three-layer feedforward neuronal network model of the left SC ($n$ = 500 neurons per layer). Left to right: colours and arrows representing ssRFs and concentric direction selectivity; visual information of a moving target could either activate DS neurons anti-aligned with motor output neurons (kinetic model); or activate ssRF neurons presynaptic to DS neurons (static model). **b**, Example simulations of the kinetic model, in which the agent (circle) orients towards a moving target (cross), plotted in absolute angular coordinates (left; displaying the trajectories of both target and agent) and in angular coordinates relative to the agent's position (right; colour represents the agent's angular speed). **c**, Activities of ten example neurons in the DS (top) or motor (middle) layers and the distance to target (bottom) for the simulation in **b**. Colour represents the encoded direction of the encoded concentric or motor vector. **d**, Example relative trajectories for 3 targets moving 3× slower than the agent in 15 directions spanning 360° (colour represents the agent's speed). **e**, Example simulations using the static model, with interception of a static target. The colour-coded receptive fields of all 500 neurons in the ssRF layer is displayed behind the relative trajectory. **f**, Example activities of neurons across layers for the simulation in **e**. **g**, Same as **d** but using the static model. **h**, Example relative trajectories for 3 targets moving 3× slower in the direction shown by the arrow, repeated 12 times with different alignment shifts between the DS and motor layer (30° steps; anti-alignment = 0 shift, colour represents shift in radians). **i**, Error of interception (°) for 10 initial locations and 12 target movement directions. **j**, Energy expenditure (a.u. ∝ J) during the simulations in **i**. **k**, Average energy expenditure per relative speed versus average performance (fraction of successful interceptions) for 12,000 simulations (both models, 40 relative speeds: 0 to 6 at 0.13 steps, each at 10 start positions and 15 directions of movement).

nature of the visual receptive field maps is mirrored in the topographic alignment of DS and OS neurons in the superficial SC[17,19,42], which we propose ultimately mediates the sensorimotor transformation process. Indeed, in our network model, the concentricity of direction-selective responses is sufficient to mediate target-directed movements within the proposed kinetic alignment framework. The narrow receptive field of putative visuo-motor cells previously described in primates has been characterized in trained animals during target-directed tasks. Therefore, it is difficult to decouple the sensory response to the cue from the underlying motor programme[47,48]. Indeed, during the execution of spontaneous saccades (that is, not in target-driven tasks), the motor tuning preferences of bona fide visuo-motor cells are only maintained in light trials, casting a doubt on their true motor nature[39]. Instead, even in primates, units that exhibit motor tuning in both dark and light trials (which one would regard as pure motor units) do not show obvious spatial receptive field responses[39].

Our study provides experimental support to an alternative kinetic alignment model, whereby DS visual neurons in the superficial and

upper intermediate layers of the SC selectively converge onto collicular motor units responsible for driving movement in the direction opposite to the incoming sensory flow. We show that a network model built on this logic is able to perform spatially accurate visually guided actions such as visual grasping towards static and moving targets. The SC has been implicated in other ethologically relevant behaviours such as escape from predators[12,49,50]. For simplicity, our model of the left SC is built in homogeneously distributed columns spanning the whole SC; however, there are input–output differences in different areas of the SC[51], which have also been associated with different behaviours. Here we have recorded in the lateral-temporal area, which preferentially encodes visual information of the lower visual field, linked to predation, and where genetically defined goal-oriented motor units are enriched[9,49,52]. Moreover, we have only tested motor tuning during goal-oriented foraging. Although the alignment that we describe could also potentially account for orienting towards shelter[53] or for the preferred direction of escape depending on the angle of approach of a predator, this remains to be tested. Furthermore, we propose that the

kinetic anti-alignment that we have uncovered in afoveate mice could also expand to other systems and be the substrate of other ethologically relevant goal-oriented behaviours observed in the animal kingdom, such as the preferential directional flight of flies towards incoming odours[54] or the bias towards upstream-orienting movements in swimming and flying animals nearing a final goal destination[55].

The proposed kinetic alignment framework, which also recognizes the existence of a localized receptive field map in superficial layers and predicts a receptive field-to-DS-to-motor alignment, would fit well not only with early studies on sensorimotor transformation in the SC but also with two recent discoveries about the organization of visual and motor responses in the SC. Namely, the existence of orientation and direction columns in the superficial layer of the SC and their concentric organization[17–19,22,42] as well as the discovery of topographically arranged motor columns in the intermediate layers of the SC[9]. The topographic motor organization shown by our model matches that previously found for head movements in mice[9] and saccadic movements in monkeys[13]. With that regard, it is possible that the alignment of these two modular systems might provide the anatomical substrate for the selective sensorimotor convergence that we have unravelled here.

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

## Methods

### Mice

Seven- to fourteen-week-old C57BL/6 wild-type, *Pitx2-cre::Tau-LSL-FlpO-INLA* (*Pitx2-Flp*, derived from *Pitx2-cre* and *Tau-LSL-FlpO-INLA* mice, provided by J. Martin and S. Arber, respectively), *Pitx2-cre::Rosa-LSL-tdTomato* (derived from *Pitx2-cre* and *Rosa-LSL-tdTomato* mice; 007914, The Jackson Laboratory) or *Vgat-cre* (016961, The Jackson Laboratory) mice were used for this study. Mice of both sexes were used for anatomical experiments and only males were used for behavioural experiments. All animal procedures were conducted in accordance with the UK Animals (Scientific procedures) Act 1986 and European Community Council Directive on Animal Care under project license PPL PCDD85C8A and approved by The Animal Welfare and Ethical Review Body (AWERB) committee of the MRC Laboratory of Molecular Biology. Animals undergoing surgical implantation were individually housed to prevent damage to implants. Lighting was set to a reversed light:dark cycle, with simulated dawn and dusk at 19:00 and 07:00, respectively. Temperature was controlled at 19–23 °C and humidity at 45–65%. For open-field recordings, mice were placed on a restricted diet sufficient to maintain 85% of their free-feeding weight. When possible, analyses were blinded to data collection. While sample sizes were not determined using statistical methods, we selected sample sizes for each experiment based on variance observed in similar studies[9,38] and on practical experimental considerations.

### Surgery

Mice were anaesthetized with isoflurane. Upon cessation of reflexes, the top of the head of the mouse was shaved, the mouse was placed on a stereotaxic frame and the skin was opened with a scalpel in a single clean vertical cut.

Viral injections in the brain were done using a nanoject (Scientific Laboratory Supplies) equipped with a pulled borosilicate glass capillary (1.5 outer diameter × 0.86 internal diameter × 100 mm, Harvard Apparatus). Up to a maximum of 300 nl of virus were injected at 3 heights within the SC (coordinates: 3.80 AP, 1 ML, 1.2, 1.5 and 1.8 DV) at a rate of 5 nl every 5 s. For retrograde tracing, AAV(1)-CMV-FRTed-TVAmCherry-2A-Gly (500 nl, titre: $3.9 \times 10^{12}$ genomic copies per ml) was injected at day 0, followed by injection of virus DG-rabies-GFP(EnvA) (500 nl, titre: $4.3 \times 10^8$ infectious units per ml) at day 21 through the same craniotomy. Mice were perfused one week after injection of rabies virus. Brain tissue was processed as described in 'Histology'. For selective long-term labelling of Pitx2[ON-PRE] neurons in the SC, we performed an initial injection with a mix of AAV(2)-hSyn1-FLEX-nucHA-2A-TVA-2A-G(N2c) (500 nl, titre: $1.5 \times 10^{12}$ genomic copies per ml) and either AAV(1)-pAAV-nEF-Cre[OFF]/Flp[ON]-ChR2(ET/TC)-EYFP (500 nl, Addgene #137141, titre: $4.3 \times 10^{12}$ genomic copies per ml) or AAV(1)-Ef1a-fDIO-GCaMP6f (500 nl, Addgene #128315, titre: $5 \times 10^{12}$ genomic copies per ml). Following 3 weeks of expression, a second injection with a self-inactivating rabies virus[35–37] SiR-N2c-Flp (EnvA) (500 nl, titre: $1 \times 10^7$ genomic copies per ml) was performed before implanting an optetrode or a cranial window for chronic Ca$^{2+}$ imaging (see below). In a subset of experiments AAV(9)-EF1a-double floxed-hChR2(H134R)-mCherry-WPRE-HGHpA (800 nl, Addgene #20297, titre: $5 \times 10^{12}$ genomic copies per ml) was injected in the SC followed by optetrode implant. For Ca$^{2+}$ experiments injections were performed at 3 different locations surrounding the area to be imaged, namely: 3.2, 3.7 and 4.2 mm posterior from Bregma, 0.8, 1.5 and 0.8 mm lateral of the midline and 1.3, 1.5 and 1.2 ventral to the brain surface. For a subset of experiments, the right eye of the mouse was injected with an AAV(2)-CAG-ChR2(H248R)-mCherry (1.5 ml, titre: $3.9 \times 10^{12}$ genomic copies per ml). A drop of 1% tropicamide and another of 2.5% phenylephrine hydrochloride were applied on the eye before injecting up to 3 μl of the virus with a Hamilton syringe. Following viral injection, a drop of 0.5% proxymetacaine

and another of 0.5% chloramphenicol were applied to the injected eye. Acute brain slices were performed 4 weeks post injection.

For all in vivo recordings, the skin covering the left hemisphere was removed and a craniotomy covering 3.5–3.8 mm AP and 0.8–1.2 ML was performed and custom-made head plate was cemented (Super-Bond C & B; Prestige Dental) around the craniotomy. Care was taken to avoid bleeding or drying of the meninges and brain tissue. For whole-cell recordings in anaesthetized mice, saline (0.9% NaCl) was superfused constantly with a 2 ml min$^{-1}$ laminar flow using a peristaltic pump and body temperature was maintained at 36 °C using a low-noise heating pad (FHC, Termobit).

For tetrode recordings, mice were implanted with moveable 17-mm-diamteer platinum-iridium (H-ML insulated) microelectrodes (California Fine Wire), configured as four tetrodes and carried by 16-channel microdrives (Axona) and with a custom-made head plate. Tetrodes were platinum electroplated to an impedance of 100–250 kΩ using a Kohlraush:gelatin (9:1, 0.5% gelatin) solution. Electrodes were implanted at the surface of the SC: at coordinates 3.8–4.2 mm posterior from Bregma, 1.25 mm lateral of the midline and 1.2 mm ventral to the brain surface. All mice were given at least one week to recover before recording and food deprivation. The same protocol was used for optetrode surgery. For in vivo chronic selective Ca$^{2+}$ imaging of Pitx2[ON-PRE] neurons in the SC recordings, mice underwent an initial viral surgical injection with an AAV-TVA and an AAV-GCaMP6f as described above and were injected with Dexafort at 2 μg g$^{-1}$ the day prior to surgery. A head post and a cranial window over the SC were implanted as previously described[17]. In brief, following isoflurane anaesthesia, Vetergesic was injected subcutaneously at 0.1 mg kg$^{-1}$ and a metal head post was affixed to the skull with Crown & Bridge Metabond. Epivicaine was splashed over the skull, and a 3-mm-diameter craniotomy was performed on the left hemisphere, centred on the rostral SC. The surface of the SC was exposed through removal of the overlying cortex and the SiR was then injected as described above. A 3-mm cannular window was then fixed on top of the colliculus using dental cement (Crown & Bridge Metabond).

### Histology

Once tetrodes were estimated to have passed beyond the SC, mice were anaesthetized with Euthatal (0.2 ml) and transcardially perfused with 4% formaldehyde in phosphate buffered saline (PBS). Brains were stored in the fixative and then 30% w/v sucrose solution for 24–48 h in order to cryoprotect the tissue. Brains were subsequently embedded in O.C.T. (VWR), frozen to −20° and cut in 30 μm coronal sections using a CM1950 cryostat (Leica). Nissl staining was used to determine tetrode depth as previously described[38].

For immunohistochemistry (IHC) experiments, 40 μm cryo-sections were performed. Free-floating sections were rinsed in PBS and incubated in blocking solution (1% donkey serum and 0.3% Triton X-100 in PBS) containing primary antibodies for 24 h at 4 °C. Sections were washed with PBS four times at room temperature and incubated for 24 h at 4 °C in blocking solution with secondary antibodies. Immunolabelled sections were washed four times with PBS at room temperature and mounted on glass slides (SuperFrost Plus, Thermo Scientific) using DAPI Fluoromount-G (SouthernBiotech). Biocitin-filled neurons were manually traced and aligned across sections to obtain the final reconstruction. Primary antibodies used in this study were: chicken anti-GFP (Aves Labs, GFP-1020, 1:2,000) and rabbit anti-RFP (Rockland, 600-401-379, 1:2,000). Secondary antibodies used were Alexa Fluor 488 donkey anti-chicken (Jackson ImmunoResearch, 703-545-155, 1:1,000), Cy3 donkey anti-rabbit (Jackson ImmunoResearch, 711-165-152, 1:1,000) and Alexa Fluor 488-conjugated streptavidin (Invitrogen, 1:2,000). Images were acquired using a Zeiss780 confocal microscope using a 20×/0.8 NA air lens (Carl Zeiss).

Retinas were dissected from eyecups, incubated in 4% formaldehyde for 24 h. Following several washes in PBS full retinas were mounted for imaging.

## Electrophysiology

**Whole-cell.** For whole-cell recordings, coronal slices (350 μm) containing the SC were prepared using a vibrating microtome (7000smz-2, Campden Instruments) in ice-cold sucrose-based cutting solution oxygenated with carbogen gas (95% $O_2$, 5% $CO_2$) and with the following composition (in mM): KCl 3, $NaH_2PO_4$ 1.25, $MgSO_4$ 2, $MgCl_2$ 1, $CaCl_2$ 1, $NaHCO_3$ 26.4, glucose 10, sucrose 206, ascorbic acid 0.40, kynurenic acid 1. Slices were incubated at 37 °C for 30 min in a submerged-style holding chamber with oxygenated artificial cerebrospinal fluid (aCSF; in mM: NaCl 126, KCl 3, $NaH_2PO_4$ 1.25, $MgSO_4$ 2, $CaCl_2$ 2, $NaHCO_3$ 26.4, glucose 10) with an osmolarity adjusted to 280–300 mOsm $l^{-1}$ and stored thereafter in the same holding chamber at room temperature for at least a further 30 min. Slices were individually transferred to the recording chamber and were superfused with oxygenated aCSF at room temperature at a flow-rate of approximately 2 ml $min^{-1}$. To block GABAergic receptors CGP (52431, 10 μM, Tocris Bioscience) and gabazine (SR95531, 10 μM, Tocris Bioscience) were diluted into the superfusate.

Whole-cell current-clamp recordings were obtained from collicular neurons using 5–8 MΩ pipettes pulled from borosilicate glass capillaries (1.5 mm outer diameter × 0.86 mm inner diameter). Pipettes were filled with artificial intracellular solution containing (in mM): potassium gluconate 150, HEPES 10, NaCl 4, magnesium ATP 4, sodium GTP 0.3 and EGTA 0.2, 0.4% biocitin; adjusted to pH 7.2 and osmolarity 270–290 mOsm $l^{-1}$. Data were recorded using an Axon Multiclamp 700B amplifier (Molecular Devices) and signals were low-pass filtered at 2 kH and acquired at 5 kHz using a digitizer (Axon Digidata 1550 A, Molecular Devices) on a PC running pClamp. Light-evoked responses were elicited using a 450–490 nm LED light (pE-300 coolLED system, Scientifica) through a 40× water immersion objective (0.8 NA).

Whole-cell in vivo recordings were performed under isoflurane anaesthesia as previously described[56]. In brief, the recording pipette was placed perpendicularly to the brain surface and a reference silver pellet electrode (A-M Systems) was placed in the saline bath covering the craniotomy. High positive pressure (>500 mbar) was applied before lowering the pipette to the surface of the brain. A 5-ms-long square pulse of voltage of 4 to 8 mV at 100 Hz was delivered via the recording electrode. The pipette was quickly advanced ~1.2 mm to reach the surface of the SC. The pressure was lowered to 40–60 mbar and the pipette was advanced in 2-μm steps. Cell contact produced a small reduction (around 10%) in resistance of the pipette, which could be seen as a proportional decrease in the size of the step of current in the oscilloscope.

**Tetrodes.** Single-unit recording was carried out using a multi-channel DacqUSB recording system (Axona) as previously described[38]. The microdrive and head stage were attached to the pre-amplifier via a lightweight cable. Signals were amplified 12–20,000 times and bandpass filtered between 500 Hz and 7 kHz. Recording thresholds were set to 70% above baseline activity levels, and data from spikes above the threshold from all channels were collected across a period spanning 200 μs preceding and 800 μs following the peak amplitude of a spike. The activity of channels from any given tetrode was referenced against the activity of a single channel from another tetrode to increase the signal-to-noise ratio. Tetrodes were advanced ventrally into the brain by ~50–75 μm after each recording session. The inertial sensor was attached to the head stage on the head of the mice using Mill-Max connectors. The signal from the sensor was passed through a lightweight cable via one Arduino for processing the signal and computing the direction cosine matrix algorithm. The control Arduino was connected to the DacqUSB system using the system's digital input–output port. A custom BASIC script was written in DacqUSB to synchronize the start of single-unit recording with the key-press initiation of inertial sensor recording (controlled using the Processing software sketchbook; https://processing.org) or visual stimulation. During head-restrained recordings, a TTL pulse was sent to an infrared LED to align eye tracking and electrophysiology.

**Optetrodes.** For simultaneous single-unit recording and optogenetic stimulation, a single optic fibre (core = 100 μm, NA = 0.22; Doric Lenses) was inserted between each bundle of tetrodes during production as previously described[9]. Light was delivered using a 473 nm laser diode module (Cobolt 06-MLD, Cobolt) coupled to a 100-μm multimode fibre (NA = 0.22) through a Schäfter + Kirchhoff fibre coupler (Cobolt). The laser power employed in all stimulation experiments was 3–5 mW at the tip of the fibre. For optotagging, mice were probed in an open-field arena. Following 10 min of acclimatization, bursts of 30× 5-ms-long pulses at 30 Hz were delivered with a 9-s rest period in between burst for a total of 900 stimulations over 5 min. Blue light-activated units were defined on the basis of the latency of the response to a pulse of light within a time window[57] of 5 ms. In a subset of experiments, *Vgat-cre* mice expressing ChR2 were recorded in head-restrained conditions to assess the change in tuning to moving gratings following light stimulation of collicular Vgat$^{ON}$. The same implants and setup was used in those experiments, although continuous 10-ms-long pulses at 30 Hz were delivered during visual stimulation in light-ON trials instead.

**$Ca^{2+}$ imaging.** Mice were imaged from one week after surgery with a two-photon microscope (Bergamo II, Thorlabs), equipped with a 16× 0.8 NA objective (Nikon). Mice were recorded awake and head-fixed on a custom-made floating platform. tdTomato-positive Pitx2$^{ON}$ neurons and SiR-infected cells were excited with a Ti:Sapphire laser at 1,030 nm and 920 nm, respectively, with a power of around 20 mW (Mai TaiDeepSee, Spectra Physics). Red and green emitted fluorescence were collected through a 607 ± 35 nm and 525 ± 25 nm filters, respectively (Brightline). For imaging of neuronal responses, recordings consisted of either multiple planes at different depths imaged quasi-simultaneously using a piezo device, or a single plane. The pixel resolution was kept at around 0.8 μm per pixel, while the number of pixels, field of view and imaging rate were adjusted to cover the labelled cells. On average this resulted in imaging frame rates of 30 to 60 Hz and pixel dwell times of 0.2 to 0.08 μs.

Two-photon recordings were then registered and ROIs were determined manually and extracted using CaImAn[58] (Flatiron Institute) in Python. Variation of fluorescent values over baseline ($\Delta F/F$) were computed and used for further analysis.

## Behaviour

**Training procedure.** Before recording, mice were acclimatized with being handled for two days and with carrying the head stage and being head-fixed for three more days. For all head-fixed recordings, mice were positioned standing over a wheel. During the acclimation phase on the wheel mice were able to run freely at all times but the duration of restraint was gradually increased from 5–10 min the first day to up to 30 min the third day.

**Visual stimulation.** Visual stimuli were generated using a customized version of Python PsychoPy toolbox, presented on a LCD monitor (Dell P2414H; mean luminance 35–45 cd $m^{-2}$) positioned 20 cm from the right eye of the mouse, spanning 31° down, 42° up, 45° nasal and 59° temporal. The screen was gamma-corrected and refreshed at 60 Hz. Electrophysiological recordings were aligned to the visual stimulus using a photodiode placed at a bottom right corner of the stimulus monitor and covered to not elicit a visual response on the mouse. The signal from the photodiode and the accelerometer on the wheel were time stamped and recorded using an Arduino.

Static receptive field position was estimated by 750-ms-long flashes of uniform black or white 9 cm² squares on a grey background. The screen was divided in 165 locations covering 73° by 104° (corresponding to 30 × 53 cm). To assess the receptive field the full protocol was repeated three times.

Full screen sinusoidal gratings of 12 different directions (30° steps) were used to determine direction and orientation selectivity.

Each grating (spatial frequency: 0.08 cycles per degree) would first remain static for 1 s, then move at 2.83 Hz for 2 s and stop for a further 1 s before changing direction. Gratings were displayed three times in a semi-randomized manner. This protocol was repeated three more times for a total of nine presentations per stimulus over the three trials.

For a subset of mice two extra visual stimulation paradigms were used: directional moving spot and moving Gabor patches were also used. A small black spot (1.3 cm diameter corresponding to ~3.7°) moving at 30 cm s⁻¹ in a grey background was used in order to assess the direction selectivity of neurons to a stimulus mimicking a prey moving. The stimulation paradigm consisted of a small black spot moving towards the centre of the screen (roughly aligned to the centre of the visual field) for 1 s, then staying static in the centre of the screen for 0.5 s before retracting to the opposite direction with the same speed of approach. Eight starting points were used: all four corners of the screen and all midpoints between two corners of the screen. Each starting location was presented three times per trial in a semi-randomized manner. Similar to the other visual stimulation paradigms, we recorded three trials for a total of nine presentations per movement direction.

In order to assess the spatial receptive field of neurons to moving stimuli, we used moving Gabor patches at 24 different locations (6 along the $x$ axis and 4 along $y$ axis) moving in 8 different directions (45° steps) for 1.5 s (spatial frequency: 0.08 cycles per degree and 2.83 Hz temporal frequency). All directions and locations were randomly presented 3 times for a total of 576 presentations corresponding to 3 times 8 directions at 24 locations.

Before all visual stimulation trials, 'spontaneous' firing rates were estimated over a black screen (15 s) followed by a grey screen of same average luminescence as the grating presentation (15 s). After each trial the mouse was shown first the grey screen (15 s) and then a black screen (15 s). Average activity during grey screen presentation was used as baseline and the transitions were used to compare responses to luminescence.

For in vivo whole-cell recordings, visual presentation was done using a Dell E176FP LCD screen. Full screen gratings (12 directions) were presented a total at least 3 times and up to 9. Flashing squares (covering a 100 locations) were presented 0 to 3 times, depending on the length and stability of recording.

**Open-field foraging.** Single units were recorded as mice foraged for droplets of 30% diluted condensed milk on a white Perspex arena (50 × 50 cm) to limit variability on visual input. Recording sessions consisted of 4× 5-min foraging trials, with the first and last occurring in light conditions and the second and third occurring in complete darkness. During dark trials all other sources of light within the experimental room such as computer screens were switched off or covered with red screens. For a subset of mice, open-field recordings were also performed in an open-field arena covered with either vertical or horizontal 3-cm-wide black and white stripes. Those recordings were performed in light conditions to enforce orientation-specific self-generated visual flow.

**Eye tracking.** For eye tracking in head-restrained conditions we used a camera (DMK 21BU04.H, The Imaging Source) with a zoom lens (MVL7000, ThorLabs) focused on the right eye. The eye was illuminated with an infrared LED lamp (LIU850A, ThorLabs) and an infrared filter was used on the camera (FEL0750, ThorLabs; with adapters SM2A53, SM2A6 and SM1L03, ThorLabs). When fully zoomed and placed ~20 cm from the mouse, this setup provided ~73 pixels per mm. The video was acquired using DacqUSB and synchronized to the electrophysiological recording using a small flashing infrared LED linked to the bottom edge of the camera.

In order to measure eye movements in freely moving mice we used a custom head-mounted eye and head tracking system, as previously described[40,59]. In brief, we used a commercially available camera module (1937, Adafruit; infrared filter removed). A custom 3D printed camera holder with a 21 G cannula (Coopers Needle Works) was used to hold the camera, IR LEDs (VSMB2943GX01, Vishay) and a 7.0 mm × 9.3 mm IR mirror (Calflex-X NIR-Blocking Filter, Optics Balzers). A connector (852-10-00810-001101, Preci-Dip) was used to attach the camera holder to the head plate of the mice. Mice were head-fixed, and the mirror's position was adjusted until the eye was in the centre of the eye camera. Epoxy (Araldite Rapid, Araldite) was used to fix the mirror position. A single-board computer (Raspberry Pi 3 model B, Raspberry Pi Foundation) recorded camera data at 30 Hz, capturing images of 1,296 × 972 pixels per frame for eye camera. The head roll, pitch, and yaw were estimated using an inertial motion unit including an accelerometer, gyroscope and magnetometer using previously described methods[38] and open source Arduino code (https://github.com/razor-AHRS) using an Arduino Mega 2560 rev 3.

## Quantification and statistical analysis

**Whole-cell in vitro and in vivo electrophysiology.** For optogenetic stimulation in acute brain slices, light was adjusted to elicit ~5 mV postsynaptic potentials following 5-ms-long pulses. In order to calculate the latency of light-evoked responses a linear fit was made between time points corresponding to 25–30% and 70–75% of peak amplitude of the excitatory postsynaptic potential (EPSP), or of the first slope in the case of polysynaptic EPSPs. The latency was measured as the time elapsed between light onset and the point of crossing between the linear fit on the EPSP slope and the resting membrane potential level. To determine the latency of response to visual stimuli recorded in whole-cell mode in vivo we first calculated the differential of the membrane potential during baseline (grey screen) to assess the s.d. of baseline presynaptic activity. For each repetition at the preferred direction or orientation of tuning we determined the time of the first event with a slope greater than 3 × s.d. of baseline presynaptic activity and of at least 0.5 mV ms⁻¹. Recordings with a s.d. of baseline presynaptic activity >1.5 mV were discarded.

**Spike sorting for tetrode recordings.** The electrophysiological data were spike sorted using Tint cluster cutting software (Axona). Cluster cutting was carried out by hand as clusters were generally well separated. Clusters were included in analysis if they were stable across all trials throughout the day and did not belong to clusters identified in previous recording days.

**3D head-rotation recording.** To determine the precise head rotation of the mouse during tetrode recordings we employed a sensor (50 Hz sampling frequency) equipped with accelerometers, gyroscopes, and magnetometers as previously described[38]. In brief, the sensor outputs were fed to a direction cosine matrix algorithm to provide measurements of head orientation expressed in Euler angles with respect to the Earth reference frame (yaw, pitch and roll). The rotation matrix is:

$$R_{xyz} = \begin{pmatrix} \cos\theta\cos\psi & \cos\theta\sin\psi & -\sin\theta \\ \sin\phi\sin\theta\cos\psi - \cos\phi & \sin\phi\sin\theta\sin\psi & \sin\phi\cos\theta \\ \sin\psi & + \cos\phi\cos\psi & \\ \cos\phi\sin\theta\cos\psi + \sin\phi & \cos\phi\sin\theta\sin\psi & \cos\phi\cos\theta \\ \sin\psi & -\sin\phi\cos\psi & \end{pmatrix} \quad (1)$$

From which one can extract the Euler angles as:

$$\begin{aligned} \phi &= \text{atan2}(R_{23}, R_{33}) \\ \theta &= -\arcsin(R_{13}) \\ \psi &= \text{atan2}(R_{12}, R_{11}) \end{aligned} \quad (2)$$

The primary source of the Euler angles is the gyroscope measurements, which are expressed as angular velocity

$$\omega = \begin{pmatrix} \omega_x \\ \omega_y \\ \omega_z \end{pmatrix} = \begin{pmatrix} \dfrac{\partial \phi}{\partial t} \\[6pt] \dfrac{\partial \theta}{\partial t} \\[6pt] \dfrac{\partial \psi}{\partial t} \end{pmatrix} \tag{3}$$

The main equation used to update the rotation matrix over time from gyroscope signals:

$$R^T(t+dt) = R^T(t) \begin{pmatrix} 1 & -\omega_z dt & \omega_y dt \\ \omega_z dt & 1 & -\omega_x dt \\ -\omega_y dt & \omega_x dt & 1 \end{pmatrix}$$

All drift corrections and calibrations were performed as previously described[38].

**Motion tuning.** The motion tuning of SC neurons was determined by carrying out STAs of head displacements. To compute the STA of motion, the angular head velocity for the 25 temporal bins (0.5 s) preceding and 50 bins (1 s) following the onset of spike were averaged for all spikes and for each of the three Eulerian components. The direction of the head at the onset of each spike was normalized to zero for each Eulerian component. For each spike, the calculated angular head velocities were cumulatively summated for each temporal bin to produce a head displacement for the 0.5 s preceding and 1 s following the onset of spike. The mean and s.e.m. of spike related head displacements were calculated for each temporal bin to illustrate the tuning of neurons. Displacement vectors for each Eulerian component were calculated as the difference between the minimum and maximum of the computed average displacement. The direction of the displacement vector was defined according to the temporal order of the minimum and maximum values of the computed displacement. A neuron was considered to be tuned to either light or dark if the average displacement vector for at least one of the Eulerian components was >5°, with the same direction and ranking >95% compared to a shuffled distribution for both light trials or dark trials, respectively. When angular difference was reported this referred to the magnitude in degrees of the average displacement angle a neuron was tuned to in a specific condition (for example, light condition) minus the displacement angle at another (for example, horizontal stripes).

**Visual tuning.** Analysis routines for visual tuning were developed in Igor Pro (WaveMetrics). The neuronal response to drifting sinusoidal gratings was averaged for all trials and normalized to the baseline firing rate. The selectivity was then calculated both in direction and orientation space (360° and 180°, respectively) by computing the mean orientation and direction vectors in polar coordinates, described by their modulus (corresponding to the selectivity index) and average angle.

$$SI = \left| \frac{\sum_k R(\theta_k)\, e^{2i\theta_k}}{\sum_k R(\theta_k)} \right|$$

$$\bar{\theta} = \operatorname{atan}\left( \frac{\sum_k R(\theta_k)\sin\theta_k}{\sum_k R(\theta_k)\cos\theta_k} \right)$$

for $\cos\theta_k < 0$, $\bar{\theta} = \operatorname{atan}\left( \dfrac{\sum_k R(\theta_k)\sin\theta_k}{\sum_k R(\theta_k)\cos\theta_k} \right) + \pi$

for $\sin\theta_k < 0$ and $\cos\theta_k > 0$,

$$\bar{\theta} = \operatorname{atan}\left( \frac{\sum_k R(\theta_k)\sin\theta_k}{\sum_k R(\theta_k)\cos\theta_k} \right) + 2\pi$$

Where $R(\theta_k)$ is the response at each sampled direction or orientation $\theta_k$ (12 for direction space and 6 for orientation space). Static gratings' response was averaged before and after drifting and probed only in orientation space. Given that the modulus of the vectors is dependent on the firing pattern and firing rate of the neuron, the same calculation was performed for shuffled spike times to obtain a probability distribution of shuffled DSI and SI for OS neurons (OSI) to probe for both direction selectivity and orientation selectivity (see 'Generation of shuffled datasets'). To determine whether a neuron was tuned to moving or static gratings 3 parameters were used: the selectivity index (≥0.1), the trial-to-trial angular variance (≤0.8; used as a measure of the reliability of selectivity) and the significance of the SI compared to the shuffled distribution. If two out of the three criteria were fulfilled the neuron was considered tuned. If the criteria were met for both direction and orientation spaces, a neuron was classified as DS if DSI > OSI, and OS if OSI > DSI.

Direction and orientation selectivity for small moving spots and for Gabor patches was measured as described for gratings. Two separate analyses of selectivity and preferred angle of tuning were performed for Gabor patches: a location-independent analysis by averaging the responses to each direction of movement across all locations and another analysis only considering the responses at the location of maximal average activity.

In order to determine the spatial tuning of neurons to static stimuli, we averaged the response (firing rate) to flashing 9 cm² squares per location in the screen (15 by 11 locations) for each colour separately (black or white) and divided it by the response during baseline conditions and obtained a 2D matrix corresponding to the increase or decrease of firing rate over baseline per location. This matrix was further transformed into $z$ scores. The maximal $z$ score was compared to those obtained performing the same analysis in 1000 shuffled trials (see 'Generation of shuffled datasets'). If the maximal $z$ score was >2 and ranked higher than the top 5% of those obtained from the shuffled distribution, a 2D Gaussian was the fit to the matrix. The centre of the Gaussian fit was used to determine the centre of the ssRF. The same analysis was performed to assess whether neurons had a spatial and kinetic receptive field as measured with Gabor patches. The centre of the location of maximal response was used as centre of the receptive field. The overlap of the tunings of individual neurons to different visual stimuli was computed using Intervene[60].

To investigate the modulation of the tuning to gratings by locomotion, trials for each angle presentation (4 s total) were divided based on whether the mouse was running or not. To identify running bouts we low-pass filtered the angular velocity of the wheel and used a threshold of 20 deg s⁻¹. We averaged the angular velocity of the wheel on each trial of visual stimulation and considered as 'run' trials, those in which the average running speed exceeded 20 deg s⁻¹.

**Eye tracking in head-restrained mice.** We used DeeplabCut[61] to extract the pupil position from the eye videos. Four cardinal pupil points located at the top, bottom, left and right extremities of the pupil were tracked (Extended Data Fig. 4h). The pupil position was then computed as the centre of mass of these four points. The pupil position varied across two axes: the horizontal nasal–temporal axis, and the vertical upwards–downwards axis.

The tuning of SC neurons to eye displacement was determined by carrying out STAs, following the same method as for head displacements. The angular amplitude of eye movements relative to the resting position was estimated as: $\alpha = \operatorname{atan}(d/r)$, where $d$ is the distance travelled by the pupil centre and $r$ is the radius of the eye, approximated to a sphere. A neuron was considered to be tuned for eye movement if the average displacement vector for at least one of the movement components was >1°, and ranking >95% compared to a shuffled distribution.

**Extraction of pupil position in freely moving mice.** Eye tracking was performed as previously described[40]. In brief, we tracked the position of the pupil, defined as its centre, together with the nasal and temporal eye corners. The eye corners were used to automatically align the horizontal eye axis. Thirty to fifty randomly selected frames were labelled manually for each recording day. The labelled data were used to train a deep convolutional network via transfer learning using open source code[61] (https://github.com/AlexEMG/DeepLabCut). The origin of the eye coordinate system was defined as the mid-point between the nasal and temporal eye corners. Pixel values in the 2D video plane were converted to angular eye positions using a model-based approach developed for the C57BL/6J mouse line used in this study[62]. Saccades were defined as rapid, high-velocity movements occurring in both eyes with a magnitude exceeding 350 deg s$^{-1}$.

**Generation of shuffled datasets.** For each cell, the spike-onset times were temporally shifted by 2–180 s in a wrap-around manner. This works to shift the relationship between the spike times and the recorded heading directions of the mice or the visual stimuli while maintaining the temporal relationship between spiking events. Once these data were shifted, analyses were carried out to determine the mean displacement vector or selectivity index of the temporally shifted data. This process was repeated 1,000 times so as to produce a random distribution.

**Visuo-motor alignment.** This analysis was performed on neurons that had a significant tuning to moving gratings and a significant tuning to head rotations. In order to determine whether any alignment existed between these 2 types of tuning we first modelled the 3D head-rotation tuning of a neuron as a 2D projection of a vector coming out of the eye of that mouse (corresponding to gaze) into the 2D plane of the screen in which the visual stimuli were shown. We considered the head position at time 0 equal to the position of the mouse head when head-restrained for visual stimuli (see '3D head rotations to 2D screen plane transformation model'). We then determined the weighted average of the newly computed motion vectors that make up the 2D trajectory. We weighted the motion vector at each time-point by the instantaneous velocity of the gaze movement. This analysis gave us a vector with an angle $\bar{\theta}_{gaze}$ on the same plane as the gratings presentation and that could be directly compared to the angle of maximum selectivity $\bar{\theta}_{gratings}$. We focussed on comparing the direction of these two vectors by subtraction: $\bar{\theta}_{gaze} - \bar{\theta}_{gratings}$.

## Modelling

**3D head rotations to 2D screen plane transformation model.** In order to project a gaze vector from the eye of the mouse onto the plane of the screen in which the visual stimuli are shown we first computed the equation of the plane in the laboratory's reference frame. $l$ is a line from point 1 to point 2 in the screen plane and since it also lies within the $xy$ plane (for the particular choice of points), we have (see Supplementary Fig. 1):

$$l \perp \hat{z}$$
$$l \cdot \hat{z} = (-x_0, y_0, 0) \cdot (0, 0, 1) = 0$$

Thus, the normal vector to the plane is given by:

$$\boldsymbol{n'} = \hat{z} \times \boldsymbol{l} = \begin{vmatrix} \hat{i} & \hat{j} & \hat{k} \\ 0 & 0 & 1 \\ -x_0 & y_0 & 0 \end{vmatrix} = (-y_0\hat{i}, -x_0\hat{j}, 0\hat{k}) \quad (4)$$

For simplicity we pick the plane normal vector as:

$$\boldsymbol{n} = -\boldsymbol{n'} = (y_0\hat{i}, x_0\hat{j}, 0\hat{k}) \quad (5)$$

Using point 1 coordinates, we can find the equation of the plane:

$$\mathbf{n} \cdot (x - P_1^x, y - P_1^y, z - P_1^z) = 0$$
$$(y_0, x_0, 0) \cdot (x - x_0, y, z) = 0 \quad (6)$$
$$y_0(x - x_0) + x_0 y = 0$$

Next, we determined the equation of a vector $\mathbf{r}_e^H$ coming out of the mouse eye in the mouse head's reference frame:

$$\mathbf{r}_e^H = \begin{pmatrix} x' \\ y' \\ z' \end{pmatrix} = \begin{pmatrix} P_0^{x'} \\ P_0^{y'} \\ P_0^{z'} \end{pmatrix} + m' \begin{pmatrix} a \\ b \\ c \end{pmatrix} \quad (7)$$

Where $a$, $b$ and $c$ are the elements of a vector that determines the direction of the gaze and $P_0'$ is the initial location of the pupil in the head frame (see Supplementary Fig. 2).

Next, we need to find the coordinates of $r_e^H$ in the inertial reference frame once the head rotation has occurred:

$$r_e^I = Rr_e^H = \begin{pmatrix} P_0^x \\ P_0^y \\ P_0^z \end{pmatrix} + m \begin{pmatrix} a \\ b \\ c \end{pmatrix} = \begin{pmatrix} x \\ y \\ z \end{pmatrix} \quad (8)$$

With $x$, $y$ and $z$ corresponding to the components of the eye vector in the inertial frame.

$$x = P_0^x + ma$$
$$y = P_0^y + mb$$
$$z = P_0^z + mc$$

where $m(t) = R(t)m'$ and $P_0(t) = R(t)P_0'$.

Now, we need to find the intersection of $r_e^I$ with the plane defined in Equation (6) at times $t$ and $t + \mathrm{d}t$:

$$r_e^I(t) = P_0(t) + m(t) \begin{pmatrix} a \\ b \\ c \end{pmatrix} \quad (9)$$

$$r_e^I(t + \mathrm{d}t) = P_0(t + \mathrm{d}t) + m(t + \mathrm{d}t) \begin{pmatrix} a \\ b \\ c \end{pmatrix} \quad (10)$$

$$r_e^I = \begin{pmatrix} \cos\theta\cos\psi & \cos\theta\sin\psi & -\sin\theta \\ \sin\phi\sin\theta\cos\psi - \cos\phi & \sin\phi\sin\theta\sin\psi & \sin\phi\cos\theta \\ \sin\psi & +\cos\phi\cos\psi & \\ \cos\phi\sin\theta\cos\psi + \sin\phi & \cos\phi\sin\theta\sin\psi & \cos\phi\cos\theta \\ \sin\psi & -\sin\phi\cos\psi & \end{pmatrix}$$
$$\begin{pmatrix} P_0^{x'} \\ P_0^{y'} \\ P_0^{z'} \end{pmatrix} + m' \begin{pmatrix} a \\ b \\ c \end{pmatrix} \quad (11)$$

$$r_{e,x}^I = P_0^{x'}(\cos\theta\cos\psi) + P_0^{y'}(\cos\theta\cos\psi) - P_0^{z'}(\sin\theta)$$
$$+ m'[a\cos\theta\cos\psi + b\cos\theta\sin\psi - c\sin\theta] \quad (12.1)$$

$$r_{e,y}^I = P_0^{x'}(\sin\phi\sin\theta\cos\psi - \cos\phi\sin\psi)$$
$$+ P_0^{y'}(\sin\phi\sin\theta\sin\psi + \cos\phi\cos\psi) + P_0^{z'}(\sin\phi\cos\theta)$$
$$+ m'[a(\sin\phi\sin\theta\cos\psi - \cos\phi\sin\psi)$$
$$+ b(\sin\phi\sin\theta\sin\psi + \cos\phi\cos\psi) + c\sin\phi\cos\theta] \quad (12.2)$$

$$
\begin{aligned}
r^l_{e,z} = {} & P^{x'}_0(\cos\phi\sin\theta\cos\psi + \sin\phi\sin\psi) \\
& + P^{y'}_0(\cos\phi\sin\theta\sin\psi - \sin\phi\cos\psi) \\
& + P^{z'}_0(\cos\phi\cos\theta) + m'[a(\cos\phi\sin\theta\cos\psi + \sin\phi\sin\psi) \\
& + b(\cos\phi\sin\theta\sin\psi - \sin\phi\cos\psi) + c\cos\phi\cos\theta]
\end{aligned}
\tag{12.3}
$$

To find the point of intersection, we can substitute the $r^l_e$ component in the equation of the plane and compute the coefficient $m$. Starting with Equation (6):

$$
\begin{aligned}
y_0(x - x_0) + x_0 y &= 0 \\
y_0(r^l_{e,x} - x_0) + x_0 r^l_{e,y} &= 0
\end{aligned}
\tag{13}
$$

In order to have a better perspective of the track that the intersection of $r^l_e$ with the screen at different times produces, we treat each of the intersection points as the endpoint of a vector with its base at the inertial frame origin. Then we can rotate these vectors around the $z$ axis of the inertial frame with a desired angle to have a better view. In other words, it is as if we have rotated the screen with that angle, since the relative geometry of the points on the screen would not change after this rotation. In order to perform this rotation, we use a Rodrigues' rotation formula[63]:

$$
\mathbf{v}_{rot} = \mathbf{v}\cos\beta + (\hat{z} \times \mathbf{v}) + \hat{z}\,(\hat{z} \cdot \mathbf{v})(1 - \cos\beta)
\tag{14}
$$

Where $\mathbf{v}$ is the vector of intersection points and $\beta$ is the rotation angle around the $\hat{z}$ axis. If we pick $\beta = \sin^{-1}\left(\frac{y_0}{\sqrt{x_0^2 + y_0^2}}\right)$, we practically rotate the screen such that it becomes parallel to $\hat{x}$ and $\hat{z}$ axes of the inertial frame and perpendicular to $\hat{y}$.

In the above treatment, the gaze vector $\mathbf{r}^H_e$ is fixed in the head frame. However, this is not correct as the pupil would move in the head frame. In order to correct for this effect, we simultaneously recorded eye and head rotations in mice during foraging, as described[40], and implemented the pupil rotations in the head frame to our model. This effectively makes the direction of the gaze dependent on pitch, roll and yaw. In mathematical terms, in Equation (7) we would make the correction:

$$
d = \begin{pmatrix} a \\ b \\ c \end{pmatrix} \rightarrow d\,(\phi, \theta, \psi) = R^H_{\text{correction}}(\phi, \theta, \psi)\begin{pmatrix} a \\ b \\ c \end{pmatrix}
\tag{15}
$$

The rest of the transformations follows as before. The correction rotation matrix in head frame, $R^H_{\text{correction}}(\phi, \theta, \psi)$, was computed from the head–eye rotations measured and represented in Extended Data Fig. 9f,g.

**Neural network model.** The neural network model consisted of 3 layers, each with 500 neurons. The first layer of neurons had receptive fields, $\mathbf{z}_i = [z_{i,\text{NT}}; z_{i,\text{VD}}] \in \mathbb{R}^2$, spanning a (signed) range of $z_{\text{NT}} \in [0, 140]$ degrees along the naso-temporal (NT) axis and $z_{\text{VD}} \in [0, 70]$ degrees along the ventro-dorsal (VD) axis. The angle in the NT–VD plane corresponding to the centre of the receptive field of neuron $i$ was denoted $\theta^{\text{RF}}_i := \tan^{-1}(z_{i,\text{VD}}/z_{i,\text{NT}})$. Each neuron in layer 1 connected to a corresponding DS neuron in layer 2, which was selective to movement in direction $\theta^{\text{DS}}_i = \theta^{\text{RF}}_i + \pi$. That is, if a RF neuron responded to stimuli in a particular part of the receptive field, the corresponding DS neuron responded to motion from this location towards the agent. Finally, layer 3 consisted of motor neurons that were 'anti-aligned' with the DS neurons, $\theta^{\text{M}}_i = \theta^{\text{DS}}_i + \pi$. Each motor neuron induced movement in the direction $\boldsymbol{m}_i = [\cos\theta^{\text{M}}_i; \sin\theta^{\text{M}}_i]$.

All simulations were run using Euler integration with a discrete timestep of size $\Delta t = 0.5$ ms and a neural time constant of $\tau_{\text{neural}} = 10$ ms. The firing rates of all neurons evolved according to $r_{t+1} = \left[r_t + \frac{\Delta t}{\tau_{\text{neural}}}(-r_t + x + \epsilon)\right]_+$, where $[\cdot]_+$ indicates a rectified linear unit nonlinearity and $\epsilon \sim N(0, \sigma^2)$ is Gaussian input noise with s.d. of $\sigma = 0.1$. $x$ indicates the input to each neuron, which is described for each layer in the following.

The receptive field neurons responded to a stimulus at location $\mathbf{s} = [s_{\text{NT}}; s_{\text{VD}}]$ in egocentric coordinates with Gaussian tuning curves of the form $x^{\text{RF}}_i = 1.5\exp(-0.6\kappa\,|\mathbf{s} - \mathbf{z}_i|_2^2)$, where $\kappa = 40$.

In the static setting, the DS neurons received input from the RF neurons such that $x^{\text{DS}}_i = r^{\text{RF}}_i$. In the kinetic setting, the input was given by $x^{\text{DS}}_i = \gamma \exp(\kappa\,[\cos(\theta^S - \theta^{\text{DS}}_i + \pi) - 1])$, where $\theta^S = \tan^{-1}(s_{\text{DV}}/s_{\text{NT}})$ is the angle of the stimulus within the visual field in egocentric coordinates. Here, $\gamma = \left[-\frac{\mathbf{s}^{\mathsf{T}}\dot{\mathbf{s}}}{|\mathbf{s}||\dot{\mathbf{s}}|}\right]_+$ is a scale factor that adjusts the input strength according to the 'concentricity' of the stimulus, such that the responses of all DS neurons are stronger when the motion of the stimulus ($\dot{\mathbf{s}}$) is 'concentric' to the stimulus location ($\mathbf{s}$) in the visual field.

Finally, motor neurons received input from the DS neurons, $x^{\text{M}}_i = r^{\text{DS}}_i$, and the motion of the agent was computed as $\Delta\mathbf{a}_t = \frac{\Delta t}{\tau_m}\sum_i r^{\text{M}}_i \mathbf{m}_i$. Simulations were terminated once (1) 2.5 s had elapsed, (2) the stimulus left the receptive field of the agent, or (3) the total agent motion exceeded 200° in the NT direction or 100° in the VD direction. $\tau_m$ took a default value of 250 ms and was adjusted to modulate the relative velocity of the agent compared to the stimulus.

For quantitative analyses, energy consumption was computed as proportional to the total cumulative movement speed, $E \propto \sum_t |\Delta\mathbf{a}_t|_2$. An 'intercept' was considered successful if the agent moved within 24° of the stimulus (20% of the receptive field).

### Statistical methods

Data were tested for normality. If the distribution was not normal, non-parametric tests such as Mann–Whitney and Kruskal–Wallis tests were used instead of $t$-tests and ANOVA. When the number of recordings per experimental condition was too low to assess the type of distribution, the data were assumed normal.

In all STA analyses, a $t$-test was applied between the displacement vector and the displacement vectors of the shuffled data with a threshold of 0.05. Neurons were considered to be motion tuned if the $t$-test determined a significant difference between the displacement vectors of the real and shuffled data. Similarly, to determine SI significant, a $t$-test was applied between the SI and the SIs of the shuffled distribution with a threshold of 0.05. To test the angular alignment of visuo-motor neurons Watson–Williams tests were performed between the dataset and a random distribution of equal s.d. centred at 0°, 90°, 180° and 270°, an $F$-statistic value lower than the critical value (3.85) indicates significant similarity between distributions.

All results are presented as mean ± s.e.m. unless otherwise stated. Results were considered statistically significant at *$P \le 0.05$, **$P \le 0.01$, ***$P \le 0.001$. Relevant $P$ values and tests used are reported in the figure legends.

### Reporting summary

Further information on research design is available in the Nature Portfolio Reporting Summary linked to this article.

### Data availability

The datasets generated during and/or analysed during the current study are available from the corresponding author on reasonable request. When possible, pre-processed electrophysiological data have been uploaded to Zenodo (https://doi.org/10.5281/zenodo.11105001)[64]. Source data are provided with this paper.

### Code availability

The code corresponding to the computational neuronal network model developed for Fig. 4 is available at https://github.com/AnaG-R/Visuomotor-model-SC.git.

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

**Acknowledgements** The authors thank the Laboratory of Molecular Biology electronics and mechanical workshops for the help with hardware development, members of the Biological Service Group for their support with animal husbandry and F. Morgese for assistance in viral production. This study was supported by the Medical Research Council core funds to M.T. (MC_UP_1201/2), the UKRI (ERC Consolidator Replacement Grant, EP/X034666/1) to M.T., the European Research Council with an ERC Starting Grant to M.T. (STG 677029), the ERANET-NEURON (Micronet, MC_PC_16036) grant to M.T. and the Wellcome Trust (J.P., 211258/Z/18/Z). A.G.-R. held a Henslow Fellowship, awarded by the Cambridge Philosophical Society and St Edmund's College (University of Cambridge). K.J. was funded by a Cambridge Gates Scholarship. M.N. was funded by a Trinity Henry Barlow scholarship, by M. C. Payne at the TCM Group and, for the last phase of revisions, by Nokia Bell Labs for research unrelated to this manuscript. D.d.M. was supported by the European Union's Horizon 2020 research and innovation programme under the Marie Sklodowska-Curie grant agreement no. 894697. J.K. was supported by a PhD studentship funded by Wolfson College, the Department of Physiology, Development and Neuroscience and the School of Biological Sciences DTP at the University of Cambridge.

**Author contributions** A.G.-R. and M.T. conceived the project. A.G.-R. performed the experiments with contributions from D.d.M., J.K. and E.W. K.J. built the neuronal network model with input from G.H. J.W. and A.G.-R. set up tetrode recordings. A.G.-R. analysed all data except for the STA of eye movements under head restraint and calcium imaging, which was performed by D.d.M., and the coupling of eye and head movement in freely moving mice, which was performed by J.K. and J.P. E.C. produced viral constructs. M.N. developed the geometrical model converting 3D head rotations into 2D gaze projections and performed associated analysis. A.G.-R. and M.T. wrote the manuscript and implemented input from all authors.

**Competing interests** The authors declare no competing interests.

**Additional information**
**Correspondence and requests for materials** should be addressed to Ana González-Rueda or Marco Tripodi.

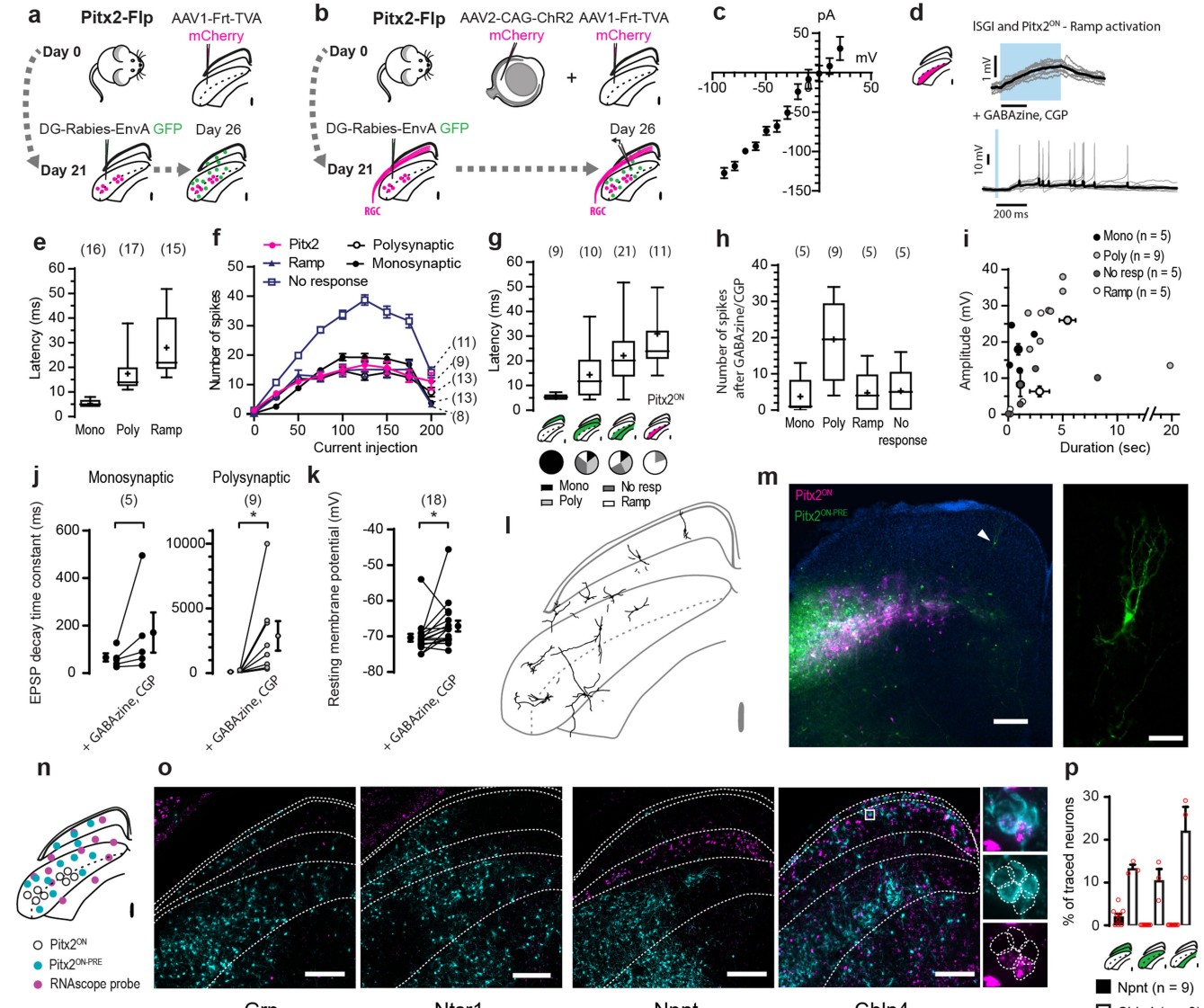

**Extended Data Fig. 1 | Characteristics of Pitx2$^{ON-PRE}$ and Pitx2$^{ON}$ neurons.**
**a**, Full diagram of the experimental design in Fig. 1a and in Fig. 1d (**b**). **c**, Reversal potential of direct retinal input onto monosynaptically connected Pitx2$^{ON-PRE}$ neurons (n = 8 neurons). **d**, Example neuron displaying a slow raising depolarisation (ramp) following long retinal stimulation. Due to the slow raising time, these neurons were considered as non-responsive in Fig. 1. **e**, Time between light pulse start and postsynaptic response initiation (latency). **f**, Number of action potentials elicited following different current injections. **g**, Latency of responses per layer also considering Ramp activation. **h**, Number of spikes elicited by light activation after GABAzine and CGP superfusion. **i**, Amplitude and duration of activation in response to light stimulation after GABAR blockade. **j**, Decay time constant of excitatory postsynaptic activity before and after GABAR block for neurons receiving direct (left, one-tailed paired t-test

$p$ = 0.0979) and indirect (right, one-tailed paired t-test $p$ = 0.0214) retinal input. **k**, Resting membrane potential before and after GABAR block (one-tailed paired t-test $p$ = 0.0260). **l**, Reconstruction obtained from 10 of the neurons included in Fig. 1. **m**, Example Pitx2$^{ON-PRE}$ narrow-field neurons traced with G-deleted Rabies virus, image extracted from dataset used for Fig. 1 (N = 4 mice, bars: 200 μm and 50 μm). **n**, In situ hybridization was performed on SC slices containing Pitx2$^{ON-PRE}$ neurons traced with G-deleted rabies using RNAscope. **o**, Example straining of markers of narrow-field (Grp and Cbln4) and wide-field (Ntsr1 and Npnt, bars: 250 μm). **p**, quantification of Npnt$^+$ and Cbln4$^+$ Pitx2$^{ON-PRE}$ neurons. Individual replicates are shown in red. All box plots indicate: minimum, 1$^{st}$ quartile, median, 3$^{rd}$ quartile and maximum; cross indicates mean. All other error bars represent ± s.e.m.

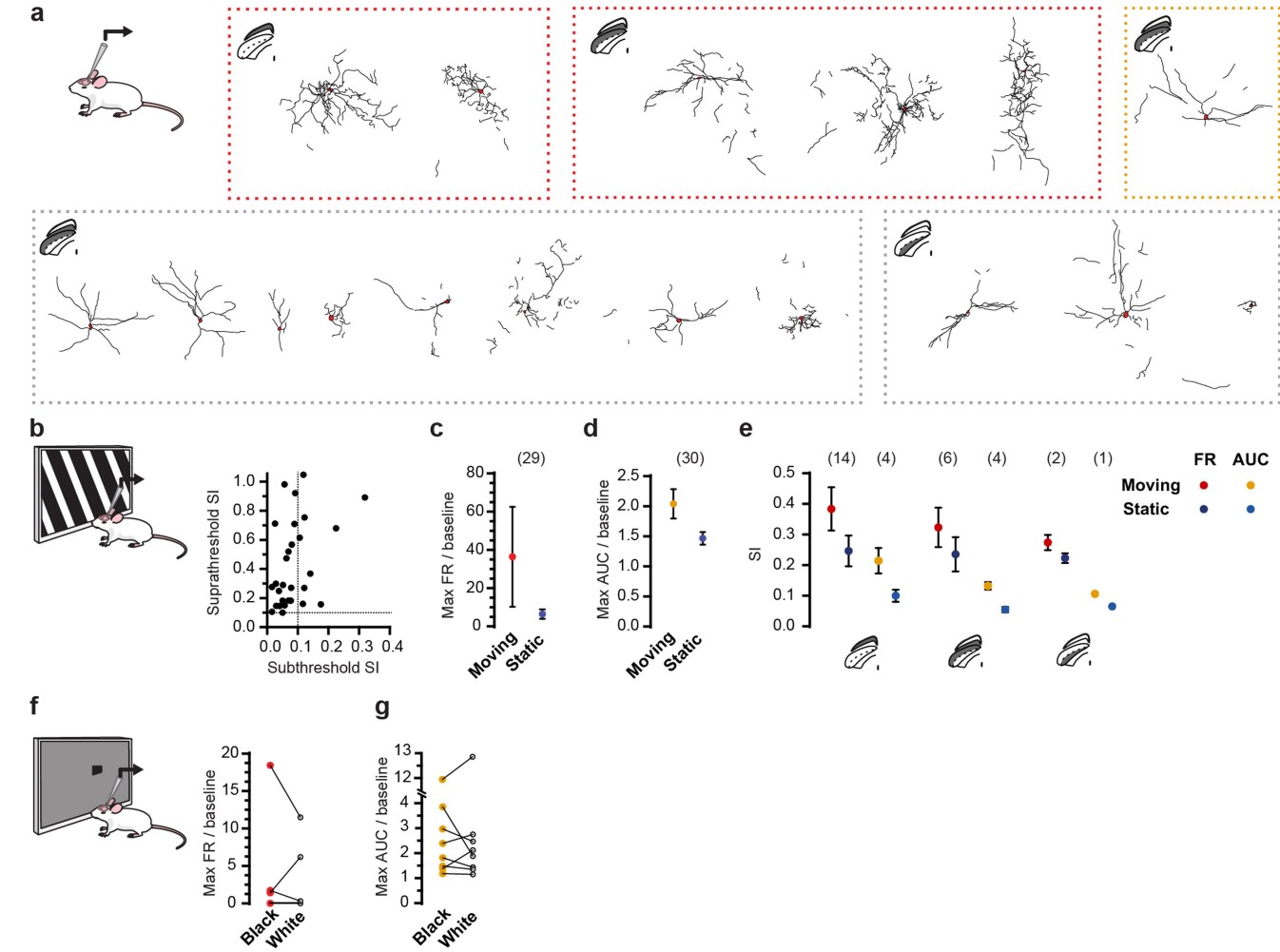

**Extended Data Fig. 2 | Reconstruction and visual properties of neurons recorded in vivo. a**, Reconstruction of biocytin-filled neurons in vivo in whole-cell mode in the Superior Colliculus. Red dashed box indicates recorded neurons tuned to moving gratings with subthreshold and suprathreshold tuning. Yellow indicates neurons tuned to moving gratings only in the subthreshold domain and Grey are neurons with no significant visual tuning. The layer in which they were recorded is shown on the top corners of each box. **b**, Comparison of subthreshold and suprathreshold selectivity indexes (SI). **c**, Increase in firing rate over baseline during the presentation of the direction eliciting maximal response for neurons tuned to moving gratings. The maximal spiking response is shown both when those gratings moved (red) and when they were static (blue). **d**, Integral (AUC) of the same responses shown in **c** and of those cells tuned only in the subthreshold domain. **e**, Distribution per layers of the SI presented in Fig. 2g. **f**, Increase in firing rate at most responsive location for neurons with a spatial receptive field measured using flashing squares. Increase in spiking activity is shown for the same neurons in response to black squares in grey background (red marker) and white squares (white marker) in grey background. **g**, AUC of the maximal response to black (yellow marker) or white squares (white marker). Error bars represent ± s.e.m.

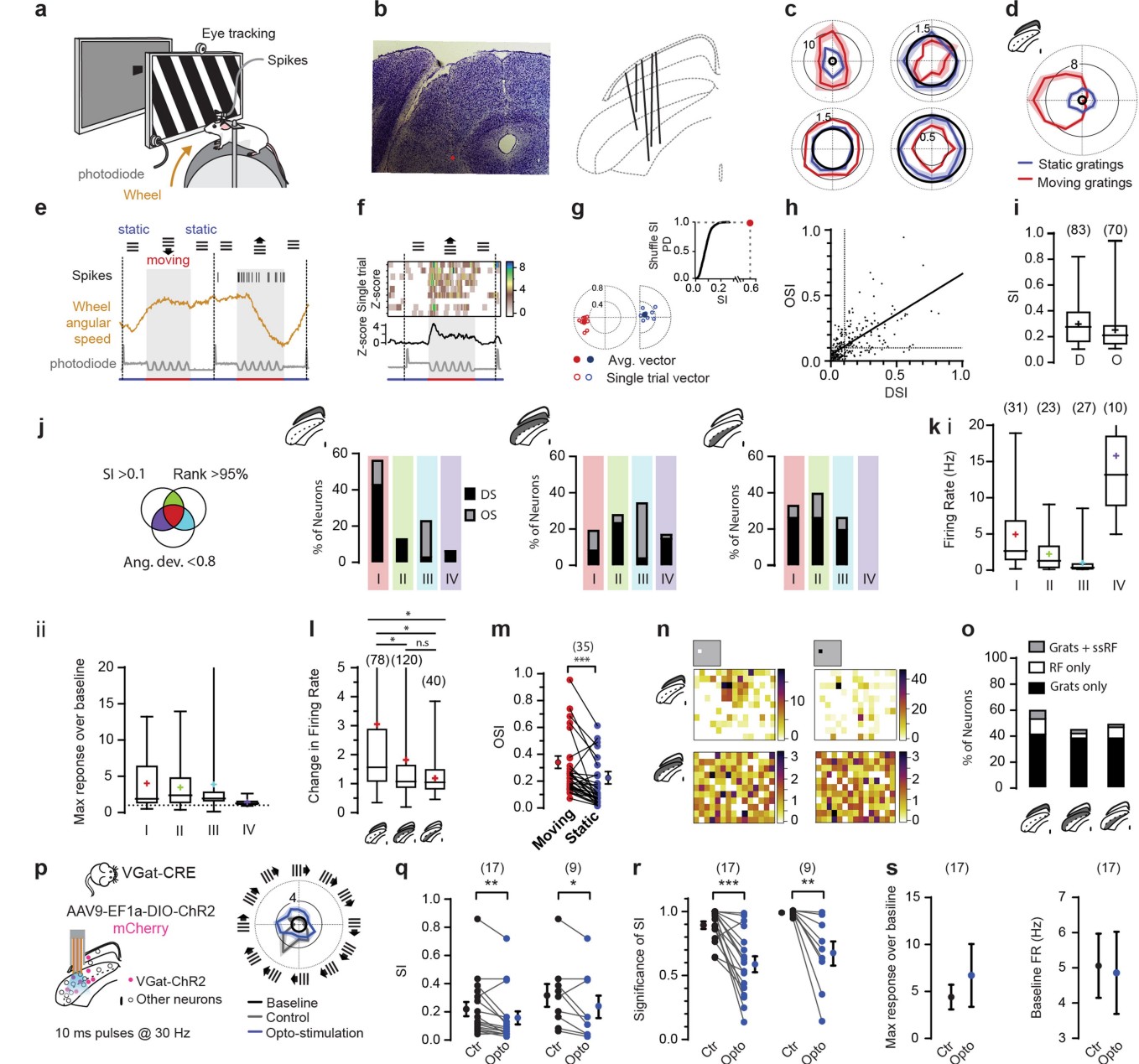

**Extended Data Fig. 3** | See next page for caption.

**Extended Data Fig. 3 | Visual tuning properties in the SC in awake mice.**
**a**, Illustration of the experimental setup to record neuronal responses to visual stimuli in the Superior Colliculus in awake conditions. **b**, Example Nissl staining of electrode tract in the SC (left) and reconstructed tracts for mice included in Fig. 2 (N = 5, right). **c**, Example tuning curves of two OS neurons (top), one with positive selectivity over baseline and another below. At the bottom two neurons tuned to moving gratings but showing no specific selectivity are shown: one increased its firing when gratings move and the other decreased it. These last two behaviours were not considered tuning to moving gratings for further analysis. **d**, Example tuning curve of a direction selective neuron (DS) in the superficial layers of the SC. **e**, Example responses of the neurons in **d** to two grating presentations. **f**, Z-score of the response of neuron in **d** to its preferred direction for all 9 trials this same direction was presented. The average Z-score is shown underneath. **g**, Single trial (open circles) and average Selectivity Indexes when gratings were moving (red) or static (blue) and the distribution of the average SI of all 1000 shuffled replicates (top). The red dot indicates how the average SI of this neuron ranks among the shuffled distribution (black). **h**, Comparison of the orientation SI (OSI) and the direction SI (DSI) for the same neurons. Linear regression y = 0.6164x + 0.04868. 95% CI of slope: 0.5322-0.7006. **i**, SIs for moving gratings >0.1 in direction space (D) or orientation space (O). **j**, Three properties were used to determine whether a neurons was tuned to moving gratings: the SI (>0.1), its significance when compared to the shuffled distribution (rank >95%, corresponding to > 0.95 in the probability distribution), and the trial to trial variability shown as the angle mean deviation of the preferred angles per trial (<0.8 rad). Neurons fulfilling all three criteria were considered strongly tuned and if they only fulfilled 2, they would still be considered tuned albeit less strongly. We did not considered a neuron tuned if only one criteria was fulfilled. **k**, Firing rate during baseline (i) and increase in response to preferred direction (ii) for neurons of all 4 classes described in **j**. **l**, Change in firing rate during gratings presentation (Kruskal-Wallis $p$ < 0.0001, Dunn's multiple comparison test $p$ = 0.0005) for neurons tuned to moving gratings across layers. **m**, SIs for Orientation selective neurons. Paired OSIs for each cell (moving gratings, red, and static gratings, blue) are shown. Mean ± s.e.m. is also displayed. Two-tailed paired t-test $p$ = 9.10 × 10$^{-5}$. **n**, Example receptive field responses for a neuron in the superficial layers of the SC (top) and deep layers (bottom). **o**, Proportion of all neurons tuned to moving gratings (Grats, black), flashing squares (RF, white) or both (grey) across layers. **p**, Vgat-cre mice, injected with an AAV-DIO-ChR2 and implanted with optrodes were recorded before and after light stimulation (n = 17 neurons in N = 3 mice; constant 10 ms pulses at 30 Hz). An example neuron losing its directional selectivity to moving gratings (red) when VGat$^+$ neurons were activated (blue) is shown. **q**, Change in SI after light stimulation (opto) for all neurons (left, one-tailed paired t-test $p$ = 0.0051) and only neurons tuned in control conditions (right, one-tailed paired t-test $p$ = 0.0308). **r**, Change in the rank of the SI compared to a shuffled distribution (significance considered for >= 0.95) before and after light stimulation for all recorded neurons (left, one-tailed paired t-test $p$ = 5.52 × 10$^{-5}$) and those tuned in control conditions (right, one-tailed paired t-test $p$ = 0.0036). **s**, Maximal response over baseline (left, one-tailed paired t-test $p$ = 0.2076) and baseline firing rate (right, one-tailed paired t-test $p$ = 0.40124), in control conditions (red) and during optogenetic stimulation (blue). All box plots indicate: minimum, 1$^{st}$ quartile, median, 3$^{rd}$ quartile and maximum; cross indicates mean. All other error bars represent ± s.e.m.

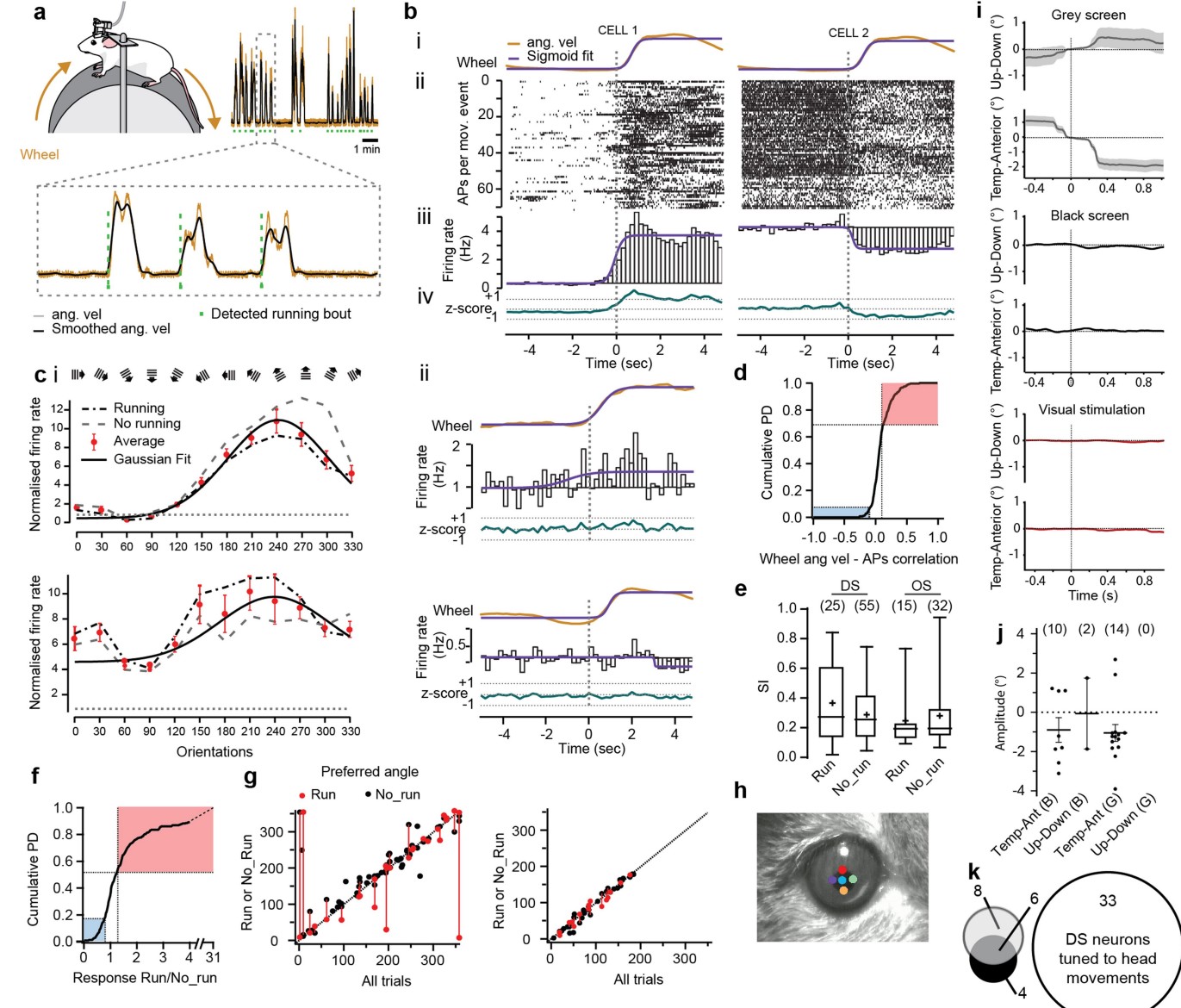

**Extended Data Fig. 4 | Modulation of visual responses in the SC. a**, Diagram of a head restrained mouse running on the wheel. The angular velocity of the wheel was extracted to detect running bouts (green). **b**, Two example neurons strongly modulated by running and recorded simultaneously. The response of both neurons to 80 running bouts is shown as a raster (ii). The average of the angular velocity of the wheel for all bouts (i), the average change in firing frequency (iii) and its corresponding z-score (iv) are also shown. **c**, Two neurons (top and bottom graphs respectively) tuned to moving gratings (9 trials, i) with their tuning either positively modulated (bottom) or negatively modulated (top) by running. Neither of these had a strong modulation to running in baseline conditions (ii). **d**, Cumulative distribution of the correlation between the wheel's angular velocity and the firing rate of the neurons (red and blue shading indicates the proportion of neurons with a positive >0.1 or negative <0.1 correlation respectively. **e**, SIs for DS or OS neurons in trials when the mouse was running on the wheel (Run) or not (No_run). **f**, Cumulative distribution of the change in response of neurons tuned to moving gratings in trials when the mouse was running vs trials when it was not (red and blue

shading indicates the proportion of neurons with an increase >20% or decreased <20% response when running). **g**, Comparison of the preferred grating angle of DS neurons (left) or OS neurons (right) for all trials and those trials when the mouse was running (red) or not (black). **h**, Frame from a video of the right eye of a mouse during visual stimulation and the markers used for deeplabcut tracking of the pupil position throughout the recording. **i**, Example STA of the right eye movements of a neuron tuned during baseline conditions (grey screen, top), but not in darkness (middle) or during visual presentation (bottom). **j**, Eye angular displacement in the temporal-anterior axis (positive = anterior movements) and the dorso-ventral axis (positive = downward movements) for neurons tuned to eye movements in darkness (black screen, B) or light (grey screen, G). **k**, Venn diagram showing the overlap of the neurons tuned to saccadic eye movements in baseline (grey screen), darkness and visuo-motor neurons as described in Fig. 3. All box plots indicate: minimum, 1st quartile, median, 3rd quartile and maximum; cross indicates mean. All other error bars represent ± s.e.m.

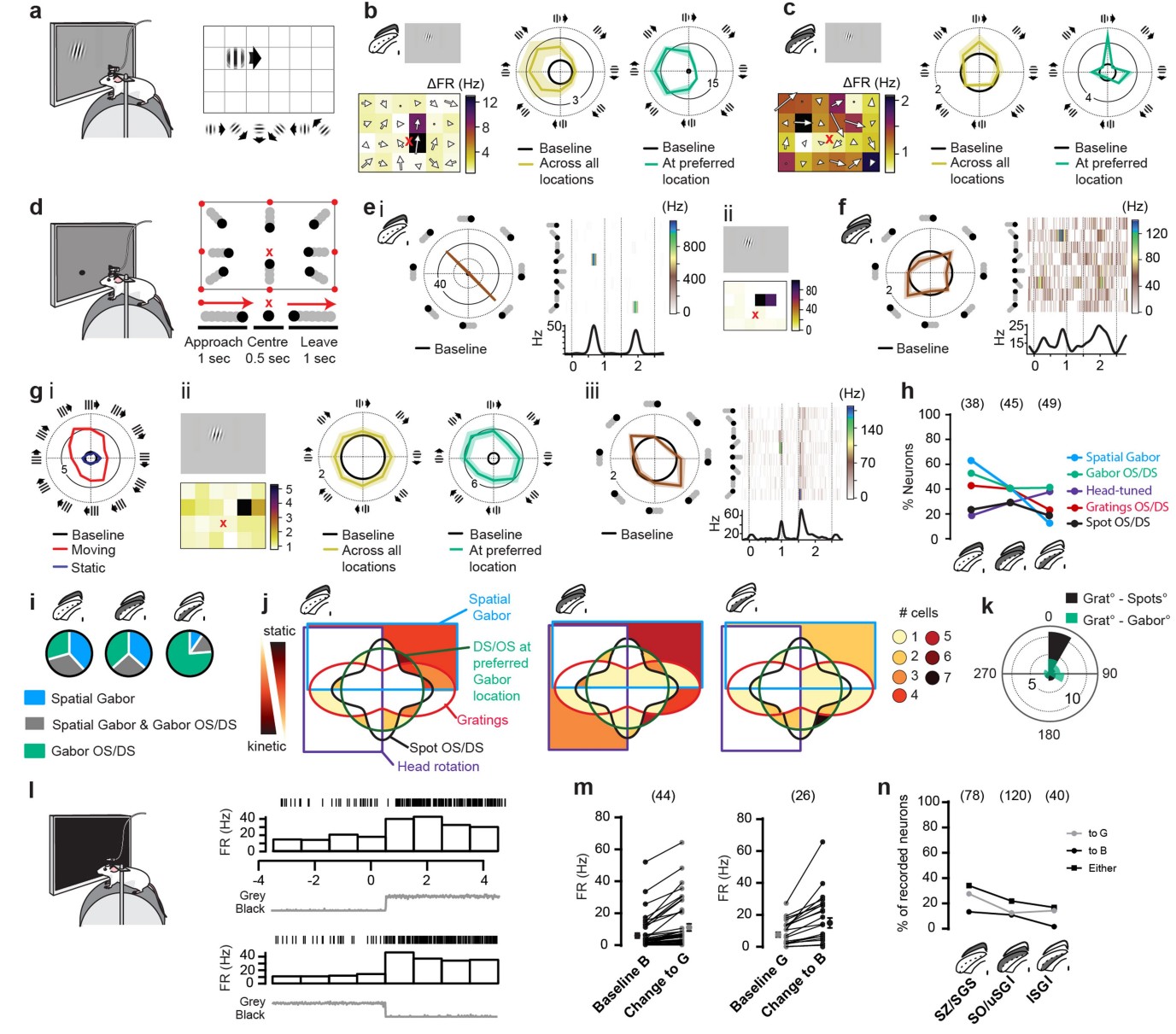

**Extended Data Fig. 5** | See next page for caption.

**Extended Data Fig. 5 | Mapping of visual tuning properties in response to drifting gabor patches, small moving spots and changes in luminescence in the SC. a**, Gabor patches drifting in 8 different directions were randomly presented at 24 locations of the screen. **b**, Example neuron displaying location-specific tuning to moving Gabor patches measured by averaging all directions of movement per location (left, overlaid arrows length and direction indicate the selectivity index and preferred direction of movement per location), location-independent directional tuning as measured by averaging the response to drifting patches for each direction across all locations (middle, yellow), and a strong direction tuning at the preferred location (right, green). Shaded area in polar plots indicates ± s.e.m. **c**, Example neuron without a location preference but tuned to particular direction of movement of the gabor patches. **d**, Mice were also presented with a small spot (1.3 cm diameter) starting at 8 locations along the corners and border of the screen (red dots) and moving in 8 directions towards and away from the mouse's centre of gaze on the screen (red cross). **e**, Example tuning of a neuron responding to a small spot moving in a particular orientation. Shaded area in polar plots indicates ± s.e.m. Average firing rate for each direction is shown as a heat map and the average change in firing rate over time (2.5 sec) across directions is also shown at the bottom (i). The tuning of this neuron corresponds to the spot crossing its spatial receptive field mapped using drifting gabor patches (ii). **f**, Example tuning to a small moving spot for a neuron without spatial receptive field. **g**, Example neuron tuned to full field moving gratings (i), gabor patches (ii) and moving spot (iii). **h**, Distribution of neurons responding to particular locations in space mapped using gabor patches (spatial gabor, blue), neurons showing direction or orientation selectivity mapped using gabor patches (gabor OS/DS, green) or full field gratings (red), neurons tuned to moving spots (spot OS/DS, black) or tuned to head rotations (head tuned, purple). The number of recorded neurons per depth of recording, indicated by the shading in the SC diagram, is shown in parenthesis (N = 4 mice). **i**, Distribution of neurons tuned to particular locations (spatial gabor, blue), direction or orientation selective neurons (Gabor OS/DS, green) or both (grey) among all neurons tuned to drifting gabor patches. **j**, Overlap of the tunings shown in h for individual neurons, computed using Intervene[60]. Similarly to a Venn diagram, this plot illustrates the logical relation between sets, with darker areas indicating a larger number of neurons at the intersection of sets. Static features are preferentially represented at the top, while kinetic ones are preferentially represented towards the bottom. Note that the number of neurons is represented by the shade of colour and not by the surface of the intersection. **k**, Angular difference between the preferred angle of tuning measured using full field gratings and moving spots (black) or gabor patches (green). **l**, Example neurons recorded with tetrodes in awake head-restrained conditions and responding to changes in luminescence (increased luminescence from black to grey, top, and decrease in luminescence, bottom). For both neurons a raster of one example trial and the average PSTH are shown. **m**, Firing rate changes from black to grey or grey to black. Number of tuned neurons is shown in parenthesis. **n**, Percentage of neurons displaying a response to luminescence across layers. Total number of neurons recorded is shown in parenthesis.

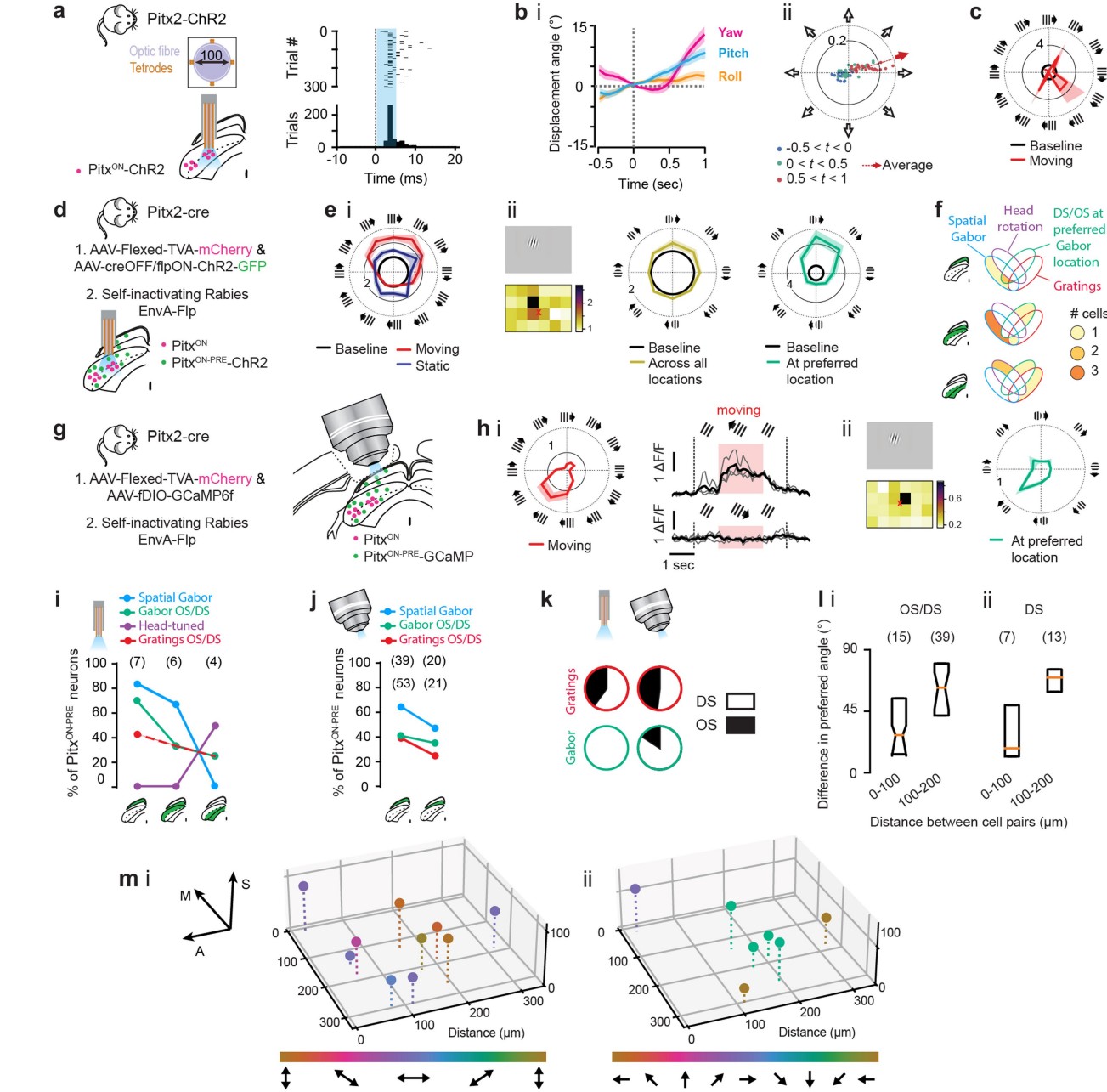

**Extended Data Fig. 6** | See next page for caption.

**Extended Data Fig. 6 | Visual and motor tuning of motor Pitx2$^{ON}$ and premotor Pitx2$^{ON-PRE}$ neurons in the SC. a**, Example opto-tagged Pitx2$^{ON}$ motor neurons. Peri-stimulus time histogram of 900 5 ms-long laser stimulations (right, bottom) and example raster plot of the 1$^{st}$ 300 stimulations are shown (right, top). **b**, Example STA of angular head displacements in yaw, pitch and roll for a Pitx2$^{ON}$ motor neuron (i). The 3D STA was projected onto the 2D plane of the visual stimulation screen to draw the tuned direction of gaze. The decomposed and average gaze direction is shown in ii. **c**, Tuning to moving gratings of the neuron in b. **d**, Pitx2$^{ON-PRE}$ neurons were opto-tagged using a combinatorial Cre$^{OFF}$/Flp$^{ON}$ virus expressing ChR2 and a Self-Inactivating Rabies virus expressing FLP recombinase in Pitx2-cre mice. **e**, Example Pitx2$^{ON-PRE}$ neuron displaying tuning to moving gratings (i) and to drifting gabor patches (ii, red cross indicates centre of gaze). **f**, Overlap of tuning to all types of visual stimuli tested in premotor neurons and their tuning to head rotations (n = 17 Pitx2$^{ON-PRE}$ neurons recorded across 10 mice). **g**, A similar strategy was used to express GCaMP6f in Pitx2$^{ON-PRE}$ neurons by injecting an AAV leading to FLP recombinant-dependent expression of GCaMP6f. **h**, Example Pitx2$^{ON-PRE}$ neuron displaying tuning to moving gratings (i) and to drifting gabor patches (ii). The average ΔF/F for gratings moving in all 12 directions is shown in red (i, left, shaded area corresponds to ± s.e.m.). The average responses to gratings of two opposing directions (black) and individual trial responses (grey) are also shown. **i** and **j**, Distribution of tuned Pitx2$^{ON-PRE}$ across layers of the SC recorded using optetrodes (i, N = 10 mice, n = 17 Pitx2$^{ON-PRE}$ neurons) or Ca$^{2+}$ imaging (i, N = 7 mice, n = 74 Pitx2$^{ON-PRE}$ neurons). **k**, proportion of direction and orientation selective Pitx2$^{ON-PRE}$ neurons. **l**, Absolute value of the difference in preferred angle of direction of motion of visual stimulus (as measured using gratings or gabor patches) plotted against the 3D distance in the SC. The number of pairs is shown in parenthesis. DS/OS correspond to pairs of either OS or DS neurons compared in orientation space (180°, left), while DS pairs only consider DS neurons in direction space (360°, right). For OS/DS neurons (N = 29 cells, n = 4 mice), orange line corresponds to the median of the population, lower and upper box boundaries correspond to the 25th and 75th percentile of the population, respectively and notches correspond to the 95% confidence interval of the median calculated by bootstraping (10000 iterations). For DS pairs (N = 20 cells, n = 3 mice), orange line corresponds to the median of the population and lower and upper box boundaries correspond to the 25th and 75th percentile of the population, respectively. **m**, All direction and orientation tuned cells to full field gratings or local Gabor patches within a volume in the SC are plotted as spheres and colour-coded according to preferred motion axis (i). A, anterior; M, medial; S, superficial. The same volume with only DS neurons colour-coded indicating their preferred direction of motion is also shown (ii).

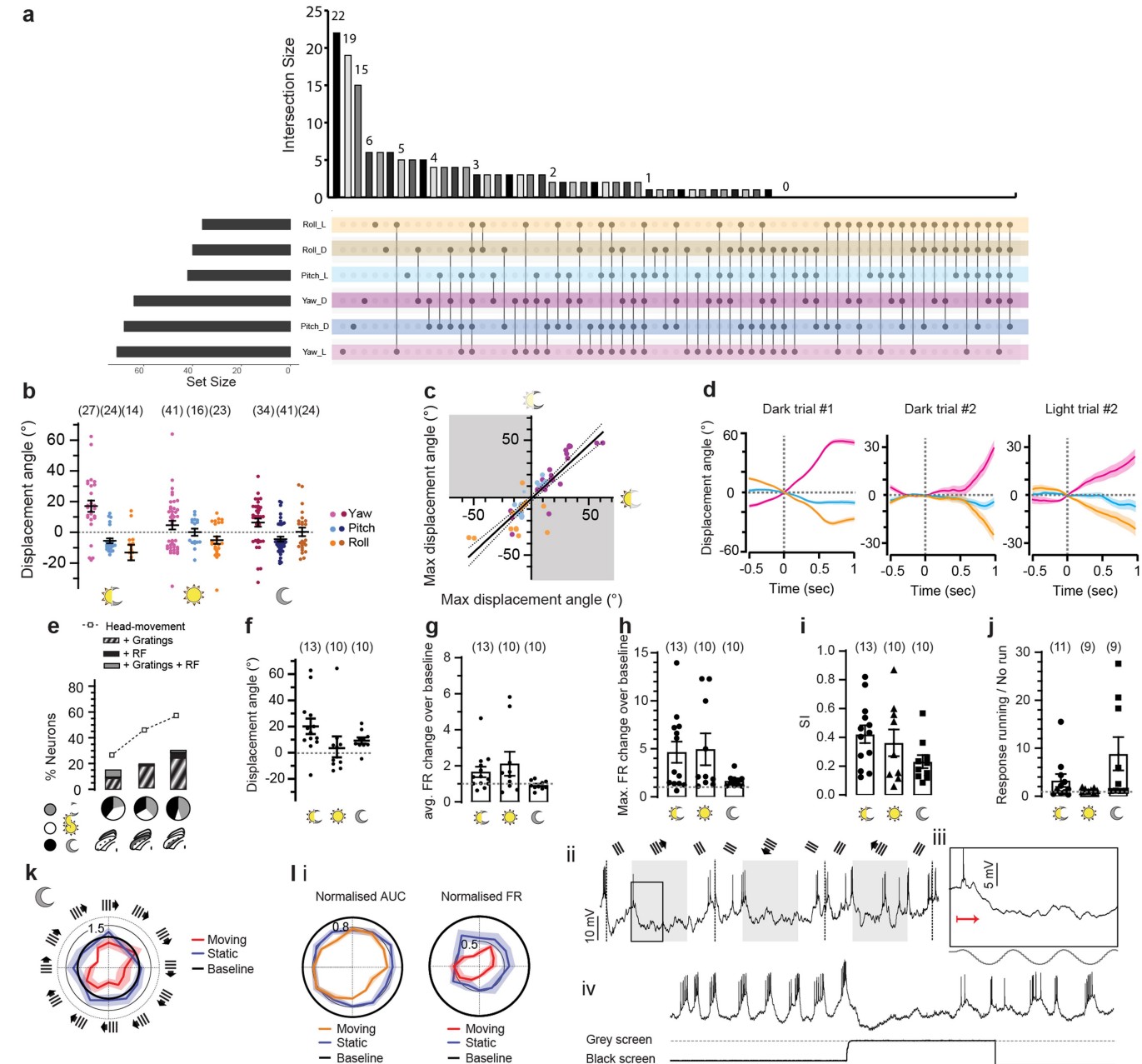

**Extended Data Fig. 7 | Distribution and characteristics of collicular neurons tuned to head rotations. a**, Intersection diagram of all 64 types of combinatorial possible tunings of the 3 head rotations (yaw, pitch, roll) tuned in light (_L) or dark (_D) conditions, produced online with Intervene[60]. **b**, Maximum average displacement angles for neurons tuned to yaw, pitch or roll either in both light and dark conditions (half-sun/half-moon symbol, tuning in light shown), light only (sun) or dark only (moon). Number of neurons tuned to each individual component are shown in parenthesis. **c**, Comparison of the displacement angle in light and dark for neurons tuned in both conditions (motor units). Linear regression: y = 0.7956x − 0.9162. Slope 95% CI: 0.6302-0.9611, shows a conservation of tuning. **d**, Spike triggered averages for the two dark trials and the second light trial corresponding to the motor neuron in Fig. 3b. **e**, Distribution of neurons tuned to head displacements as shown in

Fig. 3. Those also tuned to moving gratings, having a receptive field or both are also shown. **f**, Displacement angles for DS neurons also tuned to head movements (visuo-motor neurons). **g**, Average change in firing rate for all presentations for direction selective neurons that are also tuned to head displacements showing that neurons tuned only in darkness are overall inhibited by visual stimulation. **h**, Response to preferred direction of same neurons in **e** showing that neurons tuned only in darkness still show an increase firing over baseline to their preferred orientation. **i**, Selectivity indexes of neurons in **f. j**, Change in response to moving gratings while running for neurons in **f. k**, Example neuron tuned to moving gratings and head displacements only in darkness. **l**, Neuron recorded in vivo in whole-cell mode displaying a similar selectivity as that in **k**. The spiking activity of this cell was inhibited by light (iv). All error bars represent ± s.e.m.

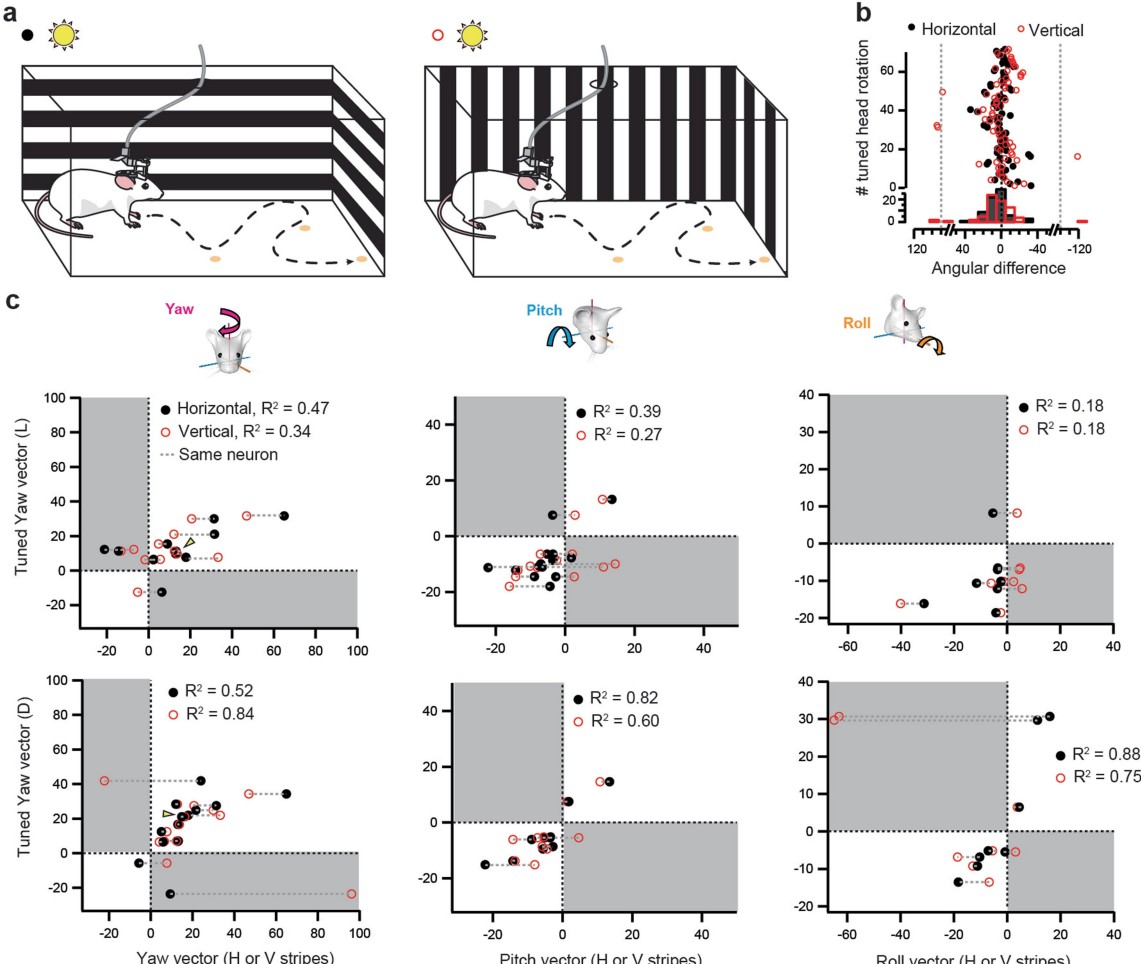

**Extended Data Fig. 8 | Head rotation tuning changes during enforced visual flow foraging. a**, Mice were left to freely forage for droplets of condensed milk while visual flow was enforced by having either horizontal (for 5 min, black dots) or vertical (for 5 min, red open circles) black and white stripes across all 4 walls of the arena (N = 4, n = 44 tuned neurons out of 134 recorded neurons). **b**, Angular difference between the maximal average displacement angle for all neurons tuned in a white arena and their displacement angle with either

horizontal or vertical stripes. **c**, Break-down of the results in **b**. The displacement angles for neurons tuned in Light (top) or Dark (bottom) for yaw (left), pitch (middle) or roll (right) is shown against their displacement angle with horizontal or vertical stripes. Shaded area indicates a change of direction of the tuned cell. The triangle indicates the markers corresponding to the same neuron tuned both in light and dark conditions. The R² are shown for all conditions.

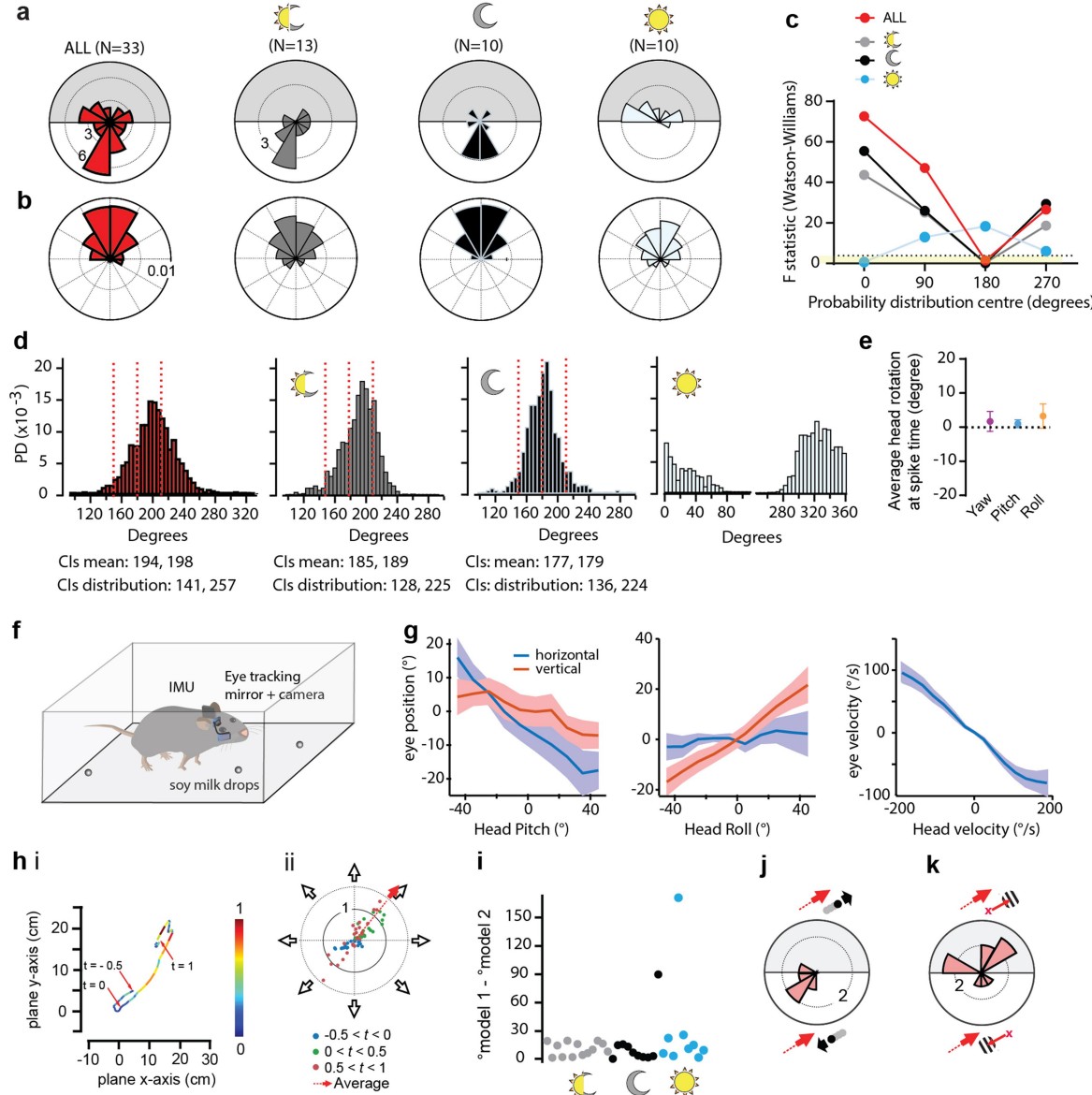

**Extended Data Fig. 9 | Characterisation of the alignment of visuo-motor neurons. a**, Visuo-motor alignment of all direction selective neurons also tuned to head displacements (red, also shown in Fig. 3). These were further divided depending on whether they were tuned to head displacements during both light and darkness (grey, also shown in Fig. 3), only dark (black) or only light (sun symbol, light blue). **b**, Probability distribution centred at 0° of a random distribution of angles with the same SD as those in **a**. Their means were also centred at 90°, 180° and 270° for Watson-Williamson analysis. **c**, Display of the F-statistics of Watson-Williamson tests performed between matching data in **a** and **b**, with the distribution at **b** centred at either 0°, 90°, 180° or 270°. Dotted line marks the critical value: 3.85. Yellow shading indicates statistical significance. **d**, Distributions of the circular means of 1000 repetitions of bootstrap with replacement for each type of neuron. The 95% confidence intervals of the means and the distribution are indicated. The red dotted lines mark 180° ± 30°. **e**, Average head rotation angle at the time of spike in yaw (magenta), pitch (blue) and roll (orange) for all neurons included in **a-d** and Fig. 3h (n = 33 neurons, error bars: ± s.e.m.). **f**, Simultaneous eye tracking and head pitch, roll and yaw measurement in freely moving mice foraging for milk drops (N = 4 mice). **g**, Horizontal (blue lines) and vertical eye position (red lines) as a function of head pitch for right eye in freely moving mice (left), head roll

(middle) and head velocity during stabilisation periods (right). Eye movements between saccadic gaze shifts counteract head rotations; Plot shows mean ± s.e.m. This was implemented on our geometrical model of the head and eye of the mouse that was used to transform 3D head rotations into gaze direction as projected onto the plane of the screen in which visual stimuli are shown. **h**, Gaze direction for the neuron in Fig. 3g after implementing compensatory eye movements to the model. Both the trajectory on the screen (i) and the decomposed and average movement vectors are shown (ii). **i**, Angular difference between the gaze direction of all visuo-motor neurons estimated using the model without (model 1) and with (model 2) compensatory eye movements implemented. **j**, Visuo-motor alignment between the direction of gaze and preferred direction of movement of a moving spot, for neurons tuned to both head rotations and moving spots (measured as in Extended Data Fig. 5). **k**, Visuo-motor alignment between the direction of gaze and the vector drawn by the centre of gaze on the screen (red cross in diagram) and the centre of the spatial receptive field measured using drifting gabor patches (red arrow to gabor patch on diagram) for neurons tuned in both conditions. 9 out of 11 of these putative premotor neurons were tuned to head rotations in light conditions only (sun).

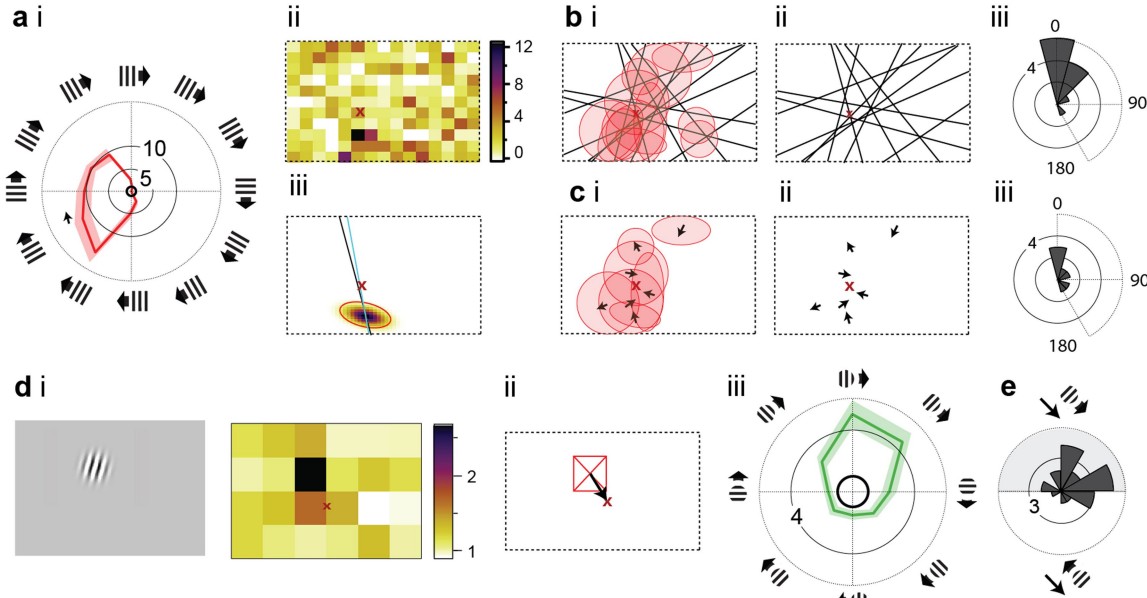

**Extended Data Fig. 10 | Concentricity of individual OS and DS neurons with localised RF. a**, Example DS neuron (i) with a spatial receptive field measured using flashing squares (ii). A 2D Gaussian was fitted to the spatial receptive field data to determine the centre of the receptive field (iii). A black line crossing the centre of the receptive field and oriented following the preferred direction angle as calculated during grating display is overlayed. The red "x" indicates the average location of gaze and the light blue line crosses both that location and the centre of the RF. **b**, All DS and OS neurons with spatial RFs. The lines indicate the orientation preference of single neurons crossing that cell's RF centre (i). ii Same as (i) but without the outline of the RFs. The angular difference between the orientation/direction preference (corresponding to black line in **a**iii) and gaze (corresponding to the light blue line in **a**iii) are also displayed (iii).

A difference of 0° indicating perfect concentricity. **c**, Same as **b** but only for DS neurons. Direction is indicated by an arrow. **d**, A second set of experiments was performed to estimate the spatial RFs of neurons across the SC using drifting gabor patches (N = 4 mice). For neurons displaying spatial and direction selective tuning, a vector was drawn between the centre of the spatial RF (red box and black arrow, ii) and the average centre of gaze on the screen (red cross). The direction of this concentric vector was compared to the preferred direction of gabor patch movement at the preferred location (green, iii, corresponding to the location marked by the red box in ii). **e**, The angular difference of those two vectors is compatible with concentric direction selectivity.

# Reporting Summary

## Statistics

For all statistical analyses, confirm that the following items are present in the figure legend, table legend, main text, or Methods section.

| n/a | Confirmed | |
|---|---|---|
| ☐ | ☒ | The exact sample size (*n*) for each experimental group/condition, given as a discrete number and unit of measurement |
| ☐ | ☒ | A statement on whether measurements were taken from distinct samples or whether the same sample was measured repeatedly |
| ☐ | ☒ | The statistical test(s) used AND whether they are one- or two-sided<br>*Only common tests should be described solely by name; describe more complex techniques in the Methods section.* |
| ☐ | ☒ | A description of all covariates tested |
| ☐ | ☒ | A description of any assumptions or corrections, such as tests of normality and adjustment for multiple comparisons |
| ☐ | ☒ | A full description of the statistical parameters including central tendency (e.g. means) or other basic estimates (e.g. regression coefficient) AND variation (e.g. standard deviation) or associated estimates of uncertainty (e.g. confidence intervals) |
| ☐ | ☒ | For null hypothesis testing, the test statistic (e.g. *F*, *t*, *r*) with confidence intervals, effect sizes, degrees of freedom and *P* value noted<br>*Give P values as exact values whenever suitable.* |
| ☒ | ☐ | For Bayesian analysis, information on the choice of priors and Markov chain Monte Carlo settings |
| ☐ | ☒ | For hierarchical and complex designs, identification of the appropriate level for tests and full reporting of outcomes |
| ☐ | ☒ | Estimates of effect sizes (e.g. Cohen's *d*, Pearson's *r*), indicating how they were calculated |

*Our web collection on statistics for biologists contains articles on many of the points above.*

## Software and code

Policy information about availability of computer code

| Data collection | Data collection for whole-cell recordings was done using Clampex 10.6 (Molecular Devices).<br>Data collection for tetrode recordings was done using Axona DaqcUSB, Arduino Mega 2560 rev 3 boards.<br><br>All further details are described in the relevant paragraph of the method section of the manuscript. |
|---|---|
| Data analysis | Analysis routines were coded in Python v2.6 or in Igor pro v6.37 (Wavemetrics).<br>Cluster cutting for single unit determination was done with Tetrode INTerface (Tint) v4.2.6 (Axona).<br>The code for computational neuronal network model developed is available at https://github.com/AnaG-R/Visuomotor-model-SC.git<br><br>All further details are described in the relevant paragraph of the method section of the manuscript |

For manuscripts utilizing custom algorithms or software that are central to the research but not yet described in published literature, software must be made available to editors and reviewers. We strongly encourage code deposition in a community repository (e.g. GitHub). See the Nature Portfolio guidelines for submitting code & software for further information.

## Data

Policy information about availability of data

All manuscripts must include a data availability statement. This statement should provide the following information, where applicable:

- Accession codes, unique identifiers, or web links for publicly available datasets
- A description of any restrictions on data availability
- For clinical datasets or third party data, please ensure that the statement adheres to our policy

Pre-processed electrophysiological data has been uploaded to Zenodo (DOI: 10.5281/zenodo.11105001). Unprocessed data will be also made available upon request.

## Human research participants

Policy information about studies involving human research participants and Sex and Gender in Research.

| Reporting on sex and gender | Not applicable |
|---|---|
| Population characteristics | Not applicable |
| Recruitment | Not applicable |
| Ethics oversight | Not applicable |

Note that full information on the approval of the study protocol must also be provided in the manuscript.

# Field-specific reporting

Please select the one below that is the best fit for your research. If you are not sure, read the appropriate sections before making your selection.

☒ Life sciences   ☐ Behavioural & social sciences   ☐ Ecological, evolutionary & environmental sciences

For a reference copy of the document with all sections, see nature.com/documents/nr-reporting-summary-flat.pdf

# Life sciences study design

All studies must disclose on these points even when the disclosure is negative.

| Sample size | Sample sizes for each experiment were selected based on variance observed in similar previous studies and taking into account practical experimental considerations. |
|---|---|
| Data exclusions | All collected data from planned experiments were included in the analyses. |
| Replication | All data were successfully replicated in independent animals (>=5) |
| Randomization | Visual stimuli were presented in a semi-randomized manner. |
| Blinding | Analysis were blinded to data collection |

# Reporting for specific materials, systems and methods

We require information from authors about some types of materials, experimental systems and methods used in many studies. Here, indicate whether each material, system or method listed is relevant to your study. If you are not sure if a list item applies to your research, read the appropriate section before selecting a response.

## Materials & experimental systems

| n/a | Involved in the study |
|---|---|
| ☐ | ☒ Antibodies |
| ☒ | ☐ Eukaryotic cell lines |
| ☒ | ☐ Palaeontology and archaeology |
| ☐ | ☒ Animals and other organisms |
| ☒ | ☐ Clinical data |
| ☒ | ☐ Dual use research of concern |

## Methods

| n/a | Involved in the study |
|---|---|
| ☒ | ☐ ChIP-seq |
| ☒ | ☐ Flow cytometry |
| ☒ | ☐ MRI-based neuroimaging |

# Antibodies

| | |
|---|---|
| Antibodies used | Primary antibodies:<br>Chicken anti-GFP (Aves Labs, GFP-1020, 1:2000)<br>Rabbit anti-RFP (Rockland, 600-401-379, 1:2000)<br>Secondary antibodies:<br>Alexa Fluor 488 donkey anti-chicken (Jackson ImmunoResearch, 703-545-155, 1:1000)<br>Cy3 donkey anti-rabbit (Jackson ImmunoResearch, 711-165-152, 1:1000)<br>Alexa Fluor 488 conjugated streptavidin (Invitrogen, 1:2000) |
| Validation | Both antibodies have been validated by the manufacturers and have been widely used by the scientific community in previous published studies.<br>In this study, we used the antibodies to assess viral expression. To evaluate specificity, we compared fluorescence between injection site and other brain areas and between injected and uninjected animals. |

# Animals and other research organisms

Policy information about studies involving animals; ARRIVE guidelines recommended for reporting animal research, and Sex and Gender in Research

| | |
|---|---|
| Laboratory animals | The following mouse strains were used: WT C57BL/6 (The Jackson Laboratory: strain #000644), Pitx2-CRE::Tau-LSL-Flp0-INLA (derived from Pitx2-CRE and Tau-LoxP-STOP-LoxP-Flp0-INLA mice, provided by Prof. James Martin and Prof. Silvia Arber respectively), Pitx2-CRE::Rosa-LSL-tdTomato (derived from Pitx2-CRE and Rosa-LoxP-STOP-LoxP-tdTomato mice, #007914, The Jackson Laboratory) and or Vgat-CRE (#016961, The Jackson Laboratory).<br>Both sexes of mice were used for anatomical experiments while only males were used for behaviour. Experiments in adult animals were performed on mice aged between 8 and 12 weeks. All experimental procedures were performed in the MRC-LMB Animal Facility. Animals were group-housed in a 12 hours light/dark cycle (7 a.m. to 7 p.m.), with temperature controlled at 19-23 C, humidity controlled at 45-65%, and with food and water ad libitum except during food restriction periods. |
| Wild animals | No wild animals were used in this study. |
| Reporting on sex | All details are described in the relevant paragraph of the method section of the manuscript. |
| Field-collected samples | No collected field samples were used in this study. |
| Ethics oversight | All animal procedures were conducted in accordance with the UK Animals (Scientific procedures) Act 1986 and European Community Council Directive on Animal Care under project license PPL PCDD85C8A and approved by The Animal Welfare and Ethical Review Body (AWERB) committee of the MRC-LMB |

Note that full information on the approval of the study protocol must also be provided in the manuscript.

