## [Peer Review file · Nature]

Manuscript Title: Kinetic features dictate sensorimotor alignment in the superior colliculus

Reviewer Comments & Author Rebuttals

Reviewer Reports on the Initial Version:

Referee #1 (Remarks to the Author):

The manuscript by González-Rueda et al. addresses the mapping of sensory to motor space in the mouse superior colliculus (SC). Using a combination of intra- and extracellular recordings, optogenetics, behavioral analysis and modeling they propose a framework for collicular function that is different from existing ones. Specifically, based on their data, the authors propose that motor units in the deeper layers of the SC that trigger gaze changes are not driven by static visual stimuli in a specific visual field location. Rather, these motor units are best excited by dynamic stimuli, like moving gratings, with a direction of movement in visual space opposite to the direction in gaze change that the neurons cause. The authors speculate that such an arrangement facilitates interception of the mouse with moving targets during behaviors like prey capture.

Overall, this idea is very intriguing, and I believe that the authors are up to something potentially very interesting. However, in its current state, the paper appears premature, with several gaps in the argument and important control experiments lacking.

1. One of the authors' main points is that deep layer motor units do not have receptive fields (title of Figure 2 legend and line 256). There is a misconception of the term receptive field (RF) here: by definition, these cells have a receptive field, otherwise they would not respond to visual stimuli. What seems to be the case, but even that is not convincingly shown, is that the RFs of these cells cannot be mapped with the flashing black and white spot stimuli used. This indeed indicates that these cells have non-linear RFs and are primarily driven by moving stimuli. However, the authors only used large field moving grating stimuli to determine the orientation and direction preference of these units. What they did not do is to map out the potential spatial structure of the RFs using e.g. small gabor patches with drifting gratings. This way it would have been possible to determine whether these RFs are located at specific positions in the visual field, and whether these locations are related to the gaze shift these neurons drive.

2. A crucial point is the authors' ability to determine the shifts in gaze coded by the deep layer neurons. They measure head movements and then infer the direction and amplitude of gaze shifts by using the data from Meyer et al.. It is unclear, however, what the coupling between eye and head movements is like under the authors' experimental conditions, with mice foraging. It would be essential to monitor eye movements in freely moving mice, as has been done by several groups.

3. Some of the data, while interesting, appear somewhat detached from the rest of the paper. In particular, the results shown in Figure 1 are not contributing a lot to the main line of reasoning. While the circuitry of a subset of cells involved is worked out, the picture is incomplete and crucial questions are not addressed. E.g. it would be important to know how wide and narrow field neurons

map onto the neurons presynaptic to Pitx2 cells. In many places of the paper the authors speak of “putative” presynaptic neurons, also reflecting that key aspects of the circuitry are unclear. These unknowns are also present in the schematic in Fig. 1j, which depicts a lot of connections, but does not convey a clear picture. In addition, there are very unclear symbols, e.g. the dots in the connection from the upper layer retino-recipient cells to Pitx2 cells, or the pink dots in the deep layers.

Minor:

4. Fig. 1b is very low res, some arrowheads point into the void.
5. Line 130: ChR2 was not injected, rather a virus.
6. Line 159: wide and narrow field neurons are mentioned here, but they are not explained, and no reference is given.
7. Line 162: I am not particularly happy with the term “static receptive field”. At the very least this should be properly defined.
8. Lines 164-166: this is very speculative. How do they know these are narrow-field neurons?
9. Line 170: I am not sure what to make of the lower responses to static gratings. Most cells will show only very small responses to static stimuli, and tuning curves based on such standing grating might not be very meaningful. Were these responses measured before and after the grating moved?
10. Line 173: would be nice to briefly state here how recordings in head-fixed, running mice were done
11. Line 174: do they mean “tuned to” or “responsive to”?
12. Line 178: unclear what the “1.9x” refers to.
13. Line 185 ff.: Not really clear why exactly these receptive field properties were measured, this seems somewhat arbitrary. Why were directional looming stimuli used, and not expanding ones at different visual field positions? Is there any relevance to the observation of oscillatory activity following changes in luminance? These (and other) data reported are not really relevant for the argument made.
14. Line 194: what are “deep intermediate layers”? In general, the designation of the SC layers is not very stringent throughout.
15. Line 202: the papers cited here are classic ones, but there are newer ones that go beyond these basic findings.
16. Fig. 2f: It is not made clear how the latencies are determined. They are very long, even in the superficial layers, (125 ms, mean or median?; what are the red crosses?), in V1 they are around 45 ms.
17. Fig. 2gi: must make clear which layers these cells are in.
18. Fig. 2h: how is a responsive neuron defined?
19. Line 240: How do they know these neurons “drove” movements? Also, what is meant by contralateral head movement?
20. Line 245: what drives these upper layer neurons in the dark?
21. Line 257: eye movements in head fixed mice are very small, so I doubt whether the absence of neuronal responses is very meaningful.
22. Line 262: an assumption that is not tested.
23. Line 287: “rapid interception of targets” is very speculative.
24. Fig. 3b: is this in light or in darkness?

25. Line 404: please put reference.
26. Line 410: "Primates": Authors should discuss that what they found in the mouse might be specific to animals that lack foveate vision, where orienting movements of the eyes and head might have a very different function from that for example in primates.
27. Line 934: "73° by 104°": add dimensions
28. Line 940: there were no colored stimuli, right?
29. Line 1055: related to point 1, it is very unclear which criterion was actually used then to determine whether a neuron had a receptive field

Referee #2 (Remarks to the Author):

The manuscript “Kinetic features dictate sensorimotor alignment in the superior colliculus” shows that neurons in the intermediate and deep layers of the superior colliculus have poor responses to static spots. Many of the neurons in these layers have activity preceding specific head-movements. The head-movements associated with the activity of individual neurons in the superior colliculus is in the opposite direction to the preferred visual flow direction of the neurons. The preferred visual motion direction is thus aligned to the direction of visual motion induced by the head movement when the neuron is active. The manuscript presents the response to moving targets in a three-layer network model that is wired in a way that is consistent with their findings in Pitx2 neurons and the literature. The motion resulting from this model would result in quickly directing the head-gaze to a target that is moving in a centripetal direction.

The main finding of the link between head-direction and visual-flow preference is novel, and the interpretation that this would benefit fast interception of centripetal moving targets is interesting. Furthermore, the paper proposes a novel behavioural benefit for the concentricity of direction selective responses in the superior colliculus.

I find the manuscript certainly interesting and it is significant for our understanding of the superior colliculus and gaze-direction. The data are well presented and the model is novel. Overall the data support the conclusion, but I am left with a number of questions, in particular about the validity of the proposed model for explaining target-intercept behaviour.

First, the paper makes predictions about the change in head-direction in response to moving stimuli, in particular that the responses to centripetal stimuli are different from those to centrifugal stimuli. How do these predictions work out in behaviour? In particular, the model suggests that a grating drifting with a centripetal motion is met with a head movement in the opposite direction. This is opposite to what I would expect from all the work on the optokinetic reflex with similar stimuli in mice and other animals, in particular the work with a rotating drum from Douglas, Prusky and other. Can the model predictions be reconciled with these earlier results on gaze-change in response to drifting gratings?

Second, direction selectivity was tested with drifting gratings and this is used as input to the model. The interpretation of the data, however, is given in the context of target interception. It would thus make sense to measure the directional selectivity to moving targets. Data to directional looming stimuli is presented in the extended data, but it is unclear how the tuning found in these recordings match the tuning on direction selectivity with gratings and whether this data supports the model. Can this analysis be added?

Third, the authors show that they cannot compute receptive fields in motor neurons using the flashed spots, and then go on to say that these neurons in the deeper layers do not have receptive fields. This seems a stronger conclusion than the presented data currently supports. Perhaps the flashed spots are not salient enough when shown in the context of a new spot appearing every 750 ms, or some local motion feature necessary is necessary to evoke a response. Do the authors think that the neurons would respond to motion anywhere on the screen, or that there still is an exclusive

visual region like a receptive field in which motion triggers response. It would be nice to see this clarified, or otherwise, that the conclusion that the neurons lack a receptive field is weakened. Furthermore, I think that the presented awake data is important for the interpretation given to the data. However, the responsiveness to the flashed spots is very low (19%) in the superficial layer. This would suggest that, in the words of the authors, neurons in the superficial layer lack receptive fields. It appears that the flashed spot stimulus is in general not an effective stimulus to drive the superior colliculus, even for the superficial layers. This weakens the conclusion about the lack of rfs in the motor layers.

Fourth, there is a large difference between the anesthetized patch data shown in Fig. 2h and the awake tetrode data in Fig. 2n. I am just curious about the origin of this difference. Can the authors show or, at least, discuss whether this difference is caused by the state of the animal or by the method of recording?

These are my main questions. I also have a number of comments on the details of the manuscript.

L52. Here and throughout the paper 'motor layers' is used, but no definition is given in terms of depth or anatomically identified layers. From Fig. 2 in combination with the text, I deduce that both the intermediate and deep layer are included in this, but please add this explicitly. I looked for help in references 15 and 16 that are cited at this point, but they do not use the term 'motor layers'.

L62 "Specifically, collicular motor units are preferentially tuned to visual flow of the opposite direction to the movement that they decode" This line also occurs in the discussion. I am a little confused by the word "decode". The motor neuron activity precedes the motion, while the word "decode" suggests that the activity follows the motion. The example neuron in Fig 3bg actually also seems to decode the head movement just before the activity, which is aligned to the visual stimulus. Is there not a better description than 'decode'? Also at this early point in the manuscript replacing "movement" by "head movement" helps the reader to understand the sentence, because otherwise movement could be taken to mean the movement of the visual stimulus.

L163. Here it says 56% of neurons in motor layers, but the figure shows $(6+3)/(10+8) = 50\%$ of the neurons. Please explain better or correct.

L194. Here 'deep intermediate layers' is used. Is this 'deep/intermediate layers' or the deeper part of the intermediate layer?

Fig 2. The figure title is "Neurons in motor layers lack receptive fields ...". This is a narrow definition of receptive field. What is shown in the figure is that neurons in the intermediate and deep layer do not respond well to small flashing spots.

Fig 2f. I cannot find in the Methods how this latency is computed. Can you add this?

Fig 2f. For the deep layer only for 2 neurons the latency was computed. It is a stretch to use an ANOVA for this. You could consider removing it.

Fig 2h suggests that there were 3 cells responsive to gratings but in Fig 2f the latency is only computed for 2. What were the exclusion criteria that were used?

Fig 2h,n. I cannot find how responsive (to gratings or RF) was defined. Can you add this?

L200: "This is surprising given that visual-motor integration in the SC has generally been hypothesised to rely on the alignment of static sensory and motor features, such as stimulus receptive field and movement end-points" No need to change the text, but this is a bit of a straw man hypothesis. I think that a general hypothesis to explain the primate data has been that of the presence of a saliency map in the superficial layers transforming into a gaze-directing map in the deep layers, rather than neurons only responding to static spots of lights. This interpretation could be reconciled with the currently presented data, by interpreting the static spots used in the study as just lacking in saliency.

L906. Should "200ms preceding and 800ms following the peak amplitude of a spike" be "200us ... 800 us..."?

L933. Was there any time between the flashes?

L1120. I find Supplementary figure 1 very confusing. The screen appears upright, i.e. parallel to the drawn z-axis. However, the point $P1 = (x0,0,0)$ is drawn on the screen above $P2 = (0,y0,0)$. The both have a z-coordinate of 0 suggesting that the screen has z-coordinate 0 and is lying flat. I am probably misinterpreting the figure, but from this start I am confused and have not checked the calculations in detail.

L1197. The firing rate update rule includes epsilon twice. The second epsilon does not have a dependence on the time step. The diffusion rate of this model would then depend on the time step. Is the given equation indeed correct? A more common choice would be to use something that does not have such an explicit dependence on the time step size.

Referee #3 (Remarks to the Author):

There is substantial data supporting that the superior colliculus is organized into several laminae with convergence of inputs from multiple sensory modalities (i.e. visual, somatosensory and auditory cues) and regulates distinct motor behaviors, including defensive behaviors (i.e. escaping or freezing to a looming visual stimulus and quick suspension of locomotion to illuminance change), and goal-orientation behaviors (i.e. hunting behavior). Undoubtedly, these behaviors are consequence of evolution for survival to avoid natural threats or harvest essential energy. On the one hand, mounting studies have made significant progress towards understanding the neural mechanisms of these behaviors from the perspective of neural circuits, which SC mainly contributes to integrating and transforming the multisensory. On the other hand, a cluster of well-aligned neurons which marked with Pitx2 and distributed in deep layer of SC is reported to control eye and head movement. Of note, goal-orientation like foraging behaviors depends on the head rotation and eye saccade in the rodent animals. However, the internal integration of intra-collicular network about sensorimotor transformation remains to be interrogated systematically. Hence, combined with the two fundamental role of sensory integration and motor modulation in SC, some intriguing and feasible points about connectivity of the two function of SC are worthy to dig out.

That being the case, in this study, a new kinetic alignment model which decodes potentially goal-orientation behavior is proposed aiming to claim the functional alignment between sensory and motor maps unambiguously. The authors dissected a microcircuit among separate layers of SC by identifying premotor neurons (Pitx2 on-pre) which synapse onto Pitx2 on neurons accurately. The putative motor units, pitx2 on neurons, received direct visual signals transmitting from Pitx2 on-pre in superficial and indirect from intermediate layers, both of which responded preferentially to kinetic visual stimuli. Then these neurons drove opposite head movement for interception of target. This manuscript pursuing some of the long-standing questions about sensorimotor transformation of superior colliculus might greatly enriches our understanding in a more general sense. It provides more compelling neurobiological evidences and theoretical basis for deconstructing the survival instinct.

However, although the research theme is intriguing, there are still some issues I concerned yet to refer.

1. In figure1h, Pitx2 on-pre neurons in intermediate layers performed strong disinhibition effect after GABAR blockade, accompanying with long duration of depolarization up to 20s. Theoretically, the cell membrane potential should be in a state of sustained depolarization. However, the example trace of EPSP showed in figure1h (middle) decayed back to baseline at about 100 ms. The statistics of the decay time of EPSP should be provided.
2. The axon terminals of RGC mainly distributed locally in superficial layer of SC as figure 1e showed. However, a fraction of neurons in intermediate deep layers of SC still received monosynaptic retinal inputs (figure1g). It is necessary to reconfirm the accuracy of the data.
3. The number of Pitx2 on neurons labeled by mCherry in figure1a was obviously less than the result reported before (ref 3 in manuscript) . It would be compelling to figure out the reason and change another figure.
4. It was interesting and attractive to find the disinhibition phenomenon of Pitx2 on-pre neurons evoked by photostimulation and GABAR blockade in intermediate/deep layers of SC in figure 1h. The potential mechanism and physiological significance would be illuminating to illustrate.
5. In figure2b I, the neurons labeled by biocitin in the superficial layers of SC were defined as narrow-

field neuron. But there was no evidence to prove the identity of the neurons. Data about the morphological, intrinsic electrophysiological properties, or specific markers (Grp-KH288) would help to clarify this. Similarly, the neurons recorded in the intermediate/deep layer of SC were assumed as pitx2 on-pre neurons in figure 2c. Although these neurons showed subthreshold responses triggered by direction-selective moving grating stimuli, some data demonstrated in figure 3e that motor units in the same layer of SC also tuned to moving gratings. Thus it is lack of solid evidence to confirm the neuron identity, Pitx2 on-pre or motor neuron. Further investigation is required to verify this entire issue

λ 6. In line 166~169, the author speculated that the tonic inhibitory currents on the collicular premotor network resulted in the lack of suprathreshold tuning in neurons located in intermediate layer of SC. It would be reasonable to infer that these neurons would display suprathreshold tuning, if GABAR antagonist was administered locally in SC combined with current stimulation from pipette, or light stimulation from an optical fiber. Additional experiments to make clear were anticipated in follow-up work.

λ 7. The number of neurons from deep layer of SC was too few. The one-way ANOVA was not recommended. It is necessary to add more experimental data.

λ 8. In line 257 and extended figure 4h~j, the data indicated no correlation between motor neurons in deep layer of SC and eye movement, which was in contradiction to the result reported before (ref 3). The prior study revealed that photostimulation of Pitx2ON neurons led to the execution of rapid but low-amplitude eye movements (<5 degrees). It was not clear whether the negative data obtained here was due to the insufficient driving force of visual stimulation, or something else. Thus some convincing explanation was expected.

9. In line 253~254, the authors discovered half of all motor-tuned neurons across all layers were also tuned to moving gratings. However, on the contrary, in the foraging test above (line 248~250), the motor tuning was not influenced by self-generated visual flow. Considering another half cluster of motor neurons tuning head movement only, we could not rule out the possibility that the no alteration of motor is on account of less data. Comprehensive evidence would be required.

Author Rebuttals to Initial Comments:

We would like to thank the reviewers for their generous commitment to the revision of our manuscript and for their insightful comments, which have contributed to strengthen our paper. We have done our best effort to experimentally address each point when possible and edited the manuscript following the reviewer's suggestions. The new version of the manuscript includes 8 new sets of experiments (listed below) and extra analysis. In order to adhere to editorial specifications; the Figures, Extended Figures and the main text have been altered to accommodate the new findings. Hereafter you will find our response to each point raised by the reviewers in blue, with clear references to any changes in the Figures and manuscript text, including line numbers, in red. Both line numbers and Figure numbers correspond to the revised version of the manuscript. References across this letter, including those contained in excerpts from the manuscript, are numbered in their order of appearance here and listed at the end of this document.

New experimental datasets:

1. New visual stimulation paradigms in WT mice: drifting Gabor patches (24 locations and 8 directions of movement) and small dot moving across the screen in 8 different directions.
2. Recording of responses to moving gratings before and after isoflurane anaesthesia.
3. *In vivo* recording and optotagging of Pitx2^{ON} neurons expressing ChR2 using optetrode recordings.
4. *In vivo* recording and optotagging of Self-Inactivating Rabies virus (SiR) labelled Pitx2^{ON-PRE} neurons expressing ChR2 using optetrode recordings both in freely moving conditions and head restrained conditions during visual stimulation.
5. *In vivo* multiphoton optical recording of SiR labelled Pitx2^{ON-PRE} neurons expressing GCaMP6 in head restrained conditions during visual stimulation in a 2-photon microscope.
6. Recording of head-eye correlation using a miniaturised eye camera and an IMU system to simultaneously record head and eye rotations in a freely moving foraging task.
7. In situ hybridisation of markers of known collicular cell types in slices containing SiR traced Pitx2^{ON-PRE} neurons.
8. Replicate of enforced visual flow during freely moving foraging.

Referee #1 (Remarks to the Author):

The manuscript by González-Rueda et al. addresses the mapping of sensory to motor space in the mouse superior colliculus (SC). Using a combination of intra- and extracellular recordings, optogenetics, behavioral analysis and modeling they propose a framework for collicular function that is different from existing ones. Specifically, based on their data, the authors propose that motor units in the deeper layers of the SC that trigger gaze changes are not driven by static visual stimuli in a specific visual field location. Rather, these motor units are best excited by dynamic stimuli, like moving gratings, with a direction of movement in visual space opposite to the direction in gaze change that the neurons cause. The authors speculate that such an arrangement facilitates interception of the mouse with moving targets during behaviors like prey capture.

Overall, this idea is very intriguing, and I believe that the authors are up to something potentially very interesting. However, in its current state, the paper appears premature, with several gaps in the argument and important control experiments lacking.

1. One of the authors' main points is that deep layer motor units do not have receptive fields (title of Figure 2 legend and line 256). There is a misconception of the term receptive field (RF) here: by definition, these cells have a receptive field, otherwise they would not respond to visual stimuli. What seems to be the case, but even that is not convincingly shown, is that the RFs of these cells cannot be mapped with the flashing black and white spot stimuli used. This indeed indicates that these cells have non-linear RFs and are primarily driven by moving stimuli. However, the authors only used large field moving grating stimuli to determine the orientation and direction preference of these units. What they did not do is to map out the potential spatial structure of the RFs using e.g. small Gabor patches with drifting gratings. This way it would have been possible to determine whether these RFs are located at specific positions in the visual field, and whether these locations are related to the gaze shift these neurons drive.

We agree with the reviewer that the use of the term "receptive field" (RF) when only referring to a particular visual stimulation paradigm could be misleading. To avoid any misinterpretation we have now replaced all the previous references to "RF" in the text and figures for static spatial receptive fields" (ssRFs). See also response to point 7 below. Furthermore, we have now changed the title of Fig. 2: "Premotor neurons and neurons in motor layers respond preferentially to visual flow. "

In order to address whether there is a spatial structure of the dynamic RFs previously mapped using moving gratings, we performed a completely new set of experiments including the presentation of drifting Gabor patches as suggested. We performed tetrode recordings in 4 C57BL6 mice across all layers of the superior colliculus (SC), first in freely moving conditions in an open field to map neurons' tuning to gaze shifts during foraging and then under head restrain conditions to map their responses to small Gabor patches (see Materials and Methods and Extended Data Figure 5, also included below). As a control, we also recorded their response to full field gratings. We found that more neurons responded to drifting Gabor patches compared to moving gratings. The difference is particularly prominent in the superficial layers where 89% of neurons responded to Gabor patches vs 63% of recorded neurons in superficial layers responding to full field moving gratings (including direction selective DS neurons, orientation selective OS neurons and neurons with significant but direction/orientation independent overall change in activity during moving gratings, 42% if only OS/DS

neurons are considered, Extended Data Figure 5h). This was likely due to the combination of location-specific and DS/OS responses elicited using Gabor stimuli. Similarly to moving gratings, the proportion of neurons tuned to drifting Gabor patches decreased in intermediate and deep layers; however, while most neurons responding to Gabor patches in superficial layers displayed location-specific responses linked to a localised spatial visual receptive field independent of the direction of drift of the Gabor patch (71%), in lower intermediate and deep layers, OS/DS neurons were predominant, accounting for 91% of all neurons responding to Gabor patches (Extended Data Figure 5i). This result further strengthens our finding that kinetic features of the visual scene are those that are primarily transmitted to motor layers in the SC. We include a couple of example responses of neurons tuned to Gabor patches, together with their proportion across layers, combined responses with other visual stimuli and angular difference of preferred direction of movement during full-field grating stimuli and Gabor patches in Extended Data Figure 5a-c, h-k.

These results are now discussed in the main text of the manuscript:

Line 206-226: “We also assessed the responses of neurons to full-field changes in luminescence, directionally moving spots mimicking a moving target, as well as to localised Gabor patches drifting in 8 different directions (see methods, Extended Data Fig. 5), as examples, respectively, of ethologically relevant full-field static and spatially localised kinetic visual stimuli. [...] Similarly, 71% of the responses to drifting Gabor patches in superficial layers corresponded to the activation of spatial receptive fields while this decreased to 24% in lower intermediate and deep layers where direction and orientation selective neurons were more prevalent (Extended Data Fig. 5g-i) , further strengthening the notion that kinetic features of the visual scene are preferentially transmitted to the motor layers of the SC. Importantly, we found a conservation of the preferred direction tuning among stimulation paradigms, whereby neurons tuned to a particular direction of gratings, were tuned to [...] drifting Gabor patches of the same direction (average preferred [...] °gratings-°Gabor: 27 °, Extended Data Fig. 5k).”

The Materials and Methods section was also updated to include these new analyses:

Line 1208-1212 (M&Ms > Behaviour > visual stimulation): “In order to assess the spatial receptive field of neurons to moving stimuli we used moving Gabor patches at 24 different locations (6 along x axis and 4 along y axis) moving in 8 different directions (45° steps) for 1.5 seconds (spatial frequency: 0.08 cycles per degree and 2.83 Hz temporal frequency). All directions and locations were randomly presented 3 times for a total of 576 presentations corresponding to 3 times 8 directions at 24 locations.”

Line 1338-1342 (M&Ms > Quantification and Statistical Analysis > visual tuning): “Direction and orientation selectivity for small moving spots and for Gabor patches was measured as described for gratings. Two separate analyses of selectivity and preferred angle of tuning were performed for Gabor patches: a location-independent analysis by averaging the responses to each direction of movement across all locations and another analysis only considering the responses at the location of maximal average activity. “

Line 1351-1353 (M&Ms > Quantification and Statistical Analysis > visual tuning): “The same analysis was performed to assess whether neurons had a spatial and kinetic RF as measured with

Gabor patches. The centre of the location of maximal response was used as centre of the RF.”

EXTENDED DATA FIGURE 5

Extended Data Figure 5. Mapping of visual tuning properties in response to drifting Gabor patches, small moving spots and changes in luminescence in the SC.

a, Gabor patches drifting in 8 different directions were randomly presented at 24 locations of the screen. **b**, Example neuron displaying location-specific tuning to moving gratings measured by averaging all directions of movement per location (left), location-independent directional tuning as measured by averaging the response to drifting patches for each direction across all locations (middle, yellow), and a strong direction tuning at the preferred location (right, green). Shaded areas in polar plots indicates \pm s.e.m. **c**, Example neuron without a location preference but tuned to particular direction of movement of the Gabor patches. **d**, Mice were also presented with a small spot (1.3 cm diameter) starting at 8 locations along the corners and border of the screen (red dots) and moving in 8 directions towards and away from the mouse’s centre of gaze on the screen (red cross). **e**, Example tuning of a neuron responding to a small spot moving in a particular orientation. Shaded areas in polar plots indicates \pm s.e.m. Average firing rate for each direction is shown as a heat map and the average time change across directions to identify if a neuron prefers approach vs retreat is shown at the bottom (i). The tuning of this neuron corresponds to the spot crossing its spatial receptive field mapped using drifting Gabor patches (ii). **f**, Example tuning to a small moving spot for a neuron without spatial receptive field. **g**, Example neuron tuned to full field moving gratings (i), Gabor patches (ii) and moving spot (iii). **h**, Distribution of neurons responding to particular locations in space mapped using Gabor patches (spatial Gabor, blue), neurons showing direction or orientation selectivity mapped using Gabor patches (Gabor OS/DS, green) or full field gratings (red), neurons tuned to moving spots (spot OS/DS, black) or tuned to head rotations (head tuned, purple). The number of recorded neurons per depth of recording, indicated

by the shading in the SC diagram, is shown in parenthesis (N = 4 mice). **i**, Distribution of neurons tuned to particular locations (spatial Gabor, blue), direction or orientation selective neurons (Gabor OS/DS, green) or both (grey) among all neurons tuned to drifting Gabor patches. **j**, Overlap of the tunings shown in **h** for individual neurons, computed using Intervene. Static features are preferentially represented at the top, while kinetic ones are preferentially represented towards the bottom. Note that the number of neurons is represented by the shade of colour and not by the surface of the intersection. **k**, Angular difference between the preferred angle of tuning measured using full field gratings and moving spots (black) or Gabor patches (green).

We found a small population of neurons tuned to head rotations that also responded to drifting Gabor patches at specific locations in space. We found that the average difference between the direction of gaze these neurons are tuned for and the vector drawn between the expected centre of gaze at rest during head restrained visual stimulation and the location of the Gabor patch in the screen eliciting the highest response was 46° , implying a rough alignment between these two vectors. This alignment is in agreement with the classical model of alignment between a location in space and the end-point of the movement needed to reach it. However, it is important to note that most of these neurons were located in superficial and upper-intermediate layers and tuned to head rotations only in light conditions, with only one out of 11 of these neurons preserving their tuning in darkness, implying these neurons are not canonical motor units (i.e. motor tuned both in light and dark conditions). We have now captured these findings in Extended Data Figure 9k.

Line 330-335: “We also found an alignment between gaze direction and preferred Gabor patch location for neurons tuned to head rotations and Gabor patches (Extended Data Fig. 9k). Over 90% of these units maintained their motor tuning only in light conditions but lost their head-rotation tuning in darkness, implying that, although these are not canonical motor units, they are reminiscent of visuo-motor units found in primates¹.”

EXTENDED DATA FIGURE 9

Extended Data Figure 9. Characterisation of the alignment of visuo-motor neurons.

[...] **k**, Visuo-motor alignment between the direction of gaze and the vector drawn by the centre of gaze on the screen (red cross on diagram) and the centre of the spatial receptive field measured using drifting Gabor patches (red arrow to Gabor patch on diagram) for neurons tuned in both conditions. Only 1 out of 11 of these putative premotor neurons were tuned to head rotations in both light and dark conditions.

These results are in line with the proposed hierarchical organisations across layers of the SC whereby location specific visual features lose relevance deeper within the SC while kinetic features gain relevance. Moreover, these results align with the modular organisation of concentric location → direction → motion we propose and were already encapsulated by the model in Figure 4 that predicts direction selective neurons in deep layers would still carry some spatial information. We have also updated Extended Data Figure 10 with the concentric alignment displayed by neurons tuned to Gabor patches with location-specific responses and direction or orientation selectivity at those locations.

EXTENDED DATA FIGURE 10

Extended Data Figure 10. Concentricity of individual OS and DS neurons with localised RF.

d, A second set of experiments was performed to estimate the spatial RFs of neurons across the SC using drifting Gabor patches ($N = 4$ mice). For neurons displaying spatial and direction selective tuning, a vector was drawn between the centre of the spatial RF (red box and black arrow, ii) and the average centre of gaze on the screen (red cross). The direction of this concentric vector was compared to the preferred direction of Gabor patch movement at the preferred location (green, iii, corresponding to the location marked by the red box in ii). e, The angular difference of those two vectors is compatible with concentric direction selectivity.

2. A crucial point is the authors' ability to determine the shifts in gaze coded by the deep layer neurons. They measure head movements and then infer the direction and amplitude of gaze shifts by using the data from Meyer et al.. It is unclear, however, what the coupling between eye and head movements is like under the authors' experimental conditions, with mice foraging. It would be essential to monitor eye movements in freely moving mice, as has been done by several groups.

In order to experimentally address this point, we collaborated with the lab of the last author of Meyer et al. publication, Dr. Jasper Poort. In the revised version of the manuscript, we have included new recordings using their eye camera system and their analysis routines in combination with our head sensor and our open field foraging paradigm to replicate our experimental conditions. Using this method we were able to assess the coupling between eye and head movements in 4 C57BL6 mice while foraging. These new recordings are comparable to their previous results. We have now included this data in Extended Data Figure 9 and implemented our geometrical model of the mouse head-eye system by replacing Meyer et al. data with the newly acquired one. The updated model, having no effect on the estimation of gaze direction, supports our previous conclusions.

Line 338-343: "Although all visuo-motor units were recorded during goal-oriented foraging, we also considered the effect of purely compensatory eye movements during head rotations by implementing our geometrical model of the mouse gaze with head-eye rotations measured during a foraging task (Extended Data Fig. 9f-h). Incorporating these compensatory eye movements in the model estimating the average gaze-direction did not change the results (Extended Data Fig. 9e)."

Line 1241-1254 (M&Ms > Behaviour > eye tracking): "In order to measure eye movements in freely moving mice we used a custom head-mounted eye and head tracking system, previously described^{2,3}. Briefly, we used a commercially available camera module (1937, Adafruit, USA; infrared filter removed). A custom 3D printed camera holder with a 21G cannula (Coopers Needle Works, UK) was used to hold the camera, IR LEDs (VSMB2943GX01, Vishay, USA) and a 7.0 mm x 9.3 mm IR mirror (Calflex-X NIR-Blocking Filter, Optics Balzers, Germany). A 340-

340-340connector (852-10-00810-001101, Preci-Dip, Switzerland) was used to attach the camera holder to the head plate of the mice. Mice were head-fixed, and the mirror's position was adjusted until the eye was in the centre of the eye camera. Epoxy (Araldite Rapid, Araldite, UK) was used to fix the mirror position. Single-board computers (Raspberry Pi 3 model B, Raspberry Pi Foundation, UK) recorded camera data at 30 Hz, capturing images of 1296 x 972 pixels per frame for eye camera. The head roll, pitch, and yaw were estimated using an inertial motion unit (IMU) including an accelerometer, gyroscope and magnetometer using previously described methods⁴ and open source arduino code (github.com/razor-AHRS) using an Arduino Mega 2560 rev 3."

Line 1374-1383 (M&Ms > Quantification and Statistical Analysis > Extraction of pupil position in freely-moving mice): "Eye tracking was performed as previously described². Briefly, we tracked the position of the pupil, defined as its center, together with the nasal and temporal eye corners. The eye corners were used to automatically align the horizontal eye axis. 30-50 randomly selected frames were labeled manually for each recording day. The labeled data were used to train a deep convolutional network via transfer learning using open source code⁵ (github.com/AlexEMG/DeepLabCut). The origin of the eye coordinate system was defined as the mid-point between the nasal and temporal eye corners. Pixel values in the 2-D video plane were converted to angular eye positions using a model-based approach developed for the C57BL/6J mouse line used in this study⁶. Saccades were defined as rapid, high-velocity movements occurring in both eyes with a magnitude exceeding 350 deg/s."

Line 1478-1482 (M&Ms > Modelling > 3D head rotations to 2D screen plane transformation model): "The correction rotation matrix in head frame, $R_{correction}^H(\phi, \theta, \psi)$, was computed from the head-eye rotations measured and represented in Extended Data Figure 9f,g."

EXTENDED DATA FIGURE 9

Extended Data Figure 9. Characterisation of the alignment of visuo-motor neurons.

[...] **f**, Simultaneous eye tracking and head pitch, roll and yaw measurement in freely moving mice foraging for milk drops (N = 4 mice). **g**, Horizontal (blue lines) and vertical eye position (red lines) as a function of head pitch for right eye in freely moving mice (left), head roll (middle) and head velocity during stabilisation periods (right). Eye movements between saccadic gaze shifts counteract head

rotations; Plot shows mean \pm s.e.m. This was implemented in our geometrical model of the head and eye of the mouse that was used to transform 3D head rotations into gaze direction as projected onto the plane of the screen in which visual stimuli are shown. **h**, Gaze direction for the neuron in Fig. 3g after adding compensatory eye movements to the model. Both the trajectory on the screen (i) and the decomposed and average movement vectors are shown (ii). **i**, Angular difference between the gaze direction of all visuo-motor neurons estimated using the model without (model 1) and with (model 2) compensatory eye movements implemented.

3. Some of the data, while interesting, appear somewhat detached from the rest of the paper. In particular, the results shown in Figure 1 are not contributing a lot to the main line of reasoning. While the circuitry of a subset of cells involved is worked out, the picture is incomplete and crucial questions are not addressed. E.g. it would be important to know how wide and narrow field neurons map onto the neurons presynaptic to Pitx2 cells.

In many places of the paper the authors speak of “putative” presynaptic neurons, also reflecting that key aspects of the circuitry are unclear. These unknowns are also present in the schematic in Fig. 1j, which depicts a lot of connections, but does not convey a clear picture. In addition, there are very unclear symbols, e.g. the dots in the connection from the upper layer retino-recipient cells to Pitx2 cells, or the pink dots in the deep layers.

We agree with the point raised by the Reviewer that while we can identify motor neurons from their electrophysiological responses during foraging; with electrophysiology alone we cannot be certain of pre-motor neurons’ tuning, hence the use of the appellation “putative”. Throughout the manuscript we have used two different methods to determine whether a neuron is a “motor neuron”, we used either their molecular identity for *in vitro* recordings or their tuning properties for the *in vivo* ones. This could have contributed to the perception of detachment between the *ex vivo* and *in vivo* sets of experiments.

We believe the results presented in Figure 1 identify, with unprecedented level of specificity, the proportion of premotor neurons (genetically-defined as Pitx2-cre^{ON-PRE}) that receive direct retinal input and their distribution across layers. However, as rightly pointed by the reviewer, the manuscript would immensely benefit from characterising the tuning properties of these unambiguously genetically-defined premotor unit *in vivo*. In order to do this and assess to what extent this population of premotor neurons overlap with known collicular cellular types, we performed 3 new sets of experiments.

1. In order to assess the visual tuning properties of genetically-defined premotor neurons in superficial layers of the SC *in vivo* we used a viral strategy to express the calcium indicator GCamp6f in Pitx2-Cre^{ON-PRE} neurons. We first injected an AAV encoding a TVA receptor protein in a CRE-dependent manner together with an AAV containing FLP-dependent GCamp6f in Pitx2-Cre::td-Tomato mice. 3 weeks later, we injected a self-inactivating G-deleted rabies virus (SiR) containing the FLP recombinase and performed a cranial window preparation exposing the superficial layer of the SC (Figure 2o). This method affords high throughput identification of the visual properties of unambiguously defined premotor neurons (i.e. presynaptic to Pitx2^{ON} neurons) across the colliculus. Mice were exposed to full field moving gratings and to drifting Gabor patches. We found similar results to those obtained using tetrode recordings,

with 42.2% of neurons recorded tuned to localised drifting Gabor patches and 33.8% to full field gratings. These results are now included in Figure 2o-r and Extended Data Figure 6. The majority of neurons displayed direction selectivity (Extended Data Figure 6k) and were organised in functional clusters of similar orientation and direction (Extended Data Figure 6l,m) as previously described in WT⁷⁻⁹. This organisation further supports the columnar organisation of the SC we propose in Figure 4. These results, together with those encompassed in next point are now discussed in the main text and corresponding Materials and Methods sections (see below).

2. We also replicated these results with a similar strategy but employing instead an intersectional strategy using aCRE^{OFF}/FLP^{ON}-Chr2 construct in order to optotag Pitx2-Cre^{ON-PRE} (but not Pitx2^{ON} neurons) in freely moving conditions. Following SiR-Flp injection we implanted an optrode consisting of a bundle of 4 tetrodes and one fibre optic. Thirteen mice were recorded while foraging in an open field arena to determine the head-rotation tuning of Pitx2-Cre^{ON-PRE} cells and then were exposed to moving gratings and Gabor patches in head restrained conditions (Figure 2o-q and Extended Data Figure 6d-f,i,k). Results were comparable to those obtained using calcium imaging and those obtained in WT mice. We also recorded Pitx2-ChR2 mice to confirm the existence of an anti-alignment in genetically defined motor neurons (Extended Data Figure 6a-c).

Line 227-238: “To confirm whether premotor neurons encode visual kinetic features, we leveraged two newly developed tools - intersectional AAVs¹⁰ and self-inactivating rabies (SiR)¹¹⁻¹³ - that allowed us to selectively record from genetically-defined Pitx2^{ON-PRE} premotor neurons. We performed an initial injection with a mix of an AAV with a flexed TVA receptor protein together with either an AAV containing FLP-dependent GCamp6f, or an AAV with CRE^{OFF}/FLP^{ON}-Chr2 in Pitx2-Cre::tdTomato mice, followed by a second injection with an SiR expressing the FLP recombinase (Fig. 2o). We recorded Pitx2^{ON-PRE} using either optetrodes, allowing us to also record the same neurons in freely moving mice, or under a 2-photon microscope yielding higher numbers of recordings to assess their visual tuning properties in superficial and upper intermediate layers. In these unambiguously defined premotor units, we found the same preservation of tuning to kinetic features across layers of the SC as previously found in WT mice (Fig. 2p-r, Extended Data Fig. 6).”

Line 328-329: “The same anti-alignment was found in genetically-defined Pitx2^{ON} motor neurons (Extended Data Fig. 6a-c).”

Line 1013-1023 (M&Ms > Surgery): “For selective long-term labelling of Pitx2^{ON-PRE} neurons in the SC, we performed an initial injection with a mix of AAV(2)-hSyn1-FLEX-nucHA-2A-TVA-2A-G(N2c) (titre: 1.5×10^{12} genomic copies/ml) and either AAV(1)-pAAV-nEF-CRE^{OFF}/FLP^{ON}-Chr2(ET/TC)-EYFP (Addgene 137141, titre: 4.3×10^{12} genomic copies/ml) or AAV(1)-Ef1a-fDIO-GCaMP6f (Addgene 128315, titre: 5×10^{12} genomic copies/ml). Following 3 weeks of expression, a second injection with a self-inactivating Rabies¹¹⁻¹³ SiR-N2c-FLP (EnvA) (titer: titre: 1×10^7 genomic copies/ml) was performed before implanting an optetrode or a cranial window for chronic Ca²⁺ imaging. In a subset of experiments AAV(9)-EF1a-double floxed-hChR2(H134R)-mCherry-WPRE-HGHpA (Addgene 20297, titre: 5×10^{12} genomic copies/ml) was injected in the SC followed by optetrode implant.”

Line 1139-1148 (M&Ms > Electrophysiology > Optetrodes): “For simultaneous single unit recording and optogenetic stimulation, a single optic fibre (core = 100 μm , NA = 0.22; Doric Lenses, Québec, Canada) was inserted between each bundle of tetrodes during production as previously described¹⁴. Light was delivered using a 473 nm laser diode module (Cobolt 06-MLD, Cobolt, Solna, Sweden) coupled to a 100 μm multimode fiber (NA = 0.22) through a Schäfter + Kirchoff fiber coupler (Cobolt, Solna, Sweden). The laser power employed in all stimulation experiments was 3-5 mW at the tip of the fiber. For optotagging, mice were probed in an open field arena. Following 10 minutes of acclimatisation, bursts of 30 5 ms-long pulses at 30 Hz were delivered with a 9 seconds rest period in between burst for a total of 900 stimulations over 5 minutes. Blue light-activated units were defined on the basis of the latency of the response to a pulse of light within a time window of 5 ms¹⁵.”

Line 1047-1057 (M&Ms > Surgery): “For *in vivo* chronic selective Ca²⁺ imaging of Pitx2^{ON-PRE} neurons in the SC recordings, mice underwent an initial viral surgical injection with an AAV-TVA and an AAV-GCaMP6f as described above and were injected with Dexafort at 2 $\mu\text{g/g}$ the day prior to surgery. A headpost and a cranial window over the SC were implanted as previously described⁹. Briefly, following isoflurane anaesthesia, Vetergesic was injected subcutaneously at 0.1 mg/kg and a metal head-post was affixed to the skull with Crown & Bridge Metabond. Epiviscaine was splashed over the skull, and a 3 mm diameter craniotomy was performed on the left hemisphere, centred on the rostral superior colliculus. The surface of the superior colliculus was exposed through removal of the overlying cortex and the SiR was then injected as described above. A 3 mm cannular window was then fixed on top of the colliculus using dental cement (Crown & Bridge Metabond).”

Line 1156-1169 (M&Ms > Electrophysiology > Ca²⁺ imaging): “Mice were imaged from one week after surgery with a two-photon microscope (Bergamo II, Thorlabs), equipped with a 16 x – 0.8 NA objective (Nikon). Mice were recorded awake and headfixed on a custom-made floating platform. td-Tomato positive Pitx2^{ON} neurons and SiR infected cells were excited with a Ti:Sapphire laser at 1030 nm and 920nm, respectively, with a power of around 20 mW (Mai TaiDeepSee, Spectra Physics). Red and green emitted fluorescence were collected through a 607 \pm 35 nm and 525 \pm 25 nm filters, respectively (Brightline). For imaging of neuronal responses, recordings consisted of either multiple planes at different depths imaged quasi-simultaneously using a piezo device, or a single plane. The pixel resolution was kept at around 0.8 μm per pixel, while the number of pixels, field of view and imaging rate were adjusted to cover the labelled cells. On average this resulted in imaging frame rates of 30 to 60 Hz and pixel dwell times of 0.2 to 0.08 μs . Two-photon recordings were then registered and ROIs were determined manually and extracted using CalmAn¹⁶ (Flatiron Institute) in Python. Variation of fluorescent values over baseline (DF/F) were computed and used for further analysis.”

FIGURE 2

Figure 2. Premotor neurons and neurons in motor layers respond preferentially to visual flow.

[...] **o**, Diagram of the viral strategy to selectively record $\text{Pitx2}^{\text{ON-PRE}}$ neurons using Self-Inactivating Rabies virus expressing FLP recombinase in $\text{Pitx2}\text{-cre}$ mice and either a combinatorial $\text{Cre}^{\text{OFF}}/\text{Flp}^{\text{ON}}$ virus expressing ChR2 or FLP-dependent GCaMP6f for opto-tagging or 2-photon Ca^{2+} imaging respectively. Peri-stimulus time histogram of 900 5 ms-long laser stimulations (ii, bottom) for an opto-tagged neuron and example raster plot of the 1st 300 stimulations are shown (ii, top). An example image of two $\text{Pitx2}^{\text{ON-PRE}}$ neurons expressing GCaMP6f is also shown (iii, bar: 100 μm). **p** Example responses of two $\text{Pitx2}^{\text{ON-PRE}}$ neurons to moving gratings, recorded either with optotetrodes (top) of 2-photon Ca^{2+} imaging (bottom). **q** and **r** Distribution of $\text{Pitx2}^{\text{ON-PRE}}$ neurons tuned to moving gratings across layers of the SC recorded with optotetrodes (q, N = 10 mice, n = 17 $\text{Pitx2}^{\text{ON-PRE}}$ neurons) or Ca^{2+} imaging (r, N = 7 mice, n = 74 $\text{Pitx2}^{\text{ON-PRE}}$ neurons).

EXTENDED DATA FIGURE 6

Extended Data Figure 6. Visual and motor tuning of motor Pitx2^{ON} and premotor Pitx2^{ON-PRE} neurons in the SC.

a, Example opto-tagged Pitx2^{ON} motor neurons. Peri-stimulus time histogram of 900 5 ms-long laser stimulations (right, bottom) and example raster plot of the 1st 300 stimulations are shown (right, top). **b**, Example STA of angular head displacements in yaw, pitch and roll for a Pitx2^{ON} motor neuron (i). The 3D STA was projected onto the 2D plane of the visual stimulation screen to draw the tuned direction of gaze. The decomposed and average gaze direction is shown in ii. **c**, Tuning to moving gratings of the neuron in b. **d**, Pitx2^{ON-PRE} neurons were opto-tagged using a combinatorial Cre^{OFF}/Flp^{ON} virus expressing ChR2 and a Self-Inactivating Rabies virus expressing FLP recombinase in Pitx2-cre mice. A first injection was performed in Pitx2-Cre mice consisting of two AAVs: one carrying a floxed copy of the EnVA receptor TVA and another promoting the expression of ChR2 in the presence of FLP recombinase but not CRE recombinase. A second injection was performed 3 weeks later consisting of an EnVA Self-Inactivating Rabies (SiR) virus expressing Flp recombinase. In this experiment Pitx2^{ON-PRE} neurons would express Flp (carried by the rabies virus) but not Cre recombinase, and as a result be the only neurons expressing ChR2. An optrode was implanted at the time of the second injection, allowing for the identification of Pitx2^{ON-PRE} neurons by opto-tagging. **e**, Example Pitx2^{ON-PRE} neuron displaying tuning to moving gratings (i) and to drifting Gabor patches (ii). **f**, Overlap of tuning to all types of visual stimuli tested in premotor neurons and their tuning to head rotations (n = 17 Pitx2^{ON-PRE} neurons recorded across 10 mice). **g**, A similar strategy was used to express GCaMP6f in Pitx2^{ON-PRE} neurons by injecting an AAV leading to FLP recombinant-dependent expression of GCaMP6f. Following the infection with the SiR, the cortex above the injection site was absorbed and a chronic cranial window for Ca²⁺ imaging was performed. **h**, Example Pitx2^{ON-PRE} neuron displaying tuning to moving gratings (i) and to drifting Gabor patches (ii). The average $\Delta F/F$ for gratings moving in all 12 directions is shown in red (i, left, shaded areas corresponds to \pm s.e.m.). The average responses to gratings of two opposing directions (black) and individual trial responses (grey) are also shown. **i** and **j**, Distribution of tuned Pitx2^{ON-PRE} across layers of the SC recorded using optotetrodes (i, N = 10 mice, n = 17 Pitx2^{ON-PRE} neurons) or Ca²⁺ imaging (i, N = 7 mice, n = 74 Pitx2^{ON-PRE} neurons). **k**, proportion of direction and orientation selective Pitx2^{ON-PRE} neurons. **l**, Absolute value of the difference in preferred angle of direction of motion of visual stimulus (as measured using gratings or Gabor patches) plotted against the 3D distance in the SC. The number of pairs is shown in parenthesis. DS/OS correspond to pairs of either OS or DS neurons compared in orientation space (180°, left), while DS pairs only consider DS neurons in direction space (360°, right). For OS/DS neurons (N = 29 cells, n = 4 mice), orange line corresponds to the median of the population, lower and upper box boundaries correspond to the 25th and 75th percentile of the population, respectively and notches correspond to the 95% confidence interval of the median calculated by bootstrapping (10000 iterations). For DS pairs (N = 20 cells, n = 3 mice), orange line corresponds to the median of the population and lower and upper box boundaries correspond to the 25th and 75th percentile of the population, respectively. **m**, All direction and orientation tuned cells to full field gratings or local Gabor patches within a volume in the SC are plotted as spheres and colour-coded according to preferred motion axis (i). A, anterior; M, medial; S, superficial. The same volume with only DS neurons colour-coded indicating their preferred direction of motion is also shown (ii).

The combination of SiR viral technology, intersectional gene delivery, electrophysiology and behaviour is a particularly powerful strategy and, to our knowledge, this is the first account of long-term functional recording of neurons presynaptic to a genetically defined neuronal population.

3. As suggested by the Reviewer we have investigated how wide and narrow field neurons map onto the neurons presynaptic to Pitx2 cells. CRE lines for at least subpopulations of both narrow field (Grp-KH288 cre) and wide field (Ntsr1-GN209 cre) neurons have previously been identified^{17,18}. However, these lines and Pitx2-Cre mice rely on the same CRE recombinase

system and cannot be used in combination to assess the overlap of Pitx2^{ON-PRE} and Grp^{ON} or Ntsr1^{ON} neurons. As an alternative, we have performed single cell in situ hybridisation on Pitx2^{ON-PRE} to test their expression of Grp and Ntsr1. This strategy, albeit the best available option, would only identify neurons actively transcribing these genes instead of all neurons that have expressed these genes at some point in the animal's life. We found that in the adult SC, Grp and Ntsr1 expressed very scarcely. In fact, it has been shown that positive neurons in adult Ntsr1-GN209 cre mice do not express Ntsr1 but selectively express Npnt instead¹⁹. Thus, we tested for expression of Npnt in Pitx2-cre^{ON-PRE} as well as expression of Cbln4, a gene identified as marking another subpopulation of narrow field neurons²⁰ (Extended Data Figure 1 n-p). We found a significant proportion of Pitx2-cre^{ON-PRE} co-localised with Cbln4. To provide further information on the characteristics of these neurons, we have also included a table with the passive membrane properties and spike properties of the neurons included in Figure 1 (Supplementary table 1) and in Figure 2 (Supplementary table 2). Supplementary table 2 also provides the tuft measurements for the subset of neurons reconstructed and included in Figure 2.

We have now briefly included these results in the main text:

Line 172-177: "Some also had morphological and physiological properties consistent with narrow-field neurons, which have been proposed to be direction and orientation selective (DS, OS), involved in orienting behaviours¹⁷, and to project vertically to intermediate and deep layers of the SC¹⁸ (Extended Data Fig. 2a, Supplementary Table 1 and 2). Indeed, we found that Pitx2^{ON-PRE} partially overlap with a genetically-defined population of narrow-field neurons (Extended Data Fig. 1n-p)."

EXTENDED DATA FIGURE 1

Extended Data Figure 1. Characteristics of Pitx2^{ON-PRE} and Pitx2^{ON} neurons.

[...] **j**, Decay time constant of excitatory postsynaptic activity before and after GABAR block for neurons receiving direct (left, one-tailed paired t-test $p = 0.0979$) and indirect (right, one-tailed paired t-test $p = 0.0214$) retinal input. **k**, Resting membrane potential before and after GABAR block (one-tailed paired t-test $p = 0.0260$) **l**, Reconstruction obtained from 10 of the neurons included in Fig 1. **m**, Example Pitx2^{ON-PRE} narrow-field neurons traced with G-deleted Rabies virus. **n**, In situ hybridization was performed on SC slices containing Pitx2^{ON-PRE} neurons traced with G-deleted rabies using RNAscope. **o**, Example staining of markers of narrow-field (Grp and Cbln4) and wide-field (Ntsr1 and Npnt). **p**, quantification of Npnt⁺ and Cbln4⁺ Pitx2^{ON-PRE} neurons.

	N	RMP (Vm)	RI (MOhm)	Sag (mV)		
Direct	13	-47.3 +- 0.9	543 +- 28	31.2 +- 5.9		
Poly.	13	-50.2 +- 0.9	627 +- 24	136.6 +- 17.3		
RAMP	10	-52.6 +- 1.2	845 +- 90	39.7 +- 7.2		
No activation	11	-43.8 +- 0.9	599 +- 24	16.4 +- 1.2		
	AP threshold (mV)	AP amplitude (mV)	AP width (ms)	AHP Amplitude (mV)	AHP width (ms)	
Direct	-37.7 +- 0.4	55.7 + 0.8	1.0 +- 0.0	13.3 +- 0.5	20.5 +- 1.8	
Poly.	-32.4 +- 0.5	43.7 +- 1.1	1.4 +- 0.0	15.5 +- 0.4	52.5 +- 2.5	
RAMP	-37.0 +- 0.8	55.7 +- 0.8	1.3 +- 0.0	15.3 +- 0.8	26.9 +- 3.2	
No activation	-38.4 +- 0.6	61 +- 1.1	0.9 +- 0.0	15.8 +- 0.6	14.3 +- 1.4	
+ 100 pA	Num of AP	Frequency of burst (Hz)	Freq. Adaptation (%)	Freq. spikes 1 to 3 (Hz)	Delay to 1 st AP (ms)	Time after last AP (ms)
Direct	19.3 +- 1.2	38.3 +- 1.5	183.2 +- 7.2	51.6 +- 1.9	24.8 +- 1.4	206 +- 24
Poly.	13.4 +- 1.2	18.2 +- 1.5	245.4 +- 45.7	26.2 +- 2.2	24.7 +- 2.8	332 +- 30
RAMP	18.4 +- 2.2	40.2 +- 2.8	172.1 +- 8.1	57.3 +- 5.1	43.2 +- 7.1	285 +- 35
No activation	33.7 +- 1.4	50.1 +- 1.2	201.2 +- 6.2	74.1 +- 1.8	22.2 +- 2	123.2 +- 23
+ 200 pA	Num of AP	Frequency of burst (Hz)	Freq. Adaptation (%)	Freq. spikes 1 to 3 (Hz)	Delay to 1 st AP (ms)	Time after last AP (ms)
Direct	3.8 +- 0.2	62.7 +- 4.2	103.1 +- 7.3	73.7 +- 4.8	3.9 +- 0.4	702 +- 16
Poly.	6.8 +- 1.4	30.7 +- 3.9	72.2 +- 10.8	40.9 +- 5.2	6.6 +- 0.5	648 +- 25
RAMP	9.7 +- 2.1	42.1 +- 5.3	82.1 +- 9.1	49.4 +- 6.4	11.6 +- 1.3	672 +- 27
No activation	14.1 +- 1.8	102.4 +- 2.3	173.8 +- 15.7	135.4 +- 5.1	4.1 +- 0.5	647 +- 22

Supplementary Table 1. Electrophysiological properties of neurons recorded in acute brain slices and included in Figure 1

	N	RMP (Vm)	RI (MOhm)	Sag (mV)	Tuft width (um)	Tuft length (um)
Superficial	24	-50.1 +- 3.4	282 +- 44	28.2 +- 17	277 +- 160 (N =2)	468 +- 3 (N =2)
Upper Intermediate	12	-41.9 +- 3.5	297 +- 97	3.8 +- 1.6	362 +- 49 (N =12)	401 +- 44 (N =12)
Lower Intermediate/ Deep	9	-42.2 +- 3.4	164 +- 25	2.3 +- 1.0	379 +- 174 (N =3)	452 +- 241 (N =3)

	AP threshold (mV)	AP amplitude (mV)	AP width (ms)	AHP Amplitude (mV)	AHP width (ms)
Superficial	-28.8 +- 2.0	21.5 +- 2.9	0.7 +- 0.1	7.3 +- 1.2	31.0 +- 7.0
Upper Intermediate	-26.0 +- 3.7	14.9 +- 2.4	0.7 +- 0.1	5.5 +- 0.9	26.5 +- 6.0
Lower Intermediate/Deep	-31.3+- 2.3	11.4 +- 0.8	0.6 +- 0.0	6.0 +- 0.7	41.0 +- 14.9

+ 100 pA	Num of AP	Frequency of burst (Hz)	Freq. Adaptation (%)	Freq. spikes 1 to 3 (Hz)	Delay to 1 st AP (ms)	Time after last AP (ms)
Superficial	8.9 +- 2.4	11.4 +- 3.2	150.5 +- 63.0	23.7 +- 7.2	198.6 +- 59.6	49.9 +- 13.9
Upper Intermediate	13.4 +- 1.2	18.2 +- 1.5	245.4 +- 45.7	26.2 +- 2.2	24.7 +- 2.8	332 +- 30
Lower Intermediate/Deep	18.4 +- 2.2	40.2 +- 2.8	172.1 +- 8.1	57.3 +- 5.1	43.2 +- 7.1	285 +- 35

+ 200 pA	Num of AP	Frequency of burst (Hz)	Freq. Adaptation (%)	Freq. spikes 1 to 3 (Hz)	Delay to 1 st AP (ms)	Time after last AP (ms)
Superficial	19.8 +- 8	33.5 +- 12.1	238.0 +- 91.6	54.2 +- 17.9	3.9 +- 0.4	702 +- 16
Upper Intermediate	6.8 +- 1.4	30.7 +- 3.9	72.2 +- 10.8	40.9 +- 5.2	6.6 +- 0.5	648 +- 25
Lower Intermediate/Deep	9.7 +- 2.1	42.1 +- 5.3	82.1 +- 9.1	49.4 +- 6.4	11.6 +- 1.3	672 +- 27

Supplementary Table 2. Electrophysiological and morphological properties of neurons recorded in whole-cell mode *in vivo* and included in Figure 2

We have now also simplified the diagram in Figure 1j as per the reviewer's suggestion.

FIGURE 1

Figure 1. Two functionally distinct premotor populations channel direct and indirect retinal input onto collicular motoneurons.

[...] j, Diagram of the intra-collicular visuo-motor network. Orange: pre-motor neurons with monosynaptic retinal input; blue: pre-motor neurons showing di-synaptic retinal excitation. Grey: Pitx2^{ON-PRE} neurons without visual responses. Percentage of each type were estimated from c and g and are shown at the bottom right.

Minor:

4. Fig. 1b is very low res, some arrowheads point into the void.

We have exchanged the image for a higher resolution and contrast one with two up-close boxes exemplifying traced neurons in superficial and intermediate layers.

FIGURE 1

Figure 1. Two functionally distinct premotor populations channel direct and indirect retinal input onto collicular motorneurons.

[...] **b**, Immunostaining of Pitx2^{ON} neurons (mCherry) and their rabies-traced presynaptic partners (GFP, bar: 100 μ m). **Expanded fields containing example Pitx2^{ON-PRE} neurons in superficial (top) and intermediate layers (bottom) are shown.**

5. Line 130: Chr2 was not injected, rather a virus.

We have now rephrased this. Line 139: "An AAV expressing Chr2 was injected [...]".

6. Line 159: wide and narrow field neurons are mentioned here, but they are not explained, and no reference is given.

This particular reference to narrow and wide field neurons has been deleted and a brief description of narrow field neurons has been included instead and their implications in the circuit. Line 173: "[...] narrow-field neurons, which have been proposed to be direction and orientation selective (DS, OS), involved in orienting behaviours¹⁷, and to project vertically to intermediate and deep layers of the SC¹⁸"

7. Line 162: I am not particularly happy with the term "static receptive field". At the very least this should be properly defined.

We agree with the reviewer that this term does not fully encompass the nature and diversity of the RFs and we have taken on board this comment and their previous suggestion to modify the way we refer to RFs across the whole manuscript. We now consider any tuning of neurons to visual stimuli as a RF and we specify whether this is spatial (elicited by the appearance of stimuli only at particular location in the visual space), static (owing to the static nature of the stimuli) or kinetic (if the visual stimuli used is either moving full field grating, drifting Gabor patch or moving spots). This would leave four possible combinations of RFs: spatial static (ssRF, as measured with flashing squares), static (as measured with static gratings), spatial kinetic (as measured with Gabor patches) or kinetic (as measured with moving gratings). Keeping in mind this new definition and to help readers through the manuscript we have only used an acronym for ssRFs as this type of RF most closely recapitulates the canonical classic model of static alignment. For all other tunings we have attached the corresponding adjective (spatial/localised, kinetic or static) without providing an acronym. ssRFs are briefly introduced

in the introduction. Line 65: “static spatial receptive fields (ssRFs)” and further described in Line 170: “well-defined spatially confined static receptive fields (ssRFs) as measured with localised flashing black or white squares”.

8. Lines 164-166: this is very speculative. How do they know these are narrow-field neurons?

The reviewer is correct in that we can only speculate about which type of neurons could be presynaptic to those recorded, hence the use of “putative”. However, we believe there is a basis to this assumption considering that narrow-field neurons are DS and are thought to project to intermediate and deep layers of the SC¹⁷. Moreover we have recorded narrow-field neurons (classified due to their visual responses and morphology) with spiking activity that would be consistent with them being the driving force behind the EPSPs recorded in deeper layers (Figure 2). We have now rephrased this sentence to be less speculative. Line 183: “The greater latency and sharp subthreshold tuning in intermediate and deep layers could reflect synaptic input from the putative presynaptic narrow field neurons recorded in the superficial layers”.

9. Line 170: I am not sure what to make of the lower responses to static gratings. Most cells will show only very small responses to static stimuli, and tuning curves based on such standing grating might not be very meaningful. Were these responses measured before and after the grating moved?

These responses were recorded before and after gratings movement and averaged across both conditions (see Materials and Methods references below). Although the fact that static stimuli would elicit smaller responses than dynamic ones might seem obvious to some readers, we feel that it is appropriate to report it in the manuscript as we aim to explicitly highlight the relevance of kinetic visual features in driving motion during behaviour and static gratings represents the closest static control of moving gratings. Line 1192: “Each grating (spatial frequency: 0.08 cycles per degree) would first remain static for 1 second, then move at 2.83 Hz for 2 seconds and stop for another further second before changing direction.” and Line 1327: “Static gratings’ response was averaged before and after drifting and probed only in orientation space”.

10. Line 173: would be nice to briefly state here how recordings in head-fixed, running mice were done

We have now briefly introduced the method as suggested. Line 193: “[...] while concurrently monitoring running bouts on a wheel equipped with an accelerometer and eye movements”.

11. Line 174: do they mean “tuned to” or “responsive to”?

Neurons responding either positively or negatively to all grating directions were not included, only either Direction Selective or Orientation Selective. We have rephrased the sentence to be more explicit. Line 194: “The overall proportion of OS and DS neurons [...]”.

12. Line 178: unclear what the “1.9x” refers to.

This has been rephrased to Line 196: “[...] visually tuned neurons in the superficial layers were more sharply tuned (Fig. 2k) and had a 1.9 fold increase in firing rate at their preferred

direction compared to visually-tuned neurons of the lower intermediate and deep layers (Fig. 2l, Extended Data Fig. 3l).”

13. Line 185 ff.: Not really clear why exactly these receptive field properties were measured, this seems somewhat arbitrary. Why were directional looming stimuli used, and not expanding ones at different visual field positions? Is there any relevance to the observation of oscillatory activity following changes in luminance? These (and other) data reported are not really relevant for the argument made.

The rationale behind using these visual stimuli was to provide examples of full field static and localised and directional types of stimuli. One could have argued that the higher response to gratings was simply because a higher portion of the visual field was stimulated, to control for that we used static full field gratings and changes in luminescence. To provide further evidence that the direction of movement is a feature notably encoded in the SC, we had included the looming stimuli, which mimic ethological threats incoming from different directions.

We have taken on board the comments and concerns of all reviewers and we have now performed new recordings with new visual stimuli including localised drifting Gabor patches as previously described and a small directionally moving spot (covering $\sim 3.7^\circ$ of the mouse's visual field). Although the stimulation paradigm with the small moving spot is very similar to the looming stimuli, it is aimed at mimicking a potential target instead of a threat. We found that 21% of neurons recorded across layers of the SC responded strongly to small moving dots that could mimic an ethologically relevant moving target (Extended Figure 5d-f and h). This is quite significant considering the spots only crossed 18% of the screen throughout the full stimulation protocol. Although neurons that also had a localised RF as measured using drifting Gabor patches responded preferentially to the moving spots crossing said RF in superficial layers (Extended Data Figure 5e), we also found neurons tuned to particular directions or orientation of sweeping spots not displaying localised RF, particularly in deeper layers of the SC (Extended Data Figure 5f). Moreover, neurons tuned to small moving targets and full field gratings have similar preferred angle of tuning (angular difference average: 30 degrees and median: 20 degrees, Extended Figure 5k), supporting the relevance of the visuomotor alignment uncovered for target interception.

Given the reviewer's input, and the limited space for publication as well as the amount of extra experiments and analysis included after revisions, we have taken the looming data out of the manuscript.

Line 212-218: “SC neurons were also tuned to target-mimicking spots moving towards and away from the centre of gaze. Although the spots only crossed 18% of the screen throughout the full stimulation protocol, 21% of all collicular neurons recorded showed a directional response to moving spots. While 75% of these responses in superficial layers reflected the activation of the neuron's spatial receptive field, this decreased to 33% in lower intermediate and deep layers (Extended Data Fig. 5j).”

Line 222-226: “Importantly, we found a conservation of the preferred tuning direction among stimulation paradigms, meaning that neurons tuned to a particular direction of gratings, would be tuned to moving spots and drifting Gabor patches of the same direction (average preferred $^\circ$ gratings- $^\circ$ spots: 30° , $^\circ$ gratings- $^\circ$ Gabor: 27° , Extended Data Fig. 5k).”

Line 1198-1206 (M&Ms > Behaviour > Visual stimulation): “A small black spot (1.3 cm diameter corresponding to $\sim 3.7^\circ$) moving at 30 cm per second in a grey background was used in order to assess the direction selectivity of neurons to a stimulus mimicking a prey moving. The stimulation paradigm consisted of a small black spot moving towards the centre of the screen (roughly aligned to the centre of the visual field) for 1 second, then staying static in the centre of the screen for 0.5 seconds before retracting to the opposite direction with the same speed of approach. 8 starting points were used: all 4 corners of the screen and all midpoints between two corners of the screen. Each starting location was presented 3 times per trial in a semi-randomised manner. Similarly to the other visual stimulation paradigms, we recorded 3 trials for a total of 9 presentations per movement direction.”

Line 1338-1339 (M&Ms > Quantification and Statistical Analysis > visual tuning): “Direction and orientation selectivity for small moving spots and for Gabor patches was measured as described for gratings”.

EXTENDED DATA FIGURE 5

Extended Data Figure 5. Mapping of visual tuning properties in response to drifting Gabor patches, small moving spots and changes in luminescence in the SC.

[...] **d**, Mice were also presented with a small spot (1.3 cm diameter) starting at 8 locations along the corners and border of the screen (red dots) and moving in 8 directions towards and away from the mouse’s centre of gaze on the screen (red cross). **e**, Example tuning of a neuron responding to a small spot moving in a particular orientation. Shaded areas in polar plots indicates \pm s.e.m. Average firing rate for each direction is shown as a heatmap and the average time change across directions to identify if a neuron prefers approach vs retreat is shown at the bottom (i). The tuning of this neuron corresponds to the spot crossing its spatial receptive field mapped using drifting Gabor patches (ii). **f**, Example tuning to a small moving spot for a neuron without spatial receptive field. **g**, Example neuron tuned to full field moving gratings (i), Gabor patches (ii) and moving spot (iii). **h**, Distribution of neurons responding to particular locations in space mapped using Gabor patches (spatial Gabor, blue), neurons showing direction or orientation selectivity mapped using Gabor patches (Gabor OS/DS, green) or full field gratings (red), neurons tuned to moving spots (spot OS/DS, black) or tuned to head rotations (head

tuned, purple). The number of recorded neurons per depth of recording, indicated by the shading in the SC diagram, is shown in parenthesis (N = 4 mice). **i**, Distribution of neurons tuned to particular locations (spatial Gabor, blue), direction or orientation selective neurons (Gabor OS/DS, green) or both (grey) among all neurons tuned to drifting Gabor patches. **j**, Overlap of the tunings shown in **h** for individual neurons, computed using Intervene. Static features are preferentially represented at the top, while kinetic ones are preferentially represented towards the bottom. Note that the number of neurons is represented by the shade of colour and not by the surface of the intersection. **k**, Angular difference between the preferred angle of tuning measured using full field gratings and moving spots (black) or Gabor patches (green).

14. Line 194: what are “deep intermediate layers”? In general, the designation of the SC layers is not very stringent throughout.

We have gone through the whole manuscript in an attempt to homogenise the references to SC layers. For all main figures we have divided the SC in three rough areas that correspond to:

- the stratum zonale (SZ) and stratum griseum superficiale (SGS), referred in the manuscript as “superficial layers”
- the stratum opticum (SO) and upper stratum griseum intermedium (uSGI), referred as “upper intermediate layers”
- lower SGI (lSGI) and SG profundum (SGP), referred as “lower intermediate and deep layers”

The reasons to divide the SC in these 3 groups is that 1) ex vivo experiments (Figure 1) showed that retinal ganglion cells project almost exclusively to SZ and SGS and 2) Pitx2^{ON} are located within the lSGI, we wanted to separate uSGI and lSGI for analysis. This division has been clearly stated in the main text and Figure 1 legend and has been respected throughout the manuscript. Line 101 : “Pitx2^{ON-PRE} neurons were pooled according to their distribution across layers within 3 main domains: 1) stratum zonale (SZ) and stratum griseum superficiale (SGS), referred hereafter as SZ/SGS or superficial layers, 2) the stratum opticum (SO) and upper stratum griseum intermedium (uSGI), referred henceforth as SO/uSGI or upper intermediate layers, and 3) lower SGI (lSGI) and SG profundum (SGP), referred as lSGI/SGP or lower intermediate and deep layers.”

15. Line 202: the papers cited here are classic ones, but there are newer ones that go beyond these basic findings.

We have added several more recent references. Line 244: “stimulus spatial receptive field and movement end-points²¹⁻²⁶.”

16. Fig. 2f: It is not made clear how the latencies are determined. They are very long, even in the superficial layers, (125 ms, mean or median?; what are the red crosses?), in V1 they are around 45 ms.

We have now included a brief sentence in Materials and Methods regarding the measurement of latencies. Line 1264-1270 (M&Ms > Quantification and Statistical Analysis > Whole-cell in vitro and in vivo electrophysiology): “To determine the latency of response to visual stimuli recorded in whole-cell mode *in vivo* we first calculated the differential of the membrane potential during baseline (grey screen) to assess the SD of baseline presynaptic activity. For each

repetition at the preferred direction or orientation of tuning we determined the time of the first event with a slope >3 SDs of baseline presynaptic activity and of at least 0.5 mV/ms. Recordings with a SD of baseline presynaptic activity > 1.5 mV were discarded."

The median is shown as a bar in the box plot while the mean (reported in the main text) is indicated by the red cross. We have explicitly written this information in the figure legend: "Box plot indicates minimum, 1st quartile, median, 3rd quartile and maximum. Red cross indicates mean."

We believe the reason the latencies appear longer than expected is due to anaesthesia. Isoflurane has been shown to reduce and slow down neuronal transmission by increasing inhibition, decreasing presynaptic release and neuronal excitability²⁷⁻²⁹. In order to test this hypothesis we recorded the responses to moving gratings across the SC in three mice before and after full isoflurane anaesthesia using tetrodes. Our project license limited the number of recording sessions allowed per mouse under anaesthesia and out of 60 neurons recorded, only 17 had significant spiking activity in anaesthetised conditions. Of those only 9 DS or OS neurons recorded in awake conditions conserved significant and consistent response to moving gratings under anaesthesia. Albeit low, this n number proved enough to test our hypothesis. These neurons, all located in the superficial layers, had an average latency of response to their preferred direction of 65 \pm 12 ms in awake conditions vs 128 \pm 15 ms under anaesthesia (Rebuttal Figure 1).

Rebuttal Figure 1: Latency to visual responses for neurons recorded in awake conditions and under isoflurane anaesthesia (n = 9 cells recorded in n = 3 mice, one-tailed paired Student's t-test $p = 0.0005$).

17. Fig. 2gi: must make clear which layers these cells are in.

These are the neurons tuned across all layers. We have now explicitly said this in the figure legend and specified the SI per layer in Extended Data Figure 2e.

EXTENDED DATA FIGURE 2

Extended Data Figure 2. Reconstruction and visual properties of neurons recorded *in vivo*.

[...] e, Distribution per layers of the SI presented in Fig. 2g.

18. Fig. 2h: how is a responsive neuron defined?

This information is available in the Materials and Methods.

Line 1318-1352 (M&Ms > Quantification and Statistical Analysis > Visual tuning): “Analysis routines for visual tuning were developed in Igor Pro (WaveMetrics Inc., Portland, OR). The neuronal response to drifting sinusoidal gratings was averaged for all trials and normalised to the baseline firing rate. The selectivity was then calculated both in direction and orientation space (360° and 180° respectively) by computing the mean orientation and direction vectors in polar coordinates, described by their modulus (corresponding to the selectivity index) and average angle.

$$SI = \left| \frac{\sum_k R(\theta_k) e^{2i\theta_k}}{\sum_k R(\theta_k)} \right|$$

$$\bar{\theta} = \text{atan} \left(\frac{\sum_k \cos \theta_k}{\sum_k \sin \theta_k} \right)$$

Where $R(\theta_k)$ is the response at each sampled direction or orientation θ_k (12 for direction space and 6 for orientation space). Static gratings were probed only in orientation space. Given that the modulus of the vectors is dependent on the firing pattern and firing rate of the neuron, the same calculation was performed for shuffled spike times to obtain a probability distribution of shuffled DSI and OSI to probe for both direction selectivity and orientation selectivity (see “generation of shuffled dataset”). To determine whether a neuron was tuned to moving or static gratings 3 parameters were used: the selectivity index (≥ 0.1), the trial-to-trial angular variance (≤ 0.8 ; used as a measure of the reliability of selectivity) and the significance of the SI compared to the shuffled distribution. If two out of the three criteria were fulfilled the neuron was considered tuned. If the criteria were met for both direction and orientation spaces, a neuron was classified as direction selective if DSI > OSI, and orientation selective if OSI > DSI.

Direction and orientation selectivity for small moving spots and for Gabor patches was measured as described for gratings. Two separate analyses of selectivity and preferred angle of tuning were performed for Gabor patches: a location-independent analysis by averaging the responses to each direction of movement across all locations and another analysis only considering the responses at the location of maximal average activity.

In order to determine the spatial tuning of neurons to static stimuli, we averaged the response (firing rate) to flashing 9 cm² squares per location in the screen (15 by 11 locations) for each colour separately (black or white) and divided it by the response during baseline conditions and obtained a 2D matrix corresponding to the increase/decrease of firing rate over baseline per location. This matrix was further transformed into Z-scores. The max Z-score was compared to those obtained performing the same analysis in 1000 shuffled trials (see “Generation of shuffled datasets”). If the maximal Z-score was >2 and ranked higher than the top 5% of those obtained from the shuffled distribution, a 2D Gaussian was the fit to the matrix. The centre of the Gaussian fit was used to determine the centre of the ssRF. The same analysis was performed to assess whether neurons had a spatial and kinetic RF as measured with Gabor patches. The centre of the location of maximal response was used as centre of the RF.”

19. Line 240: How do they know these neurons "drove" movements? Also, what is meant by contralateral head movement?

We have rephrased this sentence for clarity. Line 292: "Motor units, consistently tuned to specific head rotations in all light and dark trials, were, as a population, tuned to the expected head movement vector encoded in the area of the SC used for recordings¹⁴, which was also consistent across trials (Fig. 3a-c, Extended Data Fig. 7a-d)".

20. Line 245: what drives these upper layer neurons in the dark?

We do not have a measurable way of knowing what could be driving these neurons, these could be responding to other sensory signals such as smell coming from the droplets of condensed milk. At least a subset of these neurons appear to be inhibited by light (Extended Data Figure 7), which could explain their lack of head-rotation tuning in light trials, while still preserving their tuning in darkness. The ethological relevance of these neurons remains unclear and would be worthy of investigation in future studies. We think it is important to be forthcoming on the diversity of tuning properties across the SC; however, characterising them in depth and individually would be outside of the scope of this paper.

EXTENDED DATA FIGURE 7

Extended Data Figure 7. Distribution and characteristics of collicular neurons tuned to head rotations.

[..] **k**, Example neuron tuned to moving gratings and head displacements only in darkness. **l**, Neuron recorded in vivo in whole-cell mode displaying a similar selectivity as that in **k**. The spiking activity of this cell was inhibited by light (**iv**).

21. Line 257: eye movements in head fixed mice are very small, so I doubt whether the absence of neuronal responses is very meaningful.

Our original concern and rationale behind measuring eye movements was that head and eye movements are mostly correlated in freely moving conditions and thus, neurons tuned to head rotations could have been tuned to eye movements instead. In order to decouple this correlation, we performed the same spike-triggered analysis under head restrained conditions. We believe that if the motor neurons recorded were responsible for eye movements and not head-rotations, this should have been apparent in head restrained conditions. Although eye movements are small in head restrained conditions (~2-5 degrees) these are still measurable

and in fact we did record some neurons tuned to eye movements (Extended Data Figure 4h-k); however, these did not overlap with the head-rotation cells included in the study.

22. Line 262: an assumption that is not tested.

Motor neurons in the superior colliculus have been shown to encode head rotations independently from the initial position of the head (Masullo et al., 2018). In order to test whether this remains valid for the neurons included in this manuscript we have calculated the average head rotation at the time of spike for all neurons in Figure 3. We found this to be very close to a neutral, resting head position (2 ± 3, 1 ± 1 and 3 ± 4 degrees in yaw, pitch and roll components respectively). This data is now also included in Extended Data Figure 9e).

EXTENDED DATA FIGURE 9

23. Line 287: “rapid interception of targets” is very speculative.

We believe the full sentence “the uncovered alignment would favour the rapid interception of targets moving towards the centre of the visual field” accurately represents our hypothesis, which is described by the model in Figure 4.

24. Fig. 3b: is this in light or in darkness?

The graph presented in Figure 3b corresponds to one of 2 light trials. Similar head rotations were observed in the other 3 trials. These are now included in Extended Figure 7.

EXTENDED DATA FIGURE 7

25. Line 404: please put reference.

References as per point 15 have been included.

26. Line 410: "Primates": Authors should discuss that what they found in the mouse might be specific to animals that lack foveate vision, where orienting movements of the eyes and head might have a very different function from that for example in primates.

Our working hypothesis is that, although the specifics of the alignment might differ depending on the complexity of the motor system (having more or less degrees of freedom stemming from the number of rotation axis in question), sensori-driven orienting movements would preferentially be aligned in kinetic space. We do not believe this would be different for primates. However, we understand this is just our hypothesis and we are now explicitly mentioning in the discussion that our study was done in an afoveate model organism. Line 511: "Furthermore, we propose that the kinetic anti-alignment we have uncovered in afoveate mice could also expand to other systems and be the substrate of other ethologically relevant goal-oriented behaviours observed in the animal kingdom [...]" .

27. Line 934: "73° by 104°": add dimensions

The dimension of the screen and squares in cm has also been included. Line 1187 (M&Ms > Behaviour > Visual stimulation): "Static receptive field position was estimated by 750 ms-long flashes of uniform black or white 9 cm² squares on a grey background. The screen was divided in 165 locations covering 73° by 104° degrees (corresponding to 30 x 53 cm)."

28. Line 940: there were no colored stimuli, right?

Not beyond black and white.

29. Line 1055: related to point 1, it is very unclear which criterion was actually used then to determine whether a neuron had a receptive field

We trust this has been addressed in our responses to main point 1 and the response to minor point 18.

Referee #2 (Remarks to the Author):

The manuscript “Kinetic features dictate sensorimotor alignment in the superior colliculus” shows that neurons in the intermediate and deep layers of the superior colliculus have poor responses to static spots. Many of the neurons in these layers have activity preceding specific head-movements. The head-movements associated with the activity of individual neurons in the superior colliculus is in the opposite direction to the preferred visual flow direction of the neurons. The preferred visual motion direction is thus aligned to the direction of visual motion induced by the head movement when the neuron is active. The manuscript presents the response to moving targets in a three-layer network model that is wired in a way that is consistent with their findings in Pitx2 neurons and the literature. The motion resulting from this model would result in quickly directing the head-gaze to a target that is moving in a centripetal direction.

The main finding of the link between head-direction and visual-flow preference is novel, and the interpretation that this would benefit fast interception of centripetal moving targets is interesting. Furthermore, the paper proposes a novel behavioural benefit for the concentricity of direction selective responses in the superior colliculus.

I find the manuscript certainly interesting and it is significant for our understanding of the superior colliculus and gaze-direction. The data are well presented and the model is novel. Overall the data support the conclusion, but I am left with a number of questions, in particular about the validity of the proposed model for explaining target-intercept behaviour.

First, the paper makes predictions about the change in head-direction in response to moving stimuli, in particular that the responses to centripetal stimuli are different from those to centrifugal stimuli. How do these predictions work out in behaviour? In particular, the model suggests that a grating drifting with a centripetal motion is met with a head movement in the opposite direction. This is opposite to what I would expect from all the work on the optokinetic reflex with similar stimuli in mice and other animals, in particular the work with a rotating drum from Douglas, Prusky and other. Can the model predictions be reconciled with these earlier results on gaze-change in response to drifting gratings?

The reviewer raises an important point with regard to the apparent conflict between sensorimotor alignment strategies favourable for interception and for optokinetic and optomotor reflexes. In this regard, it's important to stress that in mammals, OKR and OMR are not mediated by the motor layers of the superior colliculus but by the accessory optic system (AOS)³⁰, which includes the nucleus of the optic tract (NOT) and the dorsal, lateral, and medial terminal nuclei (DTN, LTN, and MTN), which are ultimately responsible for the motor aspects of the OKR and OMR, with the MTN/LTN mediating vertical^{31,32} and the NOT mediating horizontal ones^{31,33}. Hence, different visuo-motor areas might obey different sensory-motor alignment rules. Our hypothesis is that the AOS system evolved to stabilise the visual field (hence its possible reliance on visuo-motor alignment) and the motor SC circuit evolved to facilitate target oriented movement (facilitated by visuo-motor anti-alignment). One interesting question is how these two systems that would lead to opposite behavioural responses are coordinated so that one prevails. Albeit undoubtedly interesting, we feel that a systematic comparison across these different visuo-motor systems would go beyond what is reasonable to cover in a single manuscript with a satisfactory level of detail and precision.

Second, direction selectivity was tested with drifting gratings and this is used as input to the model. The interpretation of the data, however, is given in the context of target interception. It would thus make sense to measure the directional selectivity to moving targets. Data to directional looming stimuli is presented in the extended data, but it is unclear how the tuning found in these recordings match the tuning on direction selectivity with gratings and whether this data supports the model. Can this analysis be added?

We agree with the reviewer that it would be valuable to add a visual stimulus that is more directly relatable with an ethologically relevant moving target. Hence, we have now included a dataset consisting of tetrode recordings performed in 4 C57BL6 mice across all layers of the superior colliculus both in freely moving conditions to map the neuron's tuning to head rotations during foraging and in head restrained conditions in response to small moving spots (1.3 cm diameter, $\sim 3.7^\circ$) in 8 different directions. As a control, we also recorded their response to full field gratings. We found that 21% of neurons recorded across layers of the SC responded strongly to small moving dots that could mimic an ethologically relevant moving target (Extended Figure 5d-f and h). This is quite significant considering the spots only crossed 18% of the screen throughout the full stimulation protocol. Although neurons that also had a localised RF as measured using drifting Gabor patches responded preferentially to the moving spots crossing said RF in superficial layers (Extended Data Figure 5e), we also found neurons tuned to particular directions or orientation of sweeping spots not displaying localised RF, particularly in deeper layers of the SC (Extended Data Figure 5f, j). Moreover, neurons tuned to small moving targets and full field gratings have similar preferred angle of tuning (angular difference average: 30 degrees and median: 20 degrees, Extended Figure 5k), supporting the relevance of the visuomotor alignment uncovered for target interception. These results are now discussed in the new version of the manuscript (see below). We are also including new sections in Materials and Methods to explain the visual stimulation paradigm and analysis.

Line 212-218: "SC neurons were also tuned to target-mimicking spots moving towards and away from the centre of gaze. Although the spots only crossed 18% of the screen throughout the full stimulation protocol, 21% of all collicular neurons recorded showed a directional response to moving spots. While 75% of these responses in superficial layers reflected the activation of the neuron's spatial receptive field, this decreased to 33% in lower intermediate and deep layers (Extended Data Fig. 5j)."

Line 222-224: "Importantly, we found a conservation of the preferred tuning direction among stimulation paradigms, meaning that neurons tuned to a particular direction of gratings, would be tuned to moving spots and drifting Gabor patches of the same direction (average preferred $^\circ$ gratings- $^\circ$ spots: 30 $^\circ$, $^\circ$ gratings- $^\circ$ Gabor: 27 $^\circ$, Extended Data Fig. 5k)."

Line 1198-1206 (M&Ms > Behaviour > Visual stimulation): "A small black spot (1.3 cm diameter corresponding to $\sim 3.7^\circ$) moving at 30 cm per second in a grey background was used in order to assess the direction selectivity of neurons to a stimulus mimicking a prey moving. The stimulation paradigm consisted of a small black spot moving towards the centre of the screen (roughly aligned to the centre of the visual field) for 1 second, then staying static in the centre of the screen for 0.5 seconds before retracting to the opposite direction with the same speed of approach. 8 starting points were used: all 4 corners of the screen and all midpoints between two corners of the screen. Each starting location was presented 3 times per trial in a semi-randomised manner. Similar to the other visual stimulation paradigms, we recorded 3 trials for a total of 9 presentations per movement direction."

Line 1338-1339 (M&Ms > Quantification and Statistical Analysis > visual tuning): “Direction and orientation selectivity for small moving spots and for Gabor patches was measured as described for gratings”.

EXTENDED DATA FIGURE 5

Extended Data Figure 5. Mapping of visual tuning properties in response to drifting Gabor patches, small moving spots and changes in luminescence in the SC.

[...] **d**, Mice were also presented with a small spot (1.3 cm diameter) starting at 8 locations along the corners and border of the screen (red dots) and moving in 8 directions towards and away from the mouse’s centre of gaze on the screen (red cross). **e**, Example tuning of a neuron responding to a small spot moving in a particular orientation. Shaded areas in polar plots indicates \pm s.e.m. Average firing rate for each direction is shown as a heat map and the average time change across directions to identify if a neuron prefers approach vs retreat is shown at the bottom (**i**). The tuning of this neuron corresponds to the spot crossing its spatial receptive field mapped using drifting Gabor patches (**ii**). **f**, Example tuning to a small moving spot for a neuron without spatial receptive field. **g**, Example neuron tuned to full field moving gratings (**i**), Gabor patches (**ii**) and moving spot (**iii**). **h**, Distribution of neurons responding to particular locations in space mapped using Gabor patches (spatial Gabor, blue), neurons showing direction or orientation selectivity mapped using Gabor patches (Gabor OS/DS, green) or full field gratings (red), neurons tuned to moving spots (spot OS/DS, black) or tuned to head rotations (head tuned, purple). The number of recorded neurons per depth of recording, indicated by the shading in the SC diagram, is shown in parenthesis ($N = 4$ mice). **i**, Distribution of neurons tuned to particular locations (spatial Gabor, blue), direction or orientation selective neurons (Gabor OS/DS, green) or both (grey) among all neurons tuned to drifting Gabor patches. **j**, Overlap of the tunings shown in **h** for individual neurons, computed using Intervene. Static features are preferentially represented at the top, while kinetic ones are preferentially represented towards the bottom. Note that the number of neurons is represented by the shade of colour and not by the surface of the intersection. **k**, Angular difference between the preferred angle of tuning measured using full field gratings and moving spots (black) or Gabor patches (green).

We found a subset of neurons tuned to small moving targets also were tuned to specific head rotations. Importantly, the visuomotor anti-alignment is conserved when using a directional small dot moving, reinforcing our findings and our ethological hypothesis. This is now included in Extended Figure 9j.

Line 328-330: “The same anti-alignment was found in genetically-defined Pitx2^{ON} motor neurons (Extended Data Fig. 6a-c) and for neurons tuned to both head rotations and moving spots (Extended Data Fig. 9j).”

EXTENDED DATA FIGURE 9

j

Extended Data Figure 9. Characterisation of the alignment of visuo-motor neurons.

[...] j, Visuo-motor alignment between the direction of gaze and preferred direction of movement of a moving spot, for neurons tuned to both head rotations and moving spots (measured as in Extended Data Figure 5) .

Third, the authors show that they cannot compute receptive fields in motor neurons using the flashed spots, and then go on to say that these neurons in the deeper layers do not have receptive fields. This seems a stronger conclusion than the presented data currently supports. Perhaps the flashed spots are not salient enough when shown in the context of a new spot appearing every 750 ms, or some local motion feature necessary is necessary to evoke a response. Do the authors think that the neurons would respond to motion anywhere on the screen, or that there still is an exclusive visual region like a receptive field in which motion triggers response. It would be nice to see this clarified, or otherwise, that the conclusion that the neurons lack a receptive field is weakened.

Furthermore, I think that the presented awake data is important for the interpretation given to the data. However, the responsiveness to the flashed spots is very low (19%) in the superficial layer. This would suggest that, in the words of the authors, neurons in the superficial layer lack receptive fields. It appears that the flashed spot stimulus is in general not an effective stimulus to drive the superior colliculus, even for the superficial layers. This weakens the conclusion about the lack of rfs in the motor layers.

In order address whether neurons in the SC could have RFs by responding to more salient moving stimuli, we performed a new set of experiments including the presentation of localised moving gratings as small Gabor patches. Although this stimulation paradigm still relies on kinetic features to drive responses in the SC, we were able to assess whether there is a spatial structure of the dynamic RFs previously mapped using moving gratings.

We performed tetrode recordings in 4 C57BL6 mice across all layers of the superior colliculus (SC), first in freely moving conditions in an open field to map neuron's tuning to gaze shifts during foraging and then under head restrain conditions to map their responses to small Gabor patches (see Materials and Methods and Extended Data Figure 5, also included below). As a control, we also recorded their response to full field gratings. We found that more neurons responded to drifting Gabor patches compared to moving gratings. The difference is particularly prominent in the superficial layers where 89% of neurons responded to Gabor patches vs 63% of recorded neurons in superficial layers responding to full field moving gratings (including direction selective DS neurons, orientation selective OS neurons and neurons with significant but direction/orientation independent overall change in

activity during moving gratings, 42% if only OS/DS neurons are considered, Extended Data Figure 5h). This was likely due to the combination of location-specific and DS/OS responses elicited using Gabor stimuli. Similarly to moving gratings, the proportion of neurons tuned to drifting Gabor patches decreased in intermediate and deep layers; however, while most neurons responding to Gabor patches in superficial layers displayed location-specific responses linked to a localised spatial visual receptive field independent of the direction of drift of the Gabor patch (71%), in lower intermediate and deep layers, OS/DS neurons were predominant, accounting for 91% of all neurons responding to Gabor patches (Extended Data Figure 5i). This result further strengthens our finding that kinetic features of the visual scene are those that are primarily transmitted to motor layers in the SC. We include a couple of example responses of neurons tuned to Gabor patches, together with their proportion across layers, combined responses with other visual stimuli and angular difference of preferred direction of movement during full-field grating stimuli and Gabor patches in Extended Data Figure 5a-c, h-k.

These results are now discussed in the main text of the manuscript:

Line 206-226: “We also assessed the responses of neurons to full-field changes in luminescence, directionally moving spots mimicking a moving target, as well as to localised Gabor patches drifting in 8 different directions (see methods, Extended Data Fig. 5), as examples, respectively, of ethologically relevant full-field static and spatially localised kinetic visual stimuli. [...] Similarly, 71% of the responses to drifting Gabor patches in superficial layers corresponded to the activation of spatial receptive fields while this decreased to 24% in lower intermediate and deep layers where direction and orientation selective neurons were more prevalent (Extended Data Fig. 5g-i) , further strengthening the notion that kinetic features of the visual scene are preferentially transmitted to the motor layers of the SC. Importantly, we found a conservation of the preferred direction tuning among stimulation paradigms, whereby neurons tuned to a particular direction of gratings, were tuned to [...] drifting Gabor patches of the same direction (average preferred [...] °gratings-°Gabor: 27 °, Extended Data Fig. 5k).”

The Materials and Methods section was also updated to include these new analyses:

Line 1208-1212 (M&Ms > Behaviour > visual stimulation): “In order to assess the spatial receptive field of neurons to moving stimuli we used moving Gabor patches at 24 different locations (6 along x axis and 4 along y axis) moving in 8 different directions (45° steps) for 1.5 seconds (spatial frequency: 0.08 cycles per degree and 2.83 Hz temporal frequency). All directions and locations were randomly presented 3 times for a total of 576 presentations corresponding to 3 times 8 directions at 24 locations.”

Line 1338-1346 (M&Ms > Quantification and Statistical Analysis > visual tuning): “Direction and orientation selectivity for small moving spots and for Gabor patches was measured as described for gratings. Two separate analyses of selectivity and preferred angle of tuning were performed for Gabor patches: a location-independent analysis by averaging the responses to each direction of movement across all locations and another analysis only considering the responses at the location of maximal average activity. “

Line 1351-1353 (M&Ms > Quantification and Statistical Analysis > visual tuning): “The same analysis was performed to assess whether neurons had a spatial and kinetic RF as measured with

Gabor patches. The centre of the location of maximal response was used as centre of the RF.”

EXTENDED DATA FIGURE 5

Extended Data Figure 5. Mapping of visual tuning properties in response to drifting Gabor patches, small moving spots and changes in luminescence in the SC.

a, Gabor patches drifting in 8 different directions were randomly presented at 24 locations of the screen. **b**, Example neuron displaying location-specific tuning to moving gratings measured by averaging all directions of movement per location (left), location-independent directional tuning as measured by averaging the response to drifting patches for each direction across all locations (middle, yellow), and a strong direction tuning at the preferred location (right, green). Shaded areas in polar plots indicates \pm s.e.m. **c**, Example neuron without a location preference but tuned to particular direction of movement of the Gabor patches. **d**, Mice were also presented with a small spot (1.5 cm diameter) starting at 8 locations along the corners and border of the screen (red dots) and moving in 8 directions towards and away from the mouse’s centre of gaze on the screen (red cross). **e**, Example tuning of a neuron responding to a small spot moving in a particular orientation. Shaded areas in polar plots indicates \pm s.e.m. Average firing rate for each direction is shown as a heat map and the average time change across directions to identify if a neuron prefers approach vs retreat is shown at the bottom (i). The tuning of this neuron corresponds to the spot crossing its spatial receptive field mapped using drifting Gabor patches (ii). **f**, Example tuning to a small moving spot for a neuron without spatial receptive field. **g**, Example neuron tuned to full field moving gratings (i), Gabor patches (ii) and moving spot (iii). **h**, Distribution of neurons responding to particular locations in space mapped using Gabor patches (spatial Gabor, blue), neurons showing direction or orientation selectivity mapped using Gabor patches (Gabor OS/DS, green) or full field gratings (red), neurons tuned to moving spots (spot OS/DS, black) or tuned to head rotations (head tuned, purple). The number of recorded neurons per depth of recording, indicated

by the shading in the SC diagram, is shown in parenthesis (N = 4 mice). **i**, Distribution of neurons tuned to particular locations (spatial Gabor, blue), direction or orientation selective neurons (Gabor OS/DS, green) or both (grey) among all neurons tuned to drifting Gabor patches. **j**, Overlap of the tunings shown in h for individual neurons, computed using Intervene. Static features are preferentially represented at the top, while kinetic ones are preferentially represented towards the bottom. Note that the number of neurons is represented by the shade of colour and not by the surface of the intersection. **k**, Angular difference between the preferred angle of tuning measured using full field gratings and moving spots (black) or Gabor patches (green).

While some neurons showed location-independent tuning to direction of motion of the Gabor patches, tuning was stronger at the location of maximal response even if no localised RF was found (Extended Data Figure 5c). This finding was already encapsulated by the model in Figure 4 that predicts direction selective neurons in deep layers would still carry some broad spatial information due to the alignment and convergence of concentric location → direction → motion columns. We have also updated Extended Data Figure 10 with the concentric alignment displayed by neurons tuned to Gabor patches with location-specific responses and direction or orientation selectivity at those locations.

EXTENDED DATA FIGURE 10

Extended Data Figure 10. Concentricity of individual OS and DS neurons with localised RF.

d, A second set of experiments was performed to estimate the spatial RFs of neurons across the SC using drifting Gabor patches (N = 4 mice). For neurons displaying spatial and direction selective tuning, a vector was drawn between the centre of the spatial RF (red box and black arrow, ii) and the average centre of gaze on the screen (red cross). The direction of this concentric vector was compared to the preferred direction of Gabor patch movement at the preferred location (green, iii, corresponding to the location marked by the red box in ii). **e**, The angular difference of those two vectors is compatible with concentric direction selectivity.

Fourth, there is a large difference between the anesthetized patch data shown in Fig. 2h and the awake tetrode data in Fig. 2n. I am just curious about the origin of this difference. Can the authors show or, at least, discuss whether this difference is caused by the state of the animal or by the method of recording?

Given that isoflurane has been shown to reduce and slow down neuronal transmission by increasing inhibition, decreasing presynaptic release and neuronal excitability²⁷⁻²⁹, the discrepancies observed between whole-cell recordings in anaesthetised animal and awake tetrode recordings are likely due to the anaesthesia. On one hand, neurons receiving direct retinal input could have stronger tuning due to the lack of biological noise but on the other we might see a steeper decrease in tuning due to increased inhibition and failure to drive deeper layers. However, we cannot discard a systematic bias during blind whole-cell recordings, for example, oversampling neurons with bigger cell bodies or neurons in locations with more stable extracellular matrix. In an attempt to address this conundrum, we recorded the response to visual stimuli across layers of the SC in 4 WT mice using tetrode recordings before and after isoflurane anaesthesia. Out of 60 neurons recorded, 17 had significant spiking activity in anaesthetised conditions. Of those, only 9 DS or OS neurons recorded in awake conditions conserved significant and consistent response to moving gratings under anaesthesia. These neurons, all located in the superficial layers, had an average latency of response to their preferred direction of 65 +/- 12 ms in awake conditions vs 128 +/- 15 ms under anaesthesia (Rebuttal Figure 1). This increase in latency and variability could contribute to the loss of responses in deeper layers under anaesthesia.

Rebuttal Figure 1: Latency to visual responses for neurons recorded in awake conditions and under isoflurane anaesthesia (n = 9 cells recorded in N = 3 mice, one-tailed paired Student's t-test $p = 0.0005$).

These are my main questions. I also have a number of comments on the details of the manuscript.

L52. Here and throughout the paper 'motor layers' is used, but no definition is given in terms of depth or anatomically identified layers. From Fig. 2 in combination with the text, I deduce that both the intermediate and deep layer are included in this, but please add this explicitly. I looked for help in references 15 and 16 that are cited at this point, but they do not use the term 'motor layers'.

We have now explicitly specified this. Line 53: "Moreover, these earlier studies have primarily focused on characterising the static and spatially restricted receptive fields of visuomotor units, while neglecting responses to kinetic visual features, which are more prominent in the intermediate and deep layers of the SC, also known as the motor layers^{34,35}".

L62 "Specifically, collicular motor units are preferentially tuned to visual flow of the opposite direction to the movement that they decode" This line also occurs in the discussion. I am a little confused by the word "decode". The motor neuron activity precedes the motion, while the word "decode" suggests that the activity follows the motion. The example neuron in Fig 3bg actually also seems to decode the head movement just before the activity, which is aligned to the visual stimulus. Is there not a better description than 'decode'? Also at this early point in the manuscript replacing "movement" by "head movement" helps the reader to understand the sentence, because otherwise movement could be taken to mean the movement of the visual stimulus.

We appreciate that the terms decoding has been often used loosely in literature; however, here we use it in agreement with its standard meaning whereby sensory information is “encoded” into neural activity and neural activity can be “decoded” into movement intentions. We agree with the reviewer with the need of specifying the type of movement, and this sentence now reads: “Specifically, collicular motor units are preferentially tuned to visual flow of the opposite direction to the head movement that they decode”.

L163. Here it says 56% of neurons in motor layers, but the figure shows $(6+3)/(10+8) = 50\%$ of the neurons. Please explain better or correct.

We thank the reviewer for spotting the mistake. Upon careful revision of the data, a preliminary version of the graph missing a datapoint was included by error, this has been now corrected.

L194. Here ‘deep intermediate layers’ is used. Is this ‘deep/intermediate layers’ or the deeper part of the intermediate layer?

We realise the references to different layers across the SC were not homogeneous throughout the manuscript. For all main figures we have divided the SC in three rough areas that correspond to:

- the stratum zonale (SZ) and stratum griseum superficiale (SGS), referred in the manuscript as “superficial layers”
- the stratum opticum (SO) and upper stratum griseum intermedium (uSGI), referred as “upper intermediate layers”
- lower SGI (lSGI) and SG profundum (SGP), referred as “lower intermediate and deep layers”

This division has been clearly stated in the main text and Figure 1 legend and has been respected throughout the manuscript. Line 101 : “Pitx2^{ON-PRE} neurons were pooled according to their distribution across layers within 3 main domains: 1) stratum zonale (SZ) and stratum griseum superficiale (SGS), referred hereafter as SZ/SGS or superficial layers, 2) the stratum opticum (SO) and upper stratum griseum intermedium (uSGI), referred henceforth as SO/uSGI or upper intermediate layers, and 3) lower SGI (lSGI) and SG profundum (SGP), referred as lSGI/SGP or lower intermediate and deep layers.”

Fig 2. The figure title is “Neurons in motor layers lack receptive fields ...”. This is a narrow definition of receptive field. What is shown in the figure is that neurons in the intermediate and deep layer do not respond well to small flashing spots.

We agree with the reviewer that the use of the term “receptive field” (RF) when only referring to a particular visual stimulation paradigm could be misleading. We now consider any tuning of neurons to visual stimuli as a RF and we specify whether this is spatial (elicited by the appearance of stimuli only at particular location in the visual space), static (owing to the static nature of the stimuli) or kinetic (if the visual stimuli used is either moving full field grating, drifting Gabor patch or moving spots). This would leave four possible combinations of RFs: spatial static (ssRF, as measured with flashing squares), static (as measured with static gratings), spatial kinetic (as measured with Gabor patches) or kinetic (as measured with moving gratings). Keeping in mind this new definition and to help readers through the manuscript we have only used an acronym for ssRFs

as this type of RF most closely recapitulates the canonical classic model of static alignment. For all other tunings we have attached the corresponding adjective (spatial/localised, kinetic or static) without providing an acronym. ssRFs are briefly introduced in the introduction. Line 65: “static spatial receptive fields (ssRFs)” and further described in Line 170: “well-defined spatially confined static receptive fields (ssRFs) as measured with localised flashing black or white squares”.

Furthermore, we have now changed the title of Fig. 2: “Figure 2. Premotor neurons and neurons in motor layers respond preferentially to visual flow.”

Fig 2f. I cannot find in the Methods how this latency is computed. Can you add this?

To determine the latency of response we calculated the differential of the membrane potential during baseline to assess the SD of baseline presynaptic activity. For each repetition at the preferred direction or orientation of tuning we determined the time of the first event with a slope >3 SDs of baseline presynaptic activity and of at least 0.5 mV/ms. This is now included in Materials and Methods. Line 1264 (M&Ms > Quantification and Statistical analysis > Whole-cell *in vitro* and *in vivo* electrophysiology) : “To determine the latency of response to visual stimuli recorded in whole-cell mode *in vivo* we first calculated the differential of the membrane potential during baseline (grey screen) to assess the SD of baseline presynaptic activity. For each repetition at the preferred direction or orientation of tuning we determined the time of the first event with a slope >3 SDs of baseline presynaptic activity and of at least 0.5 mV/ms.”

Fig 2f. For the deep layer only for 2 neurons the latency was computed. It is a stretch to use an ANOVA for this. You could consider removing it.

We have now removed this analysis.

Fig 2h suggests that there were 3 cells responsive to gratings but in Fig 2f the latency is only computed for 2. What were the exclusion criteria that were used?

Albeit displaying clear responses to moving gratings, the level of depolarisation in some recordings prevented the accurate determination of the rising phase of the first EPSP. We discarded recordings with SD of baseline presynaptic activity > 1.5 mV. This is now included in Materials and Methods “Quantification and Statistical Analysis – Whole cell *in vitro* and *in vivo* electrophysiology”. Line 1269: “Recordings with a SD of baseline presynaptic activity > 1.5 mV were discarded.”

Fig 2h,n. I cannot find how responsive (to gratings or RF) was defined. Can you add this?

This information is available in the Materials and Methods under “quantification and statistical analysis”. Line 1318 (M&Ms > Quantification and Statistical Analysis > Visual tuning): “Analysis routines for visual tuning were developed in Igor Pro (WaveMetrics Inc., Portland, OR). The neuronal response to drifting sinusoidal gratings was averaged for all trials and normalised to the baseline firing rate. The selectivity was then calculated both in direction and orientation space (360° and 180° respectively) by computing the mean orientation and direction vectors in polar

coordinates, described by their modulus (corresponding to the selectivity index) and average angle.

$$SI = \left| \frac{\sum_k R(\theta_k) e^{2i\theta_k}}{\sum_k R(\theta_k)} \right|$$

$$\bar{\theta} = \text{atan} \left(\frac{\sum_k \cos \theta_k}{\sum_k \sin \theta_k} \right)$$

Where $R(\theta_k)$ is the response at each sampled direction or orientation θ_k (12 for direction space and 6 for orientation space). Static gratings were probed only in orientation space. Given that the modulus of the vectors is dependent on the firing pattern and firing rate of the neuron, the same calculation was performed for shuffled spike times to obtain a probability distribution of shuffled DSI and OSI to probe for both direction selectivity and orientation selectivity (see “generation of shuffled dataset”). To determine whether a neuron was tuned to moving or static gratings 3 parameters were used: the selectivity index (≥ 0.1), the trial-to-trial angular variance (≤ 0.8 ; used as a measure of the reliability of selectivity) and the significance of the SI compared to the shuffled distribution. If two out of the three criteria were fulfilled the neuron was considered tuned. If the criteria were met for both direction and orientation spaces, a neuron was classified as direction selective if $DSI > OSI$, and orientation selective if $OSI > DSI$.

Direction and orientation selectivity for small moving spots and for Gabor patches was measured as described for gratings. Two separate analyses of selectivity and preferred angle of tuning were performed for Gabor patches: a location-independent analysis by averaging the responses to each direction of movement across all locations and another analysis only considering the responses at the location of maximal average activity.

In order to determine the spatial tuning of neurons to static stimuli, we averaged the response (firing rate) to flashing 9 cm² squares per location in the screen (15 by 11 locations) for each colour separately (black or white) and divided it by the response during baseline conditions and obtained a 2D matrix corresponding to the increase/decrease of firing rate over baseline per location. This matrix was further transformed into Z-scores. The max Z-score was compared to those obtained performing the same analysis in 1000 shuffled trials (see “Generation of shuffled datasets”). If the maximal Z-score was >2 and ranked higher than the top 5% of those obtained from the shuffled distribution, a 2D Gaussian was the fit to the matrix. The centre of the Gaussian fit was used to determine the centre of the ssRF. The same analysis was performed to assess whether neurons had a spatial and kinetic RF as measured with Gabor patches. The centre of the location of maximal response was used as centre of the RF.”

L200: “This is surprising given that visual-motor integration in the SC has generally been hypothesised to rely on the alignment of static sensory and motor features, such as stimulus receptive field and movement end-points” No need to change the text, but this is a bit of a straw man hypothesis. I think that a general hypothesis to explain the primate data has been that of the presence of a saliency map in the superficial layers transforming into a gaze-directing map in the deep layers, rather than neurons only responding to static spots of lights. This interpretation could be reconciled with the currently presented data, by interpreting the static spots used in the study as just lacking in saliency.

We agree with the reviewer that this manuscript could be interpreted as a study of what visual

features are more “salient” for the premotor SC. In monkeys, two types of maps have been proposed in the SC to explain sensory-guided goal oriented actions³⁶. These are 1) a saliency map of visual features in superficial layers of the SC, where regions of the visual space are represented by any salient visual feature, and 2) a downstream priority map in deep layers of the SC which would integrate the saliency map with task- and goal-relevant information^{36,37}. The prevalent Itti saliency model³⁸ implies a feature-agnostic saliency map with a retinotopic organisation, meaning that all types of visual features (colour, luminescence, orientation, motion) would be weighted as equally salient within each location in space. This model has been quite successful at explaining gaze shifts in humans during image presentation³⁹, as well as predicting superficial SC activity during dynamic video presentations in monkeys⁴⁰ and is further supported by the broad types of stimuli preferences in superficial layers in Monkeys, together with the lower prevalence of direction selective neurons (10-17% depending on the study)⁴¹⁻⁴³.

In our previous version of the manuscript we were oversimplifying the problem by using a spatial and static visual stimulus (flashing squares) and a non-spatial and kinetic stimulus (full-field gratings). Some of our data could be in agreement with a retinotopic saliency map, while others are harder to reconcile. We are now including a new dataset where localised Gabor patches were presented as an example of spatial yet kinetic visual stimulus. We observed a very high proportion of neurons tuned to Gabor patches, indicating that motion could be more salient than flashing stimuli, as pointed out by the reviewer, but most importantly, we saw a shift between which visual feature of Gabor patches was preferentially encoded in superficial (spatial) to deep (kinetic) layers of the SC. Overall, although we agree that the visual responses in the superficial layers could be in agreement with a saliency map, we believe that we have uncovered a previously overlooked feature-biased saliency-to-priority map conversion.

Although an in depth discussion on saliency maps would be beyond the scope of this paper, we agree with the reviewer that this is an interesting and potentially very important feature in the SC that remains unanswered. We now briefly discuss it in **Line 468: “While our superficial layer recordings support the existence of a saliency map in retinotopic space^{36,38}, we also uncovered a previously overlooked feature-biased saliency-to-action map conversion in deeper layers of the SC”**

L906. Should “200ms preceding and 800ms following the peak amplitude of a spike” be “200us ... 800 us...”?

We thank the reviewer for spotting the mistake. This has now been corrected.

L933. Was there any time between the flashes?

No time was left between flashes. However, the flash duration, the number of repetitions and locations limits any bleeding through from previous presentation.

L1120. I find Supplementary figure 1 very confusing. The screen appears upright, i.e. parallel to the drawn z-axis. However, the point P1 = (x0,0,0) is drawn on the screen above P2 = (0,y0,0). The both have a z-coordinate of 0 suggesting that the screen has z-coordinate 0 and is lying flat. I am probably misinterpreting the figure, but from this start I am confused and have not checked the calculations in detail.

Indeed, the triangle p1-origin-p2 is in the x-y plane with a $z = 0$ elevation. In other words, p1 and p2 are the intersections of x and y axis with the screen, while the screen is parallel to the z axis. We have re-drawn this image for clarity.

Supplementary Fig. 1: The geometrical set-up of the screen in the laboratory's frame of reference.

Supplementary Fig. 2: The geometric set-up of the r_e^H vector in the head frame (which coincides with the lab frame at time zero).

L1197. The firing rate update rule includes epsilon twice. The second epsilon does not have a dependence on the time step. The diffusion rate of this model would then depend on the time step. Is the given equation indeed correct? A more common choice would be to use something that does not have such an explicit dependence on the time step size.

We thank the reviewer for noticing this typo in our equation as written in the manuscript, where there should only be a single noise term. We have now removed the second epsilon from the equation. All computations were performed with a single noise term.

Referee #3 (Remarks to the Author):

There is substantial data supporting that the superior colliculus is organized into several laminae with convergence of inputs from multiple sensory modalities (i.e. visual, somatosensory and auditory cues) and regulates distinct motor behaviors, including defensive behaviors (i.e. escaping or freezing to a looming visual stimulus and quick suspension of locomotion to illuminance change), and goal-orientation behaviors (i.e. hunting behavior). Undoubtedly, these behaviors are consequence of evolution for survival to avoid natural threats or harvest essential energy. On the one hand, mounting studies have made significant progress towards understanding the neural mechanisms of these behaviors from the perspective of neural circuits, which SC mainly contributes to integrating and transforming the multisensory. On the other hand, a cluster of well-aligned neurons which marked with Pitx2 and distributed in deep layer of SC is reported to control eye and head movement. Of note, goal-orientation like foraging behaviors depends on the head rotation and eye saccade in the rodent animals. However, the internal integration of intra-collicular network about sensorimotor transformation remains to be interrogated systematically. Hence, combined with the two fundamental role of sensory integration and motor modulation in SC, some intriguing and feasible points about connectivity of the two function of SC are worthy to dig out. That being the case, in this study, a new kinetic alignment model which decodes potentially goal-orientation behavior is proposed aiming to claim the functional alignment between sensory and motor maps unambiguously. The authors dissected a microcircuit among separate layers of SC by identifying premotor neurons (Pitx2 on-pre) which synapse onto Pitx2 on neurons accurately. The putative motor units, pitx2 on neurons, received direct visual signals transmitting from Pitx2 on-pre in superficial and indirect from intermediate layers, both of which responded preferentially to kinetic visual stimuli. Then these neurons drove opposite head movement for interception of target. This manuscript pursuing some of the long-standing questions about sensorimotor transformation of superior colliculus might greatly enriches our understanding in a more general sense. It provides more compelling neurobiological evidences and theoretical basis for deconstructing the survival instinct. However, although the research theme is intriguing, there are still some issues I concerned yet to refer.

1. In figure1h, Pitx2 on-pre neurons in intermediate layers performed strong disinhibition effect after GABAR blockade, accompanying with long duration of depolarization up to 20s. Theoretically, the cell membrane potential should be in a state of sustained depolarization. However, the example trace of EPSP showed in figure1h (middle) decayed back to baseline at about 100 ms. The statistics of the decay time of EPSP should be provided.

The example EPSP shown in Figure 1h (middle) was recorded before adding GABAR antagonists. The membrane potential of all neurons recorded was initially held at -70 ± 1 mV with up to -50 pA current injection at an average of -20 ± 6 pA current injection. The addition of GABA_A and CGP lead to a mild, although significant, depolarisation of the membrane potential (-67 ± 2 mV; paired one-tailed t-test $p = 0.0260$). This information is now included in Extended Data Figure 1k.

The average decay time constant for neurons presenting polysynaptic activation, like that shown in Figure 1h (middle), was 110 ± 24 ms. We have now added this data, together with the time constants before and after the addition of the GABAR antagonists to Extended Figure 1j. It is worth noting that

light activation elicited spiking in most of the recordings included in Extended Data Figure 1j right panel after GABAzine and CGP superfusion.

EXTENDED DATA FIGURE 1

Extended Data Figure 1. Characteristics of Pitx2^{ON-PRE} and Pitx2^{ON} neurons.

[...] j, Decay time constant of excitatory postsynaptic activity before and after GABAR block for neurons receiving direct (left, one-tailed paired t-test $p = 0.0979$) and indirect (right, one-tailed paired t-test $p = 0.0214$) retinal input. k, Resting membrane potential before and after GABAR block (one-tailed paired t-test $p = 0.0260$).

2. The axon terminals of RGC mainly distributed locally in superficial layer of SC as figure 1e showed. However, a fraction of neurons in intermediate deep layers of SC still received monosynaptic retinal inputs (figure1g). It is necessary to reconfirm the accuracy of the data.

The recordings in acute brain slices were performed using a Scientifica set-up consisting of a fluorescent upright microscope equipped with a camera, an LED light stimulation system and a high resolution monitor. Since we were able to monitor in line the location of the cell body recorded within the SC, we are very confident of the location and identification of the Pitx2-cre^{ON-PRE} neurons recorded for this experiment. We have used the soma location at the time of recording to assign recorded neurons to particular layers but it is highly likely that the dendritic tuft of those few neurons displaying direct retinal responses would expand into superficial layers. Although we filled all neurons recorded in acute brain slices with biocitin for post-hoc reconstruction using streptavidin with a far-red fluorophore, the expression of red fluorescent markers in retinal axons in superficial layers rendered the task very difficult, particularly for projections within the superficial layer and we could only obtain reliable partial reconstructions. When possible, soma location was re-confirmed after imaging. The partial reconstructions with their relative soma locations are now included in Extended Data Figure 1l.

EXTENDED DATA FIGURE 1

Extended Data Figure 1. Characteristics of Pitx2^{ON-PRE} and Pitx2^{ON} neurons.

l, Reconstruction obtained from 10 of the neurons included in Fig 1.

3. The number of Pitx2 on neurons labeled by mCherry in figure1a was obviously less than the result

reported before (ref 3 in manuscript) . It would be compelling to figure out the reason and change another figure.

We have changed the image in Figure 1a to a better quality one that more faithfully represents the expression and distribution of premotor neurons within the colliculus in line with previous literature. We have gone through the raw data included in Masullo et al., 2019 Figure 7A-D and found the same level of expression and variability across mice. Most of Masullo et al., experiments were done using transgenic lines labelling all Pitx2 positive neurons, while here, and in Figure 7 of Masullo et al., paper, we use an FRTed AAV to express the TVA receptor in a subset of Pitx2-Flp neurons. Viral injection only labels a subpopulation of neurons, which will vary depending on the titre of the viral batch, the injection site and the volume of injection. However, after having analysed all data included in both studies we found no major differences. The number of total starting Pitx2^{ON} neurons per mouse for Masullo et al., dataset was 203 +- 21 (n = 3 mice) and we found 178 +- 25 (n = 4 mice, two-tailed Student's t test p value = 0.5).

FIGURE 1

Figure 1. Two functionally distinct premotor populations channel direct and indirect retinal input onto collicular motorneurons.

[...] **b**, Immunostaining of Pitx2^{ON}neurons (mCherry) and their rabies-traced presynaptic partners (GFP, bar: 100 μ m). **Expanded fields containing example Pitx2^{ON-PRE} neurons in superficial (top) and intermediate layers (bottom) are shown.**

4. It was interesting and attractive to find the disinhibition phenomenon of Pitx2 on-pre neurons evoked by photostimulation and GABAR blockade in intermediate/deep layers of SC in figure 1h. The potential mechanism and physiological significance would be illuminating to illustrate.

We agree with the Reviewer that there is probably a very interesting mechanism by which inhibition can gate goal-oriented actions. However, dissecting the precise role of inhibition in sensorimotor transformation would require an extensive battery of new experiments, both *ex vivo* and *in vivo* and would be outside of the scope of the paper. Nevertheless, in this rebuttal we have further investigated the relevance and circuitry responsible for this inhibition. While by no means exhaustive, we believe that we have significantly enriched our understanding of the inhibitory mechanism described. We refer the Reviewer to the answer to point 6 for a more detailed description of these new findings.

5. In figure 2b I, the neurons labeled by biocitin in the superficial layers of SC were defined as narrow-field neuron. But there was no evidence to prove the identity of the neurons. Data about the morphological, intrinsic electrophysiological properties, or specific markers (Grp-KH288) would help to clarify this.

The reviewer raises a very important point, which we decided to address with three separate experimental approaches.

First, we now include a supplementary table with the spike properties and resting membrane properties of neurons recorded both in acute brain slices for Figure 1 and *in vivo* for Figure 2 together with the tuft measurements of those successfully reconstructed *in vivo*.

	N	RMP (Vm)	RI (MOhm)	Sag (mV)		
Direct	13	-47.3 +- 0.9	543 +- 28	31.2 +- 5.9		
Poly.	13	-50.2 +- 0.9	627 +- 24	136.6 +- 17.3		
RAMP	10	-52.6 +- 1.2	845 +- 90	39.7 +- 7.2		
No activation	11	-43.8 +- 0.9	599 +- 24	16.4 +- 1.2		
	AP threshold (mV)	AP amplitude (mV)	AP width (ms)	AHP Amplitude (mV)	AHP width (ms)	
Direct	-37.7 +- 0.4	55.7 + 0.8	1.0 +- 0.0	13.3 +- 0.5	20.5 +- 1.8	
Poly.	-32.4 +- 0.5	43.7 +- 1.1	1.4 +- 0.0	15.5 +- 0.4	52.5 +- 2.5	
RAMP	-37.0 +- 0.8	55.7 +- 0.8	1.3 +- 0.0	15.3 +- 0.8	26.9 +- 3.2	
No activation	-38.4 +- 0.6	61 +- 1.1	0.9 +- 0.0	15.8 +- 0.6	14.3 +- 1.4	
+ 100 pA	Num of AP	Frequency of burst (Hz)	Freq. Adaptation (%)	Freq. spikes 1 to 3 (Hz)	Delay to 1 st AP (ms)	Time after last AP (ms)
Direct	19.3 +- 1.2	38.3 +- 1.5	183.2 +- 7.2	51.6 +- 1.9	24.8 +- 1.4	206 +- 24
Poly.	13.4 +- 1.2	18.2 +- 1.5	245.4 +- 45.7	26.2 +- 2.2	24.7 +- 2.8	332 +- 30
RAMP	18.4 +- 2.2	40.2 +- 2.8	172.1 +- 8.1	57.3 +- 5.1	43.2 +- 7.1	285 +- 35
No activation	33.7 +- 1.4	50.1 +- 1.2	201.2 +- 6.2	74.1 +- 1.8	22.2 +- 2	123.2 +- 23
+ 200 pA	Num of AP	Frequency of burst (Hz)	Freq. Adaptation (%)	Freq. spikes 1 to 3 (Hz)	Delay to 1 st AP (ms)	Time after last AP (ms)
Direct	3.8 +- 0.2	62.7 +- 4.2	103.1 +- 7.3	73.7 +- 4.8	3.9 +- 0.4	702 +- 16
Poly.	6.8 +- 1.4	30.7 +- 3.9	72.2 +- 10.8	40.9 +- 5.2	6.6 +- 0.5	648 +- 25
RAMP	9.7 +- 2.1	42.1 +- 5.3	82.1 +- 9.1	49.4 +- 6.4	11.6 +- 1.3	672 +- 27
No activation	14.1 +- 1.8	102.4 +- 2.3	173.8 +- 15.7	135.4 +- 5.1	4.1 +- 0.5	647 +- 22

Supplementary Table 1. Electrophysiological properties of neurons recorded in acute brain slices and included in Figure 1

	N	RMP (Vm)	RI (MOhm)	Sag (mV)	Tuft width (um)	Tuft length (um)
Superficial	24	-50.1 +- 3.4	282 +- 44	28.2 +- 17	277 +- 160 (N = 2)	468 +- 3 (N = 2)

Upper Intermediate	12	-41.9 +- 3.5	297 +- 97	3.8 +- 1.6	362 +- 49 (N =12)	401 +- 44 (N =12)
Lower Intermediate/Deep	9	-42.2 +- 3.4	164 +- 25	2.3 +- 1.0	379 +- 174 (N =3)	452 +- 241 (N =3)
AP threshold (mV)						
Superficial		-28.8 +- 2.0	21.5 +- 2.9	0.7 +- 0.1	7.3 +- 1.2	31.0 +- 7.0
Upper Intermediate		-26.0 +- 3.7	14.9 +- 2.4	0.7 +- 0.1	5.5 +- 0.9	26.5 +- 6.0
Lower Intermediate/Deep		-31.3+- 2.3	11.4 +- 0.8	0.6 +- 0.0	6.0 +- 0.7	41.0 +- 14.9
AP amplitude (mV)						
AP width (ms)						
AHP Amplitude (mV)						
AHP width (ms)						
+ 100 pA						
	Num of AP	Frequency of burst (Hz)	Freq. Adaptation (%)	Freq. spikes 1 to 3 (Hz)	Delay to 1st AP (ms)	Time after last AP (ms)
Superficial	8.9 +- 2.4	11.4 +- 3.2	150.5 +- 63.0	23.7 +- 7.2	198.6 +- 59.6	49.9 +- 13.9
Upper Intermediate	13.4 +- 1.2	18.2 +- 1.5	245.4 +- 45.7	26.2 +- 2.2	24.7 +- 2.8	332 +- 30
Lower Intermediate/Deep	18.4 +- 2.2	40.2 +- 2.8	172.1 +- 8.1	57.3 +- 5.1	43.2 +- 7.1	285 +- 35
+ 200 pA						
	Num of AP	Frequency of burst (Hz)	Freq. Adaptation (%)	Freq. spikes 1 to 3 (Hz)	Delay to 1st AP (ms)	Time after last AP (ms)
Superficial	19.8 +- 8	33.5 +- 12.1	238.0 +- 91.6	54.2 +- 17.9	3.9 +- 0.4	702 +- 16
Upper Intermediate	6.8 +- 1.4	30.7 +- 3.9	72.2 +- 10.8	40.9 +- 5.2	6.6 +- 0.5	648 +- 25
Lower Intermediate/Deep	9.7 +- 2.1	42.1 +- 5.3	82.1 +- 9.1	49.4 +- 6.4	11.6 +- 1.3	672 +- 27

Supplementary Table 2. Electrophysiological and morphological properties of neurons recorded in whole-cell mode *in vivo* and included in Figure 2

Secondly, while CRE lines for at least subpopulations of both narrow field (Grp-KH288 cre) and wide field (Ntsr1-GN209 cre) neurons have previously been identified^{17,18}, these lines and Pitx2-cre mice rely on the same CRE recombinase system and cannot be used in combination to assess the overlap between Pitx2^{ON-PRE} and Grp^{ON} or Ntsr1^{ON} neurons. For Figure 1 we used a Pitx2-Flp line; however, this line still relies on the use of the Pitx2-Cre mice (i.e. Pitx2-Cre::Tau-LSL-FLP) and hence also expressed the CRE recombinase. As an alternative, we have performed single cell in situ hybridisation on Pitx2^{ON-PRE} neurons to test their expression of Grp and Ntsr1. This strategy, albeit the best available option, would only identify neurons actively transcribing these genes instead of all neurons that have

expressed these genes at some point in the animal's life. We found that in the SC of adult mice, Grp and Ntsr1 were expressed very scarcely. It has been shown that positive neurons in adult Ntsr1-GN209 cre mice do not express Ntsr1 but differentially express Npnt¹⁹. Thus, we tested for expression of Npnt in Pitx2-cre^{ON-PRE} as well as expression of Cbln4, a gene identified as marking another subpopulation of narrow field neurons²⁰ (Extended Data Figure 1 n-p). We found a significant proportion of Pitx2-cre^{ON-PRE} co-localised with Cbln4.

EXTENDED DATA FIGURE 1

Extended Data Figure 1. Characteristics of Pitx2^{ON-PRE} and Pitx2^{ON} neurons.

[...] **j**, Decay time constant of excitatory postsynaptic activity before and after GABAR block for neurons receiving direct (left, one-tailed paired t-test $p = 0.0979$) and indirect (right, one-tailed paired t-test $p = 0.0214$) retinal input. **k**, Resting membrane potential before and after GABAR block (one-tailed paired t-test $p = 0.0260$) **l**, Reconstruction obtained from 10 of the neurons included in Fig 1. **m**, Example Pitx2^{ON-PRE} narrow-field neurons traced with G-deleted Rabies virus. **n**, In situ hybridization was performed on SC slices containing Pitx2^{ON-PRE} neurons traced with G-deleted rabies using RNAscope. **o**, Example staining of markers of narrow-field (Grp and Cbln4) and wide-field (Ntsr1 and Npnt). **p**, quantification of Npnt⁺ and Cbln4⁺ Pitx2^{ON-PRE} neurons.

Finally, in order to assess the visual tuning properties of unambiguously identified premotor neurons in superficial layers of the SC *in vivo* and get a sense of whether they would be characteristic of narrow or wide field neurons, we used a viral strategy to express the calcium indicator GCamp6f in Pitx2-cre^{ON-PRE} neurons. We first injected an AAV expressing a TVA receptor protein upon CRE recombination together with an AAV containing FLP-dependent GCamp6f in Pitx2-Cre::tdTomato mice. 3 weeks later, we injected a self-inactivating G-deleted rabies virus (SiR) containing the FLP recombinase and performed a cranial window preparation exposing the superficial layer of the SC. This method affords high throughput identification of the visual properties of unambiguously defined premotor neurons (i.e. presynaptic to Pitx2^{ON} neurons) across the colliculus. Mice were exposed to full field moving gratings and to drifting Gabor patches. We found similar results to those obtained using tetrode recordings, with 42.2% of neurons recorded tuned to localised drifting Gabor patches and 33.8% to full field gratings. These results are now included in Figure 2o-r and Extended Data Figure 6. The majority of neurons displayed direction selectivity (Extended Data Figure 6k) and were organised in functional clusters of similar orientation and direction (Extended Data Figure 6l,m) as previously described in WT⁷⁻⁹. These results are in agreement with the existence of at least a narrow-field subpopulation of Pitx2-cre^{ON-PRE} neurons displaying direction selectivity and narrow retinotopic receptive fields. This organisation further supports the columnar organisation of the SC we propose in Figure 4. These results, together with those obtained to answer the reviewer's following question are now discussed in the main text and corresponding Materials and Methods sections (see below).

Line 227-238: “To confirm whether premotor neurons encode visual kinetic features, we leveraged two newly developed tools - intersectional AAVs¹⁰ and self-inactivating rabies (SiR)¹¹⁻¹³ - that allowed us to selectively record from genetically-defined Pitx2^{ON-PRE} premotor neurons. We performed an initial injection with a mix of an AAV with a flexed TVA receptor protein together with either an AAV containing FLP-dependent GCamp6f, or an AAV with creOFF/flipON-ChR2 in Pitx2-Cre-Tomato mice, followed by a second injection with an SiR expressing the FLP recombinase. We recorded Pitx2^{ON-PRE} using either optetrodes, allowing us to also record the same neurons in freely moving mice, or under a 2-photon microscope yielding higher numbers of recordings to assess their visual tuning properties in superficial and upper intermediate layers. In these unambiguously defined premotor units, we found the same preservation of tuning to kinetic features across layers of the SC as previously found in WT mice (Fig. 2o-r, Extended Data Fig. 6).”

Line 1013-1023 (M&Ms > Surgery): “For selective long-term labelling of Pitx2^{ON-PRE} neurons in the SC, we performed an initial injection with a mix of AAV(2)-hSyn1-FLEX-nucHA-2A-TVA-2A-G(N2c) (titre: 1.5×10^{12} genomic copies/ml) and either AAV(1)-pAAV-nEF-Coff/Fon-ChR2(ET/TC)-EYFP (Addgene 137141, titre: 4.3×10^{12} genomic copies/ml) or AAV(1)-Ef1a-fDIO-GCaMP6f (Addgene 128315, titre: 5×10^{12} genomic copies/ml). Following 3 weeks of expression, a second injection with a self-inactivating Rabies¹¹⁻¹³ SiR-N2c-FLP (EnvA) (titer: titre: 1×10^7 genomic copies/ml) was performed before implanting an optetrode or a cranial window for chronic Ca²⁺ imaging. In a subset of experiments AAV(9)-EF1a-double floxed-hChR2(H134R)-mCherry-WPRE-HGHpA (Addgene 20297, titre: 5×10^{12} genomic copies/ml) was injected in the SC followed by optetrode implant.”

Line 1047-1057 (M&Ms > Surgery): “For *in vivo* chronic selective Ca²⁺ imaging of Pitx2^{ON-PRE} neurons in the SC recordings, mice underwent an initial viral surgical injection with an AAV-TVA and an AAV-GCaMP6f as described above and were injected with Dexafort at 2 µg/g the day prior to surgery. A headpost and a cranial window over the SC were implanted as previously described⁹. Briefly, following isoflurane anaesthesia, Vetergesic was injected subcutaneously at 0.1 mg/kg and a metal head-post was affixed to the skull with Crown & Bridge Metabond. Epivaine was splashed over the skull, and a 3 mm diameter craniotomy was performed on the left hemisphere, centred on the rostral superior colliculus. The surface of the superior colliculus was exposed through removal of the overlying cortex and the SiR was then injected as described above. A 3 mm cannular window was then fixed on top of the colliculus using dental cement (Crown & Bridge Metabond).”

Line 1156-1169 (M&Ms > Electrophysiology > Ca²⁺ imaging): “Mice were imaged from one week after surgery with a two-photon microscope (Bergamo II, Thorlabs), equipped with a 16 x – 0.8 NA objective (Nikon). Mice were recorded awake and headfixed on a custom-made floating platform. Td Tomato positive Pitx2^{ON} neurons and SiR infected cells were excited with a Ti:Sapphire laser at 1030 nm and 920nm, respectively, with a power of around 20 mW (Mai TaiDeepSee, Spectra Physics). Red and green emitted fluorescence were collected through a 607±35 nm and 525±25 nm filters, respectively (Brightline). For imaging of neuronal responses, recordings consisted of either multiple planes at different depths imaged quasi-simultaneously using a piezo device, or a single plane. The pixel resolution was kept at around 0.8 µm per pixel, while the number of pixels, field of view and imaging rate were adjusted to cover the labelled cells. On average this resulted in imaging frame rates of 30 to 60 Hz and pixel dwell times of 0.2 to 0.08 µs. Two-photon recordings were then registered

and ROIs were determined manually and extracted using CalmAn¹⁶ (Flatiron Institute) in Python. Variation of fluorescent values over baseline (DF/F) were computed and used for further analysis.”

Similarly, the neurons recorded in the intermediate/deep layer of SC were assumed as pitx2 on-pre neurons in figure 2c. Although these neurons showed subthreshold responses triggered by direction-selective moving grating stimuli, some data demonstrated in figure3e that motor units in the same layer of SC also tuned to moving grating. Thus it is lack of solid evidence to confirm the neuron identity, Pitx2 on-pre or motor neuron. Further investigation is required to verify this entire issue

Again, the reviewer makes a valid point and an important one in the context of our work. Since we could not be completely certain of whether the neuron recorded in Figure 2C are premotor only, we referred to them as “putative” premotor neurons. However, we do agree with the reviewer that stronger evidence on the nature of visual and motor tuning of unambiguously identified Pitx2-Cre^{ON-PRE} and Pitx2-Cre^{ON} neurons is needed.

In order to address this issue, we performed a new set of experiments aimed at characterising the visual and motor tuning properties of unambiguously defined Pitx2^{ON-PRE} units. We first infected collicular neurons in Pitx2-Cre mice with a combinatorial CRE^{OFF}/FLP^{ON}-ChR2 AAV together with a flexed AAV containing the TVA receptor to prime Pitx2^{ON} neurons for rabies transduction. Secondly, we transynaptically traced Pitx2^{ON-PRE} neurons using Self inactivating Rabies (SiR) virus expressing the FLP recombinase (SiR-FLP) to drive the expression of the FLP dependent CRE^{OFF}/FLP^{ON}-ChR2 AAV selectively in Pitx2^{ON-PRE} neurons (but not in Pitx2^{ON} neurons themselves). Following the SiR-Flp injection, we implanted an optrode consisting of a bundle of 4 tetrodes and one fibre optic allowing us to opto-tag and hence identify neurons presynaptic to Pitx2 neurons during our recordings. Thirteen mice were recorded while foraging in an open field arena to determine the head-rotation tuning of Pitx2^{ON-PRE} cells and then were exposed to moving gratings and Gabor patches in head restrained conditions (Figure 3k,l and Extended Data Figure 6m-o). Results were comparable to those obtained using calcium imaging and those obtained in WT mice. Premotor neurons in superficial layers were strongly tuned to Gabor patches at particular locations in the screen; however, while this feature was lost in deeper layers, some neurons still conserved tuning to direction of movement (Figure 3l and Extended Data Figure 6n). We also recorded Pitx2-ChR2 mice to confirm the existence of an antialignment in genetically defined motor neurons (Figure 3i,j).

FIGURE 2

Figure 2. Premotor neurons and neurons in motor layers respond preferentially to visual flow.

[...] **o**, Diagram of the viral strategy used to selectively record $\text{Pitx2}^{\text{ON-PRE}}$ neurons using Self-Inactivating Rabies virus expressing FLP recombinase in Pitx2 -cre mice and either a combinatorial $\text{Cre}^{\text{OFF}}/\text{Flp}^{\text{ON}}$ virus expressing Chr2 or FLP-dependent GCaMP6f for opto-tagging or 2-photon Ca^{2+} imaging respectively. Peri-stimulus time histogram of 900 5 ms-long laser stimulations (ii, bottom) for an opto-tagged neuron and example raster plot of the 1st 300 stimulations are shown (ii, top). An example image of two $\text{Pitx2}^{\text{ON-PRE}}$ neurons expressing GCaMP6f is also shown (iii, bar: 100 μm). **p** Example responses of two $\text{Pitx2}^{\text{ON-PRE}}$ neurons to moving gratings, recorded either with optotetrodes (top) of 2-photon Ca^{2+} imaging (bottom). **q** and **r** Distribution of $\text{Pitx2}^{\text{ON-PRE}}$ neurons tuned to moving gratings across layers of the SC recorded with optotetrodes (q, N = 10 mice, n = 17 $\text{Pitx2}^{\text{ON-PRE}}$ neurons) or Ca^{2+} imaging (r, N = 7 mice, n = 74 $\text{Pitx2}^{\text{ON-PRE}}$ neurons).

EXTENDED DATA FIGURE 6

Extended Data Figure 6. Visual and motor tuning of motor Pitx2^{ON} and premotor Pitx2^{ON-PRE} neurons in the SC.

a, Example opto-tagged Pitx2^{ON} motor neurons. Peri-stimulus time histogram of 900 5 ms-long laser stimulations (right, bottom) and example raster plot of the 1st 300 stimulations are shown (right, top). **b**, Example STA of angular head displacements in yaw, pitch and roll for a Pitx2^{ON} motor neuron (i). The 3D STA was projected onto the 2D plane of the visual stimulation screen to draw the tuned direction of gaze. The decomposed and average gaze direction is shown in ii. **c**, Tuning to moving gratings of the neuron in b. **d**, Pitx2^{ON-PRE} neurons were opto-tagged using a combinatorial Cre^{OFF}/Flp^{ON} virus expressing ChR2 and a Self-Inactivating Rabies virus expressing FLP recombinase in Pitx2-cre mice. A first injection was performed in Pitx2-Cre mice consisting of two AAVs: one carrying a floxed copy of the EnVA receptor TVA and another promoting the expression of ChR2 in the presence of FLP recombinase but not CRE recombinase. A second injection was performed 3 weeks later consisting of an EnVA Self-Inactivating Rabies (SiR) virus expressing Flp recombinase. In this experiment Pitx2^{ON-PRE} neurons would express Flp (carried by the rabies virus) but not Cre recombinase, and as a result be the only neurons expressing ChR2. An optrode was implanted at the time of the second injection, allowing for the identification of Pitx2^{ON-PRE} neurons by opto-tagging. **e**, Example Pitx2^{ON-PRE} neuron displaying tuning to moving gratings (i) and to drifting Gabor patches (ii). **f**, Overlap of tuning to all types of visual stimuli tested in premotor neurons and their tuning to head rotations (n = 17 Pitx2^{ON-PRE} neurons recorded across 10 mice). **g**, A similar strategy was used to express GCaMP6f in Pitx2^{ON-PRE} neurons by injecting an AAV leading to FLP recombinant-dependent expression of GCaMP6f. Following the infection with the SiR, the cortex above the injection site was absorbed and a chronic cranial window for Ca²⁺ imaging was performed. **h**, Example Pitx2^{ON-PRE} neuron displaying tuning to moving gratings (i) and to drifting Gabor patches (ii). The average $\Delta F/F$ for gratings moving in all 12 directions is shown in red (i, left, shaded areas corresponds to \pm s.e.m.). The average responses to gratings of two opposing directions (black) and individual trial responses (grey) are also shown. **i** and **j**, Distribution of tuned Pitx2^{ON-PRE} across layers of the SC recorded using optotetrodes (i, N = 10 mice, n = 17 Pitx2^{ON-PRE} neurons) or Ca²⁺ imaging (i, N = 7 mice, n = 74 Pitx2^{ON-PRE} neurons). **k**, proportion of direction and orientation selective Pitx2^{ON-PRE} neurons. **l**, Absolute value of the difference in preferred angle of direction of motion of visual stimulus (as measured using gratings or Gabor patches) plotted against the 3D distance in the SC. The number of pairs is shown in parenthesis. DS/OS correspond to pairs of either OS or DS neurons compared in orientation space (180°, left), while DS pairs only consider DS neurons in direction space (360°, right). For OS/DS neurons (N = 29 cells, n = 4 mice), orange line corresponds to the median of the population, lower and upper box boundaries correspond to the 25th and 75th percentile of the population, respectively and notches correspond to the 95% confidence interval of the median calculated by bootstrapping (10000 iterations). For DS pairs (N = 20 cells, n = 3 mice), orange line corresponds to the median of the population and lower and upper box boundaries correspond to the 25th and 75th percentile of the population, respectively. **m**, All direction and orientation tuned cells to full field gratings or local Gabor patches within a volume in the SC are plotted as spheres and colour-coded according to preferred motion axis (i). A, anterior; M, medial; S, superficial. The same volume with only DS neurons colour-coded indicating their preferred direction of motion is also shown (ii).

The combination of SiR viral technology, transectional gene delivery, electrophysiology and behaviour is a particularly powerful strategy and, to our knowledge, this is the first account of long-term functional recording of neurons presynaptic to a genetically defined neuronal population.

Line 227-238: “To confirm whether premotor neurons encode visual kinetic features, we leveraged two newly developed tools - intersectional AAVs¹⁰ and self-inactivating rabies (SiR)¹¹⁻¹³ - that

allowed us to selectively record from genetically-defined Pitx2^{ON-PRE} premotor neurons. We performed an initial injection with a mix of an AAV with a flexed TVA receptor protein together with either an AAV containing FLP-dependent GCaMP6f, or an AAV with creOFF/FlpON-ChR2 in Pitx2-Cre-Tomato mice, followed by a second injection with an SiR expressing the FLP recombinase. We recorded Pitx2^{ON-PRE} using either optetrodes, allowing us to also record the same neurons in freely moving mice, or under a 2-photon microscope yielding higher numbers of recordings to assess their visual tuning properties in superficial and upper intermediate layers. In these unambiguously defined premotor units, we found the same preservation of tuning to kinetic features across layers of the SC as previously found in WT mice (Fig. 2o-r, Extended Data Fig. 6)."

Line 328-329: "The same anti-alignment was found in genetically-defined Pitx2^{ON} motor neurons (Extended Data Fig. 6a-c)."

Line 1013-1023 (M&Ms > Surgery): "For selective long-term labelling of Pitx2^{ON-PRE} neurons in the SC, we performed an initial injection with a mix of AAV(2)-hSyn1-FLEX-nucHA-2A-TVA-2A-G(N2c) (titre: 1.5×10^{12} genomic copies/ml) and either AAV(1)-pAAV-nEF-Coff/Fon-ChR2(ET/TC)-EYFP (Addgene 137141, titre: 4.3×10^{12} genomic copies/ml) or AAV(1)-Ef1a-fDIO-GCaMP6f (Addgene 128315, titre: 5×10^{12} genomic copies/ml). Following 3 weeks of expression, a second injection with a self-inactivating Rabies¹¹⁻¹³ SiR-N2c-FLP (EnvA) (titer: titre: 1×10^7 genomic copies/ml) was performed before implanting an optetrode or a cranial window for chronic Ca²⁺ imaging. In a subset of experiments AAV(9)-EF1a-double floxed-hChR2(H134R)-mCherry-WPRE-HGHpA (Addgene 20297, titre: 5×10^{12} genomic copies/ml) was injected in the SC followed by optetrode implant."

Line 1139-1148 (M&Ms > Electrophysiology > Optetrodes): "For simultaneous single unit recording and optogenetic stimulation, a single optic fibre (core = 100 μ m, NA = 0.22; Doric Lenses, Québec, Canada) was inserted between each bundle of tetrodes during production as previously described¹⁴. Light was delivered using a 473 nm laser diode module (Cobolt 06-MLD, Cobolt, Solna, Sweden) coupled to a 100 μ m multimode fiber (NA = 0.22) through a Schäfter + Kirchoff fiber coupler (Cobolt, Solna, Sweden). The laser power employed in all stimulation experiments was 3-5 mW at the tip of the fiber. For optotagging, mice were probed in an open field arena. Following 10 minutes of acclimatisation, bursts of 30 5 ms-long pulses at 30 Hz were delivered with a 9 seconds rest period in between burst for a total of 900 stimulations over 5 minutes. Blue light-activated units were defined on the basis of the latency of the response to a pulse of light within a time window of 5 ms¹⁵."

λ 6. In line 166~169, the author speculated that the tonic inhibitory currents on the collicular premotor network resulted in the lack of suprathreshold tuning in neurons located in intermediate layer of SC. It would be reasonable to infer that these neurons would display suprathreshold tuning, if GABAR antagonist was administrated locally in SC combined with current stimulation from pipette, or light stimulation from an optical fiber. Additional experiments to make clear were anticipated in follow-up work.

The experiment proposed by the reviewer is of extreme difficulty, involving whole-cell patch clamp recordings in vivo and simultaneous local administration of GABAR antagonist and potential use of an

optic fibre. To our knowledge this has never been done in the SC, a notoriously difficult area for whole-cell blind recordings. Attempts were made to perform whole-cell recordings using an opto-patch system, theoretically allowing for the identification of genetically defined neurons or stimulation/inhibition of GABAergic neurons during recordings. However, following a couple of months of troubleshooting we were unable to successfully further pursue this experimental avenue.

In order to attempt to address the reviewer's point to the best of our abilities we have instead performed recordings in Vgat-Cre mice expressing ChR2 using tetrode recordings in combination with light stimulation of GABAergic neurons. We reasoned that if the reviewer's hypothesis is correct we should also be able to observe the contrary, that is, a loss of direction selective tuning when GABAergic neurons within the SC are stimulated. We recorded 17 neurons in the intermediate layers of the SC of 4 Vgat-Cre mice injected with a floxed AAV expressing ChR2. Both viral injection and optrode implant were performed at the same coordinates. Trials with and without light stimulation were alternated for a total of 6 trials, each consisting of 3 semi-randomised gratings presentations. A 3 min rest period was enforced between trials. Light stimulation was performed via a 100 um diameter fibre optic and consisted of 2-10 mW pulses of 10 ms at 30 Hz. As hypothesised, we found a reduced tuning to moving gratings both considering the full population and only those neurons significantly tuned to moving gratings in control conditions, i. e. without light stimulation. To prevent over excitation, light intensity was adjusted each recording day to elicit a mild modulation of the LFP. This stimulation did not modify the maximal response to moving gratings or the baseline firing rate of the neurons recorded. This data is now included in Extended Data Figure 3p-s. The results are briefly mentioned in the main text and detailed in the methods section and figure legend.

Line 417-418: "strong inhibitory gate primarily impinging on this pathway (Fig. 1h-j) and affecting direction and orientation selectivity (Extended Data Fig. 3p-s)."

Line 1148-1153 (M&Ms > Electrophysiology > Optotetrodes): "In a subset of experiments, Vgat-cre mice expressing ChR2 were recorded in head-restrained conditions to assess the change in tuning to moving gratings following light stimulation of collicular Vgat^{ON}. The same implants and set-up was used in those experiments, although continuous 10 ms-long pulses at 30Hz were delivered during visual stimulation in light-ON trials instead."

EXTENDED DATA FIGURE 3

Extended Data Figure 3. Visual tuning properties in the SC in awake mice.

[...] **p**, Vgat-cre mice, injected with an AAV-DIO-ChR2 and implanted with optrodes, were recorded before and after light stimulation (n = 17 neurons in N = 3 mice; constant 10 ms pulses at 30 Hz). An example direction-selective neuron to moving gratings (grey: control before light stimulation), losing its tuning when nearby Vgat^{ON} neurons are activated (blue) is shown. **q**, Change in SI after light stimulation (opto) for all neurons (left, one-tailed paired t-test $p = 0.0051$) and only neurons tuned in control conditions (right, one-tailed paired t-test $p = 0.0308$). **r**, Change in the rank of the SI compared to a shuffled distribution (significance considered for ≥ 0.95) before and after light stimulation for all

recorded neurons (left, one-tailed paired t-test $p < 0.0001$) and those tuned in control conditions (right, one-tailed paired t-test $p = 0.0036$). s, Maximal response over baseline (left, one-tailed paired t-test $p = 0.2076$) and baseline firing rate (right, one-tailed paired t-test $p = 0.40124$), in control conditions (Ctr, black) and during optogenetic stimulation (Opto, blue).

λ 7. The number of neurons from deep layer of SC was too few. The one-way ANOVA was not recommended. It is necessary to add more experimental data.

We understand there is an apparent lack of data in that subfigure as it represents a very small subpopulation of neurons recorded, namely those recorded *in vivo* in whole-cell patch clamp mode in deep layers of the SC with a clear subthreshold response to visual stimuli. As a reference, we performed a total of 41 high quality *in vivo* whole-cell recordings across the SC out of 178 mice. Since the results are not critical to the main message of the paper we find hard to justify employing the resources and use of animals that would be required to complete this dataset. However, we understand that an ANOVA is not the right analysis for the dataset and we have removed it from the manuscript.

λ 8. In line 257 and extended figure 4h~j, the data indicated no correlation between motor neurons in deep layer of SC and eye movement, which was in contradiction to the result reported before (ref 3). The prior study revealed that photostimulation of Pitx2^{ON} neurons led to the execution of rapid but low-amplitude eye movements (<5 degrees). It was not clear whether the negative data obtained here was due to the insufficient driving force of visual stimulation, or something else. Thus some convincing explanation was expected.

These two datasets are not necessarily contradictory, since they are different type of recordings involving different subsets of neurons. Furthermore, while Masullo et al., showed that stimulation of Pitx2 neurons lead to small amplitude eye movement it is important to stress that they never showed whether recorded spontaneous activity of pitx2 neurons in head fixed mice correlates with eye movements, which is what has been tested here. Here, we uncover that at least a subpopulation of classically defined motor neurons, firing before head rotations both in presence and absence of light during a goal-oriented task, are also tuned to moving visual stimuli. However, as pointed out, we find that these neurons are not involved in the generation of spontaneous saccades in head restrained conditions. Our initial hypothesis is that these neurons could be a subpopulation of Pitx2^{ON} neurons. While driving all Pitx2^{ON} neurons can elicit eye movements, as previously shown, the subpopulation of neurons responsible for goal-independent eye movements is not necessarily the one we describe. We find no generation of eye movements during visual presentation. Nevertheless, since our open field recordings are a priori agnostic to eye movements and neurons tuned to head movements could be tuned to combined eye and head movements, we have produced a new set of data to assess the coupling between eye and head movements while foraging in our experimental conditions. In collaboration with Dr. Jasper Poort, we implemented a miniaturised cameras mounted on the headstage in combination with our head sensor allowing the simultaneous recording of 3D head movements and eye movements. We included this data in our 3D to 2D gaze projection model. We have now included this data in Extended Data Figure 9. The updated model has no effect on the estimation of gaze direction and fully supports our previous conclusions. We include hereby the detailed methods now included in the manuscript.

Line 1241-1254 (M&Ms > Behaviour > eye tracking): “In order to measure eye movements in freely-moving mice we used a custom head-mounted eye and head tracking system, previously

described^{2,3}. Briefly, we used a commercially available camera module (1937, Adafruit, USA; infrared filter removed). A custom 3D printed camera holder with a 21G cannula (Coopers Needle Works, UK) was used to hold the camera, IR LEDs (VSMB2943GX01, Vishay, USA) and a 7.0 mm x 9.3 mm IR mirror (Calflex-X NIR-Blocking Filter, Optics Balzers, Germany). A connector (852-10-00810-001101, Preci-Dip, Switzerland) was used to attach the camera holder to the head plate of the mice. Mice were head-fixed, and the mirror's position was adjusted until the eye was in the centre of the eye camera. Epoxy (Araldite Rapid, Araldite, UK) was used to fix the mirror position. Single-board computers (Raspberry Pi 3 model B, Raspberry Pi Foundation, UK) recorded camera data at 30 Hz, capturing images of 1296 x 972 pixels per frame for eye camera. The head roll, pitch, and yaw were estimated using an inertial motion unit (IMU) including an accelerometer, gyroscope and magnetometer using previously described methods⁴ and open source arduino code (github.com/razor-AHRS) using an Arduino Mega 2560 rev 3.”

Line 1373-1382 (M&Ms > Quantification and Statistical Analysis > Extraction of pupil position in freely-moving mice): “Eye tracking was performed as previously described². Briefly, we tracked the position of the pupil, defined as its center, together with the nasal and temporal eye corners. The eye corners were used to automatically align the horizontal eye axis. 30-50 randomly selected frames were labeled manually for each recording day. The labeled data were used to train a deep convolutional network via transfer learning using open source code⁵ (github.com/AlexEMG/DeepLabCut). The origin of the eye coordinate system was defined as the mid-point between the nasal and temporal eye corners. Pixel values in the 2-D video plane were converted to angular eye positions using a model-based approach developed for the C57BL/6J mouse line used in this study⁶. Saccades were defined as rapid, high-velocity movements occurring in both eyes with a magnitude exceeding 350 deg/s.”

Line 1478-1481 (M&Ms > Modelling > 3D head rotations to 2D screen plane transformation model): “The correction rotation matrix in head frame, $R_{correction}^H(\phi, \theta, \psi)$, was computed from the head-eye rotations measured and represented in Extended Data Figure 9f,g.”

EXTENDED DATA FIGURE 9

Extended Data Figure 9. Characterisation of the alignment of visuo-motor neurons.

[...] **f**, Simultaneous eye tracking and head pitch, roll and yaw measurement in freely moving mice foraging for milk drops (N = 4 mice). **g**, Horizontal (blue lines) and vertical eye position (red lines) as a function of head pitch for right eye in freely moving mice (left), head roll (middle) and head velocity during stabilisation periods (right). Eye movements between saccadic gaze shifts counteract head rotations; Plot shows mean \pm s.e.m. This was implemented in our geometrical model of the head and eye of the mouse that was used to transform 3D head rotations into gaze direction as projected onto the plane of the screen in which visual stimuli are shown. **h**, Gaze direction for the neuron in Fig. 3g after adding compensatory eye movements to the model. Both the trajectory on the screen (i) and the decomposed and average movement vectors are shown (ii). **i**, Angular difference between the gaze direction of all visuo-motor neurons estimated using the model without (model 1) and with (model 2) compensatory eye movements implemented.

In order to further assess whether our sensorimotor neurons could be a subpopulation of Pitx2^{ON} neurons and hence, not directly responsible for eye movements, we performed optrode recordings of Pitx2^{ON} neurons. An example of Pitx2^{ON} neurons displaying the uncovered sensorimotor anti-alignment is now included in Extended Data Figure 6.

EXTENDED DATA FIGURE 6

Extended Data Figure 6. Visual and motor tuning of motor Pitx2^{ON} and premotor Pitx2^{ON-PRE} neurons in the SC.

a, Example opto-tagged Pitx2^{ON} motor neurons. Peri-stimulus time histogram of 900 5 ms-long laser stimulations (right, bottom) and example raster plot of the 1st 300 stimulations are shown (right, top). **b**, Example STA of angular head displacements in yaw, pitch and roll for a Pitx2^{ON} motor neuron (i). The 3D STA was projected onto the 2D plane of the visual stimulation screen to draw the tuned direction of gaze. The decomposed and average gaze direction is shown in ii. **c**, Tuning to moving gratings of the neuron in b.

9. In line 253~254, the authors discovered half of all motor-tuned neurons across all layers were also tuned to moving gratings. However, on the contrary, in the foraging test above (line248~250), the motor tuning was not influenced by self-generated visual flow. Considering another half cluster of motor neurons tuning head movement only, we could not rule out the possibility that the no alteration of motor is on account of less data. Comprehensive evidence would be required.

We have now replicated the experiment in open field with enforced visual flow in 2 more mice. This data is included in Extended Data Figure 8. Our new results are consistent with our previous data.

EXTENDED DATA FIGURE 8

Extended Data Figure 8. Head rotation tuning changes during enforced visual flow foraging.

a, Mice were left to freely forage for droplets of condensed milk while visual flow was enforced by having either horizontal (for 5 min, black dots) or vertical (for 5 min, red open circles) black and white stripes across all 4 walls of the arena ($N = 4$, $n = 44$ tuned neurons out of 134 recorded neurons). **b**, Angular difference between the maximal average displacement angle for all neurons tuned in a white arena and their displacement angle with either horizontal or vertical stripes. **c**, Break-down of the results in **b**. The displacement angles for neurons tuned in Light (left) or Dark (right) for yaw (top), pitch (middle) or roll (bottom) is shown against their displacement angle with horizontal or vertical stripes. Shaded areas indicates a change of direction of the tuned cell. The triangle indicates the markers corresponding to the same neuron tuned both in light and dark conditions. The R^2 are shown for all conditions.

REFERENCES

- 1 Schiller, P. H. & Koerner, F. Discharge characteristics of single units in superior colliculus of the alert rhesus monkey. *J Neurophysiol* **34**, 920-936, doi:10.1152/jn.1971.34.5.920 (1971).
- 2 Meyer, A. F., O'Keefe, J. & Poort, J. Two Distinct Types of Eye-Head Coupling in Freely Moving Mice. *Curr Biol* **30**, 2116-2130 e2116, doi:10.1016/j.cub.2020.04.042 (2020).
- 3 Meyer, A. F., Poort, J., O'Keefe, J., Sahani, M. & Linden, J. F. A Head-Mounted Camera System Integrates Detailed Behavioral Monitoring with Multichannel Electrophysiology in Freely Moving Mice. *Neuron* **100**, 46-+, doi:10.1016/j.neuron.2018.09.020 (2018).
- 4 Wilson, J. J., Alexandre, N., Trentin, C. & Tripodi, M. Three-Dimensional Representation of Motor Space in the Mouse Superior Colliculus. *Curr Biol* **28**, 1744-1755 e1712, doi:10.1016/j.cub.2018.04.021 (2018).
- 5 Mathis, A. *et al.* DeepLabCut: markerless pose estimation of user-defined body parts with deep learning. *Nat Neurosci* **21**, 1281-+, doi:10.1038/s41593-018-0209-y (2018).
- 6 Sakatani, T. & Isa, T. PC-based high-speed video-oculography for measuring rapid eye movements in mice. *Neurosci Res* **49**, 123-131, doi:10.1016/j.neures.2004.02.002 (2004).
- 7 Feinberg, E. H. & Meister, M. Orientation columns in the mouse superior colliculus. *Nature* **519**, 229-232, doi:10.1038/nature14103 (2015).
- 8 Li, Y. T., Turan, Z. & Meister, M. Functional Architecture of Motion Direction in the Mouse Superior Colliculus. *Curr Biol* **30**, 3304-3315 e3304, doi:10.1016/j.cub.2020.06.023 (2020).
- 9 de Malmazet, D., Kühn, N. K. & Farrow, K. Retinotopic Separation of Nasal and Temporal Motion Selectivity in the Mouse Superior Colliculus. *Current Biology* **28**, 2961-+, doi:10.1016/j.cub.2018.07.001 (2018).
- 10 Fenno, L. E. *et al.* Comprehensive Dual- and Triple-Feature Intersectional Single-Vector Delivery of Diverse Functional Payloads to Cells of Behaving Mammals. *Neuron* **107**, 836-+, doi:10.1016/j.neuron.2020.06.003 (2020).
- 11 Ciabatti, E., González-Rueda, A., Mariotti, L., Morgese, F. & Tripodi, M. Life-Long Genetic and Functional Access to Neural Circuits Using Self-Inactivating Rabies Virus. *Cell* **170**, 382-+, doi:10.1016/j.cell.2017.06.014 (2017).
- 12 Lee, H. S. *et al.* Combining long-term circuit mapping and network transcriptomics with SiR-N2c. *Nat Methods* **20**, doi:10.1038/s41592-023-01787-1 (2023).
- 13 Ciabatti, E. *et al.* Genomic stability of self-inactivating rabies. *Elife* **12**, doi:ARTN e8345910.7554/eLife.83459 (2023).
- 14 Masullo, L. *et al.* Genetically Defined Functional Modules for Spatial Orienting in the Mouse Superior Colliculus. *Curr Biol* (2019).
- 15 Anikeeva, P. *et al.* Optetrode: a multichannel readout for optogenetic control in freely moving mice. *Nat Neurosci* **15**, 163-U204, doi:10.1038/nn.2992 (2012).
- 16 Giovannucci, A. *et al.* CalmAn an open source tool for scalable calcium imaging data analysis. *Elife* **8**, doi:ARTN e3817310.7554/eLife.38173 (2019).
- 17 Hoy, J. L., Bishop, H. I. & Niell, C. M. Defined Cell Types in Superior Colliculus Make Distinct Contributions to Prey Capture Behavior in the Mouse. *Current Biology* **29**, 4130-+, doi:10.1016/j.cub.2019.10.017 (2019).
- 18 Gale, S. D. & Murphy, G. J. Distinct representation and distribution of visual information by specific cell types in mouse superficial superior colliculus. *J Neurosci* **34**, 13458-13471, doi:10.1523/JNEUROSCI.2768-14.2014 (2014).
- 19 Tsai, N. Y. *et al.* Trans-Seq maps a selective mammalian retinotectal synapse instructed by Nephronectin. *Nat Neurosci* **25**, 659-+, doi:10.1038/s41593-022-01068-8 (2022).

- 20 Liu, Y. M. *et al.* Mapping visual functions onto molecular cell types in the mouse superior colliculus. *Neuron* **111**, doi:10.1016/j.neuron.2023.03.036 (2023).
- 21 Schiller, P. H. & Stryker, M. Single-unit recording and stimulation in superior colliculus of the alert rhesus monkey. *J Neurophysiol* **35**, 915-924, doi:10.1152/jn.1972.35.6.915 (1972).
- 22 Drager, U. C. & Hubel, D. H. Physiology of visual cells in mouse superior colliculus and correlation with somatosensory and auditory input. *Nature* **253**, 203-204, doi:10.1038/253203a0 (1975).
- 23 Sparks, D., Rohrer, W. H. & Zhang, Y. The role of the superior colliculus in saccade initiation: a study of express saccades and the gap effect. *Vision Res* **40**, 2763-2777, doi:10.1016/s0042-6989(00)00133-4 (2000).
- 24 Marino, R. A., Rodgers, C. K., Levy, R. & Munoz, D. P. Spatial relationships of visuomotor transformations in the superior colliculus map. *J Neurophysiol* **100**, 2564-2576, doi:10.1152/jn.90688.2008 (2008).
- 25 Sadeh, M., Sajad, A., Wang, H. Y., Yan, X. G. & Crawford, J. D. Spatial transformations between superior colliculus visual and motor response fields during head-unrestrained gaze shifts. *Eur J Neurosci* **42**, 2934-2951, doi:10.1111/ejn.13093 (2015).
- 26 Chen, C. Y., Hoffmann, K. P., Distler, C. & Hafed, Z. M. The Foveal Visual Representation of the Primate Superior Colliculus. *Current Biology* **29**, 2109-+, doi:10.1016/j.cub.2019.05.040 (2019).
- 27 Detsch, O., Kochs, E., Siemers, M., Bromm, B. & Vahle-Hinz, C. Differential effects of isoflurane on excitatory and inhibitory synaptic inputs to thalamic neurones. *Brit J Anaesth* **89**, 294-300, doi:DOI 10.1093/bja/aef170 (2002).
- 28 Ou, M. C. *et al.* The General Anesthetic Isoflurane Bilaterally Modulates Neuronal Excitability. *Iscience* **23**, doi:ARTN 10076010.1016/j.isci.2019.100760 (2020).
- 29 Wang, H. Y., Eguchi, K., Yamashita, T. & Takahashi, T. Frequency-Dependent Block of Excitatory Neurotransmission by Isoflurane via Dual Presynaptic Mechanisms. *Journal of Neuroscience* **40**, 4103-4115, doi:10.1523/Jneurosci.2946-19.2020 (2020).
- 30 Kretschmer, F., Tariq, M., Chatila, W., Wu, B. & Badea, T. C. Comparison of optomotor and optokinetic reflexes in mice. *Journal of Neurophysiology* **118**, 300-316, doi:10.1152/jn.00055.2017 (2017).
- 31 Badea, T. C., Cahill, H., Ecker, J., Hattar, S. & Nathans, J. Distinct Roles of Transcription Factors Brn3a and Brn3b in Controlling the Development, Morphology, and Function of Retinal Ganglion Cells. *Neuron* **61**, 852-864, doi:10.1016/j.neuron.2009.01.020 (2009).
- 32 Sun, L. O. *et al.* Functional Assembly of Accessory Optic System Circuitry Critical for Compensatory Eye Movements. *Neuron* **86**, 971-984, doi:10.1016/j.neuron.2015.03.064 (2015).
- 33 Osterhout, J. A., Stafford, B. K., Nguyen, P. L., Yoshihara, Y. & Huberman, A. D. Contactin-4 Mediates Axon-Target Specificity and Functional Development of the Accessory Optic System. *Neuron* **86**, 985-999, doi:10.1016/j.neuron.2015.04.005 (2015).
- 34 Ito, S., Feldheim, D. A. & Litke, A. M. Segregation of Visual Response Properties in the Mouse Superior Colliculus and Their Modulation during Locomotion. *J Neurosci* **37**, 8428-8443, doi:10.1523/JNEUROSCI.3689-16.2017 (2017).
- 35 Lee, K. H., Tran, A., Turan, Z. & Meister, M. The sifting of visual information in the superior colliculus. *Elife* **9** (2020).
- 36 Veale, R., Hafed, Z. M. & Yoshida, M. How is visual salience computed in the brain? Insights from behaviour, neurobiology and modelling. *Philos T R Soc B* **372**, doi:ARTN 2016011310.1098/rstb.2016.0113 (2017).
- 37 Fecteau, J. H. & Munoz, D. P. Saliency, relevance, and firing: a priority map for target selection. *Trends Cogn Sci* **10**, 382-390, doi:10.1016/j.tics.2006.06.011 (2006).
- 38 Itti, L., Koch, C. & Niebur, E. A model of saliency-based visual attention for rapid scene analysis. *Ieee T Pattern Anal* **20**, 1254-1259, doi:Doi 10.1109/34.730558 (1998).

- 39 Parkhurst, D., Law, K. & Niebur, E. Modeling the role of salience in the allocation of overt visual attention. *Vision Research* **42**, 107-123, doi:Doi 10.1016/S0042-6989(01)00250-4 (2002).
- 40 White, B. J. *et al.* Superior colliculus neurons encode a visual saliency map during free viewing of natural dynamic video. *Nature Communications* **8**, doi:ARTN 1426310.1038/ncomms14263 (2017).
- 41 Goldberg, M. E. & Wurtz, R. H. Activity of Superior Colliculus in Behaving Monkey .1. Visual Receptive Fields of Single Neurons. *Journal of Neurophysiology* **35**, 542-&, doi:DOI 10.1152/jn.1972.35.4.542 (1972).
- 42 Davidson, R. M. & Bender, D. B. Selectivity for Relative Motion in the Monkey Superior Colliculus. *Journal of Neurophysiology* **65**, 1115-1133, doi:DOI 10.1152/jn.1991.65.5.1115 (1991).
- 43 Marrocco, R. T. & Li, R. H. Monkey Superior Colliculus - Properties of Single Cells and Their Afferent Inputs. *Journal of Neurophysiology* **40**, 844-860, doi:DOI 10.1152/jn.1977.40.4.844 (1977).

Reviewer Reports on the First Revision:

Referee #1 (Remarks to the Author):

The authors have done an amazing job in addressing my (and the other reviewer's) concerns. They have performed several sets of new experiments, some of which are technically novel and very demanding. In many places they have changed the text to clarify ambiguities. The paper has been improved a whole lot, and it now makes a really important contribution to our understanding of sensory-motor mapping in the superior colliculus.

I have a few remaining points, mostly concerning display of the data.

1. Extended Data Figure 5 b and c. I think there might be a better way to display these data. Instead of showing the spatial layout of the RF and the preferred direction separately for the preferred and all locations, they could indicate the direction tuning for each location in the xy plot, using arrows whose direction and length indicate the preference and selectivity, respectively, at that position. It is also not explained what the numbers are, and what the red cross indicates (this is done only later in the Figure legend).
2. The plots in Extended Data Figure 5 j are very hard to read with their various color and area codes. Intervene should be briefly explained, the citation should be given here already.
3. Extended Figure 9 i is not very clear. What are the various symbols? Is there any sort-order?
4. Extended Data Figure 6. The legend is largely a repeat of parts of the methods section. This can be substantially shortened and/or should go to the methods.
5. In various places throughout the paper, when talking about visuo-motor alignment, the authors should carefully check whether they really mean to refer to the direction of gaze, or rather the change in the direction of gaze.

Referee #2 (Remarks to the Author):

The authors did an impressive amount of extra work and complex new experiments to try to thoroughly answer all our concerns and questions. For my questions, they managed to do so. I have no remaining major concerns. I have a few points that came up in the revised version.

A big question in the first version of the manuscript was the issue whether motor neurons did really not have spatially selective receptive fields, or were just not responsive to static spots. The new experiments summarized in Ext Data Fig 5h show that they indeed do not have spatially selective RFs. This is an important point made in the paper and contrasting the properties of these neurons in the deeper layers in the mouse with the properties of visuomotor neurons found in the primate. To me, it would be more appropriate to show this in a main figure.

P36: In the formula for $\bar{\theta}$, to me it seems that the $R(\theta_k)$ are missing and that it should be the reciprocal of the given formula (as $\tan(\phi) = \sin(\phi)/\cos(\phi)$).

L238: "Together these data show that premotor neurons in lower intermediate and deep layers of the SC receive preferential input reporting kinetic features of the visual scene, such as externally generated motion flow". I am not sure about how the word 'preferential' is meant? Can the authors clarify its meaning, and perhaps more explicitly link the evidence for its intended meaning. Figure 2q-r do not show a comparison of Pitx2-on-pre neurons to the general population of neurons, or a preference of these neurons to kinetic stimuli over static stimuli, or the absence of static receptive fields in these neurons.

L471: "Essentially, the logic governing sensorimotor transformation in the SC is not primarily driven by the spatial mapping of visual receptive fields and moment vector endpoint, as commonly hypothesised" This seems to be in contrast to what the authors write in L330: "We also found an alignment between gaze direction and preferred Gabor patch location for neurons tuned to head rotations and Gabor patches (Extended Data Fig. 9k)". Is the correlation they show in Extended Data Fig. 9k not in line with the previously hypothesized spatial mapping of RFs and gaze rotation?

L328: "The same anti-alignment was found in genetically-defined Pitx2ON motor neurons (Extended Data Fig. 6a-c)" For this statement, I miss the evidence. In Ext. Data Fig 6a-c, I see just the results of one neuron, which seems to have a difference between head-rotation and visual preference somewhere between alignment and anti-alignment. Please clarify or remove the statement.

L992: "Pitx-2-Flp, Jackson Laboratories". Can you add the Jackson Lab strain numbers? I could not find the strain number for the Pitx2-cre line on the Jackson website. Can you confirm that the mice are pigmented (in contrast to the white mice depicted in all the figure panels) or on which background they were bred.

L475-495. This text seems to suggest that this manuscript shows that "broadly accepted model of a mapping of spatial static visual receptive field onto movement vector end point" is wrong, but the data they show in Extended Data Fig. 9k is that, also in the mouse, there might still be some validity to this model, if the receptive fields are mapped with kinetic features. Perhaps the authors can

comment on this, if they agree with me.

Ext Data Fig 5. What does “Centre of gaze” mean? How was this determined? Is this a fixed direction related to the mouse head?

Ext Data Fig 6e,h. Add in the caption that the red cross (presumably) denotes the center of gaze, as in Ext Data Fig 5.

Referee #3 (Remarks to the Author):

The sensorimotor transformation of converting sensory stimuli into motor command in superior colliculus has been well delineated in orchestrating behavioral actions such as predator avoidance and prey capture. SC is topographically-organized into several laminae with convergence of multi-modal sensory inputs. To be more specific, the superficial layer of SC is innervated by retinal ganglion cells; the intermediate/deep layers receive afferents from somatosensory and auditory sources. Intriguingly, the deep layers are involved in eye movement, head orientation and locomotion. However, the internal integration of intra-collicular network about sensorimotor transformation remains to be interrogated systematically. This manuscript provides compelling evidence to dissect a microcircuit from premotor neurons (Pitx2 on-pre) to Pitx2 on motor neurons preferentially processing kinetic visual flows opposite to head movement. Furthermore, the authors propose a novel kinetic alignment model which decodes potentially goal-orientation behavior aiming to claim the functional alignment between sensory and motor maps unambiguously. The figures are nicely laid out and the data are clearly summarized. The technology using in this study, including RV tracing strategy, two photon imaging, whole-cell recording and tetrode recording, make the story meaningful and convincing. This manuscript fully interpret the concept that every behavior is a phenotypical manifestation of some well-orchestrated network activity and greatly enriches our understanding of sensorimotor transformation in superior colliculus in a more general sense. Overall, these are novel findings that will advance the field.

Author Rebuttals to First Revision:

Referees' comments:

Referee #1 (Remarks to the Author):

The authors have done an amazing job in addressing my (and the other reviewer's) concerns. They have performed several sets of new experiments, some of which are technically novel and very demanding. In many places they have changed the text to clarify ambiguities. The paper has been improved a whole lot, and it now makes a really important contribution to our understanding of sensory-motor mapping in the superior colliculus.

I have a few remaining points, mostly concerning display of the data.

1. Extended Data Figure 5 b and c. I think there might be a better way to display these data. Instead of showing the spatial layout of the RF and the preferred direction separately for the preferred and all locations, they could indicate the direction tuning for each location in the xy plot, using arrows whose direction and length indicate the preference and selectivity, respectively, at that position. It is also not explained what the numbers are, and what the red cross indicates (this is done only later in the Figure legend).

We have now updated Figure 5 b and c with the suggested changes. We have included an arrow for each location indicating the preferred tuning of that neuron at that location.

2. The plots in Extended Data Figure 5 j are very hard to read with their various color and area codes. Intervene should be briefly explained, the citation should be given here already.

We have increased the size of this plots to make them more readable and added the reference as suggested. We have also included the following explanatory sentence: "Similarly to a Venn diagram, this plot illustrates the logical relation between sets, with darker areas indicating a larger number of neurons at the intersection of sets."

3. Extended Figure 9 i is not very clear. What are the various symbols? Is there any sort-order?

The symbols used in Extended Data Figure 9 are the same used in Figure 3 and Extended Data Figures 9a-d. With the sun symbol indicating neurons tuned to head rotations in light conditions, moon symbols indicating neurons tuned to head rotations in dark conditions, and a mixed symbol indicating neurons tuned in both conditions. Furthermore, we have re-coloured the circular markers as in Extended Data Figure 9c for clarity. We only grouped the neurons for display but they are not sorted depending on their angular difference.

4. Extended Data Figure 6. The legend is largely a repeat of parts of the methods section. This can be substantially shortened and/or should go to the methods.

This has now been shortened as suggested.

5. In various places throughout the paper, when talking about visuo-motor alignment, the authors should carefully check whether they really mean to refer to the direction of gaze, or rather the change in the direction of gaze.

We agree with the reviewer that this term can be confusing as we refer to a direction of gaze in a 2D projected plane occurring during a head movement. In order to avoid misunderstandings we have specified in the manuscript that we refer to a change in the direction of gaze when appropriate.

Referee #2 (Remarks to the Author):

The authors did an impressive amount of extra work and complex new experiments to try to thoroughly answer all our concerns and questions. For my questions, they managed to do so. I have no remaining major concerns. I have a few points that came up in the revised version.

A big question in the first version of the manuscript was the issue whether motor neurons did really not have spatially selective receptive fields, or were just not responsive to static spots. The new experiments summarized in Ext Data Fig 5h show that they indeed do not have spatially selective RFs. This is an important point made in the paper and contrasting the properties of these neurons in the deeper layers in the mouse with the properties of visuomotor neurons found in the primate. To me, it would be more appropriate to show this in a main figure.

We would like to thank the reviewer for their suggestion. We agree that the new results obtained using drifting Gabor patches further strengthen the point made in Figure 2 and could well be included in the main figure. However, upon consideration and in order to maintain the flow and narrative of the paper simple, we opted against. While these results are insightful, we think that including stimulation paradigms that have not been consistently used across conditions in the main figures could distract from the main message of the paper. Ultimately, since those results do not change our original conclusions but only strengthen them, we feel they are still well suited for Extended Data Figure material.

P36: In the formula for $\bar{\theta}$, to me it seems that the $R(\theta_k)$ are missing and that it should be the reciprocal of the given formula (as $\tan(\phi) = \sin(\phi)/\cos(\phi)$).

We want to thank the reviewer for spotting the mistake. The reviewer is correct and the formula has been updated, also including the corrections for negative cosine and sine values.

$$\bar{\theta} = \text{atan} \left(\frac{\sum_k R(\theta_k) \sin \theta_k}{\sum_k R(\theta_k) \cos \theta_k} \right)$$

for $\cos \theta_k < 0$,

$$\bar{\theta} = \text{atan} \left(\frac{\sum_k R(\theta_k) \sin \theta_k}{\sum_k R(\theta_k) \cos \theta_k} \right) + \pi$$

for $\sin \theta_k < 0$ and $\cos \theta_k > 0$,

$$\bar{\theta} = \text{atan} \left(\frac{\sum_k R(\theta_k) \sin \theta_k}{\sum_k R(\theta_k) \cos \theta_k} \right) + 2\pi$$

L238: "Together these data show that premotor neurons in lower intermediate and deep layers of the SC receive preferential input reporting kinetic features of the visual scene, such as externally generated motion flow". I am not sure about how the word 'preferential' is meant? Can the authors clarify its meaning, and perhaps more explicitly link the evidence for its intended meaning. Figure 2q-r do not show a comparison of Pitx2-on-pre neurons to the general population of neurons, or a preference of these neurons to kinetic stimuli over static stimuli, or the absence of static receptive fields in these neurons.

Since we are recording the output of these neurons we cannot completely rule out the possibility that they would receive some visual spatial information; however, since their output is tuned to moving visual stimuli rather than static ones, we can conclude that the input they receive reports preferentially information about visual kinetic features over static ones.

Due to technical constraints, we could not record bona fide non-Pitx2^{ON-PRE} neurons together with Pitx2^{ON-PRE} neurons. The only possible comparison is between independently recorded populations of all neurons in WT mice (Figure 2n and Extended Data Figure 5h) and the Pitx2^{ON-PRE} neurons (Figure 2q and Extended Data Figure 6i). We observed a loss of spatial visual encoding in lower intermediate and deep layers of the SC both in WT mice (measured with flashing spots and localised drifting Gabor patches, Figure 2n and Extended Data Figure 5h) and in Pitx2^{ON-PRE} (measured with localised Gabor patches, Extended Data Figure 6i).

L471: “Essentially, the logic governing sensorimotor transformation in the SC is not primarily driven by the spatial mapping of visual receptive fields and moment vector endpoint, as commonly hypothesised” This seems to be in contrast to what the authors write in L330: “We also found an alignment between gaze direction and preferred Gabor patch location for neurons tuned to head rotations and Gabor patches (Extended Data Fig. 9k)”. Is the correlation they show in Extended Data Fig. 9k not in line with the previously hypothesized spatial mapping of RFs and gaze rotation?

The following sentence from the manuscript recapitulates our interpretation on this point: “Over 90% of these units maintained their motor tuning only in light conditions but lost their head-rotation tuning in darkness, implying that, although these are not canonical motor units, they are reminiscent of visuo-motor units found in primates”. It is particularly important to note that those neurons were not canonical motor units since they were not tuned to head rotations in darkness. Hence, the only alignment that we observe for true motor units (e.g. tuned in both light and dark) remains the kinetic one that we describe in the manuscript.

L328: “The same anti-alignment was found in genetically-defined Pitx2ON motor neurons (Extended Data Fig. 6a-c)” For this statement, I miss the evidence. In Ext. Data Fig 6a-c, I see just the results of one neuron, which seems to have a difference between head-rotation and visual preference somewhere between alignment and anti-alignment. Please clarify or remove the statement.

The example neuron shown in Extended Data Figure 6 had an alignment of 167 degrees, which is within the CI of the measured anti-alignment in WT mice (Extended Data Figure 9d).

L992: “Pitx-2-Flp, Jackson Laboratories”. Can you add the Jackson Lab strain numbers? I could not find the strain number for the Pitx2-cre line on the Jackson website. Can you confirm that the mice are pigmented (in contrast to the white mice depicted in all the figure panels) or on which background they were bred.

We thank the reviewer for highlighting this point. To generate Pitx2-Flp mice, Pitx2-cre and Tau-LSL-Flp0-INLA mice were bred. The correct strains and origins are now updated on the Methods section “*Pitx2-CRE::Tau-LSL-Flp0-INLA* (Pitx2-Flp, derived from *Pitx2-CRE* and *Tau-LSL-Flp0-INLA* mice,

provided by Prof. James Martin and Prof. Silvia Arber respectively)". We can confirm the mice are indeed pigmented.

L475-495. This text seems to suggest that this manuscript shows that "broadly accepted model of a mapping of spatial static visual receptive field onto movement vector end point" is wrong, but the data they show in Extended Data Fig. 9k is that, also in the mouse, there might still be some validity to this model, if the receptive fields are mapped with kinetic features. Perhaps the authors can comment on this, if they agree with me.

We propose a kinetic alignment framework which also recognises the existence of a localised RF map in superficial layers and predicts a ssRF-to-DS-to-motor alignment. We believe this alignment fits well with early studies on sensorimotor transformation in the SC. However, these earlier studies would have focussed mostly on the ssRF or the motor layers in trained animals, overlooking the kinetic alignment we have found. We believe that we have uncovered a novel dimension of the sensorimotor alignment process.

Ext Data Fig 5. What does "Centre of gaze" mean? How was this determined? Is this a fixed direction related to the mouse head?

The "centre of gaze" was calculated geometrically as the projection of the centre of the right eye of the mouse onto the screen.

Ext Data Fig 6e,h. Add in the caption that the red cross (presumably) denotes the center of gaze, as in Ext Data Fig 5.

This has now been included.

Referee #3 (Remarks to the Author):

The sensorimotor transformation of converting sensory stimuli into motor command in superior colliculus has been well delineated in orchestrating behavioral actions such as predator avoidance and prey capture. SC is topographically-organized into several laminae with convergence of multi-modal sensory inputs. To be more specific, the superficial layer of SC is innervated by retinal ganglion cells; the intermediate/deep layers receive afferents from somatosensory and auditory sources. Intriguingly, the deep layers are involved in eye movement, head orientation and locomotion. However, the internal integration of intra-collicular network about sensorimotor transformation remains to be interrogated systematically. This manuscript provides compelling evidence to dissect a microcircuit from premotor neurons (Pitx2 on-pre) to Pitx2 on motor neurons preferentially processing kinetic visual flows opposite to head movement. Furthermore, the authors propose a novel kinetic alignment model which decodes potentially goal-orientation behavior aiming to claim the functional alignment between sensory and motor maps unambiguously. The figures are nicely laid out and the data are clearly summarized. The technology using in this study, including RV tracing strategy, two photon imaging, whole-cell recording and tetrode recording, make the story meaningful and convincing. This manuscript fully interpret the concept that every behavior is a phenotypical manifestation of some well-orchestrated network activity and greatly enriches our understanding of sensorimotor transformation in superior colliculus in a more general sense. Overall, these are novel findings that will advance the field.